# A novel method for the extraction of local gravity wave parameters from gridded three-dimensional data: description, validation and application

Lena Schoon[1] and Christoph Zülicke[1]

[1]Leibniz-Institute of Atmospheric Physics, Schlossstrasse 6, 18225 Kühlungsborn, Germany

*Correspondence to:* Lena Schoon (schoon@iap-kborn.de)

**Abstract.** For the local diagnosis of wave properties we develop, validate and apply a novel method which is based on the Hilbert transform. It is named "Unified Wave Diagnostics" (UWaDi). It provides the wave amplitude and three-dimensional wave number at any grid point for gridded three-dimensional data. UWaDi is validated for a synthetic test case comprising two different wave packets. In comparison with other methods, the performance of UWaDi is very good with respect to wave properties and their location. For a first practical application of UWaDi, a minor sudden stratospheric warming on 30 January 2016 is chosen. Specifying the diagnostics for hydrostatic inertia-gravity waves in analyses from the European Centre for Medium-Range Weather Forecasts, we detect local occurence of gravity waves throughout the middle atmosphere. The local wave characteristics are discussed in terms of vertical propagation using the diagnosed local amplitudes and wave numbers. We also note some hint on local inertia-gravity wave generation by the stratospheric jet from the detection of shallow slow waves in the vicinity of its exit region.

## 1 Introduction

The importance of gravity waves (GWs) for the dynamics of the Earth's atmosphere is without controversy. They influence dynamics from planetary scales to turbulent microscales and play an important role in the middle atmosphere circulation (Fritts and Alexander, 2003). The GWs typically appear as packets localised in space and time. Hence, it is desirable to diagnose them locally as precise as possible. Here, we want to introduce a new method named "Unified Wave Diagnosis" (UWaDi). It provides phase-independent local wave quantities like amplitude and wave number without any prior assumption. In the following, we develop, validate and apply this novel method. The application concentrates on the analysis of GWs for locally varying background wind conditions in the winter 2015/16.

In the past, several methods were developed to estimate wave properties like amplitudes and wave number vectors. All of them have to deal with the fact that the data sampling procedure influences the results. A common approach to obtain vertical wave numbers and GW frequency of high-passed filtered wind fluctuations are Stokes parameters (Vincent and Fritts, 1987). This method is based on the polarisation relations and works for single-column measurements. It provides the wave properties in preselected vertical height sections of finite lengths. Next to its original application on radar measurements it is used for radiosonde data (Kramer et al., 2015). A supplement to this method named DIV was introduced by Zülicke and Peters (2006). It

determines the dominating harmonic wave in a box from the first zero-crossing of the auto-correlation function. The maximal detectable wavelength is restricted by the box size. The analysed quantity is the horizontal divergence to get the ageostrophic flow without numerical filtering. A further technique is based on sinusoidal few wave fits (S3D) (Lehmann et al., 2012). This method was created for the analysis of three-dimensional data from remote sensing observations (Ern et al., 2017; Krisch et al., 2017) but is also applicable to model data (Preusse et al., 2014). The first modes with highest variance are taken from a fit that minimises the variance-weighted squared deviations over all points in a finite analysis box. Only a small number of sinusoidal curves are fitted and there might remain uncovered variances in the analysis volume. These methods have in common, that the analysed spatial scales are dependent on the predefined analysis box size and the assumption of spatial homogeneity of the wave field in these boxes is essential. Nevertheless, these methods are superior to a classic Fourier transform in that point that they allow to search for waves with bigger wavelengths than the box size.

Another three-dimensional spectral analysis method is the 3D Stockwell-transform (3D ST) (Wright et al., 2017). This method is capable of analysing the full range of length scales sampled in satellite data and is not restricted to fixed box sizes. At every grid point, a local wave spectrum is estimated using a window function of frequency-dependent width. S3D and 3D ST assume homogeneity inside the analysis box. Furthermore, they use a small number of the most-prominent waves for the estimation of variances. This leaves some variance unattributed and hence means a loss of information. We search for a method which detects the full variance in each data point.

With UWaDi we find the dominating wave with the Hilbert transform at every data point. This approach does not rely on choosing the size of an analysis volume aforehand. The calculation of wave quantities at every grid point is computationally cheap. There is no need of assuming homogeneity and no restriction to detectable wavelengths besides the Nyquist wavelength. Here, the method is developed to work with three-dimensional equally-gridded data. In general, the Hilbert transform can be applied to data of any dimensionality. Wave properties such as the amplitude and wave number are estimated phase-independently while all variance is attributed to one wave mode. Every variable including any kind of wave-like structure can be diagnosed. Zimin et al. (2003) used the method to obtain the envelope of a train of Rossby waves in one dimension. A supplement was made for waves not in-line with grids by an extension of the formulation to stream lines to obtain quasi-one-dimensional wave packets (Zimin et al., 2006). (Sato et al., 2013) provide a three-dimensional application on Rossby and GWs. Glatt and Wirth (2014) use the Hilbert-transform to identify Rossby wave trains on a longitude-time plane and introduce their approach as an "objective identification method". Our method focuses on the local site of GW occurence and the additionally provision of the wave number in every dimension. The latter was not presented before. We aim to cover the retrieval of local wave properties from arbitrarily orientated wave packets. Amplitude and wave number are returned on the same grid as the input data. After the mathematical description of the method and how it is implemented, it will be validated with synthetic data to demonstrate its quality in comparison with other methods.

For a demonstration of a practical application in geophysical context, we will investigate GWs. Their sources are usually found in the troposphere where waves are generated by flow over orography, by convection, frontal systems and jet imbalances. These waves propagate upwards with increasing amplitudes and break in the middle atmosphere where they deposit their momentum to the background flow. Strong influence is exerted on global circulation patterns in the mesosphere as well as in the stratos-

phere (Holton, 1983; Garcia and Solomon, 1985). GWs play crucial roles in the modulating of the quasi-biennial oscillation (QBO) and the Brewer-Dobson circulation (Dunkerton, 1997; Alexander and Vincent, 2000; Ern et al., 2014). Another stratospheric phenomenon where GWs play a role are sudden stratospheric warmings (SSW). For this phenomenon, a variety of definitions exists (Butler et al., 2015), but the most common one is given by the World Meteorological Organization stating

that an SSW is characterised by a reversal of the $60°$ N to $90°$ N-temperature gradient. Major warmings are associated with a wind reversal at $10\,\mathrm{hPa}$ and $60°$ N; minor SSWs (mSSWs) with a wind deceleration at $10\,\mathrm{hPa}$ and $60°$ N, where the prevailing westerlies are not turned into easterlies. While planetary waves are the most important drivers of SSWs (Andrews et al., 1987), GWs are affected by the differing background wind conditions during SSWs and are suspected to modulate the polar vortex in the postwarming phase of an SSW (Albers and Birner, 2014). The behaviour of GWs and planetary waves during an SSW was

investigated by simulations and different measurement techniques. It was found that these GWs are, next to selective transmission, subjects of variable sources including unbalanced flow adjustment (Yamashita et al., 2010; Limpasuvan et al., 2011). We are interested in the longitude-dependent transmission of GWs during an SSW. Pioneering work was done by Dunkerton and Butchart (1984). They analysed model data and found that selective transmission of GWs during an SSW is dependent on longitude according to the planetary wave structures. Therefore, regions where vertical wave propagation is inhibited exist as

well as regions where waves can propagate up to the mesosphere. The analysis of Dunkerton and Butchart (1984) was restricted to parameterised GWs of the "intermediate range", that they defined between $50\,\mathrm{km}$ and $200\,\mathrm{km}$. They state that it remains unclear, in what kind GWs of larger scale will act during SSWs. A study on a self-generated SSW in a model showed that GWs reverse the circulation in the mesosphere-lower thermosphere during an SSW by altering the altitude of GW breaking. This altitude is highly dependent on the specification of GW momentum flux in the lower atmosphere (Liu and Roble, 2002;

Zülicke and Becker, 2013). In the UWaDi application, we want to locate the occurrence of GWs precisely in space and give first interpretations using the information on their changing amplitude and wave number in local vertical profiles.

The northern winter 2015/16 brought up several interesting features, including specific GW patterns. The beginning of the winter was characterised by an extraordinarily strong and cold polar vortex driven by a deceleration of planetary waves in November/December 2015 (Matthias et al., 2016). Thereinafter, for the end of that winter a record Arctic ozone loss was

expected (Manney and Lawrence, 2016). Furthermore, the extraordinarily polar vortex caused a southward shift of planetary waves leading to anomalies in the QBO (Coy et al., 2017). Inbetween, a joint field campaign of the research projects METROSI, GW-LCYCLE2 (both part of ROMIC) and PACOG (MS-GWaves) took place in Scandinavia in January 2016. Stober et al. (2017) found a summer-like zonal wind reversal in the upper mesosphere lasting until the end of January 2016, leading to different GW filtering processes in the mesosphere compared to usual winter-like wind conditions. During the field cam-

paign first tomographic observations of GWs by an infrared limb imager provide a full three-dimensional picture of a GW packet above Iceland (Krisch et al., 2017). Additionally, a remarkable comparative study shows that forecasts of the current operational cycle (41r2) of the European Centre for Medium-Range Weather Forecasts (ECMWF) Integrated Forecast System (IFS) shows good accordance with space-borne lidar measurements while picturing large-scale and mesoscale wave structures in polar stratospheric clouds (Dörnbrack et al., 2017). We choose the mid-winter of 2016 for an first application of UWaDi

because it is very well sampled with observations of GW properties. We intend to provide additional impulses to the evaluation

of observations and start with a study of ECMWF analyses.

These gridded data are suitable to analyse the local occurence and their coupling as the analyses resolve essential parts of GW dynamics in the stratosphere. Validation studies with satellite measurements point out that ECMWF analyses capture GWs well in the mid- and high-latitudes (Yamashita et al., 2010; Preusse et al., 2014). Especially mid-latitude GWs are captured well being driven by orographic and jet-stream associated sources (Shutts and Vosper, 2011; Jewtoukoff et al., 2015). Our approach concentrates on fields of horizontal divergence of ECMWF IFS data. By choosing the horizontal divergence as the analysis variable the calculation of residuals is omitted. Hence, an explicit separation of a slowly varying background wind which may include further uncertainties is avoided. The horizontal divergence is a dynamical indicator for GWs because it is free from geostrophically balanced structures (Plougonven et al., 2003; Zülicke and Peters, 2006). Studying mountain waves, Dörnbrack et al. (2012) and Khaykin et al. (2015) have shown that the divergence values correlate with the GW-induced temperature anomalies. Furthermore, we use the horizontal divergence to derive the total wave energy in Appendix B from the polarisation relations. In this study we concentrate on vertical profiles of GW occurrence to give a first impression of the functionality of this method. Recently, the importance of oblique propagation of GWs in a general context was discussed by Yamashita et al. (2013), Kalisch et al. (2014) and Ehard et al. (2017). We point out that meridional propagation of GWs is important for the deposition of GW drag from the stratosphere up into the mesosphere. For instance, Krisch et al. (2017) discuss the role of oblique propagation for the redistribution of selective transmission of GWs in the upper troposphere and lower stratosphere. These are phenomena which require a detailed analysis of the propagation of localised wave packets.

Here, the authors focus on the introduction of the novel method and give first preliminary scientific results from a demonstrative application. We show locally diagnosed GW properties and give some hints on physical interpretation. A full three-dimensional spatial-temporal analysis of GWs during the SSW 2015/16 goes beyond the scope of this paper and will be made subject of subsequent publications.

The paper is organised as follows. After providing a step-by-step introduction and validation of the novel method in Section 2, we give a short overview of the estimation of wave quantities for synthetic data and describe the analysis data. In Section 3 we show our results for the application on the mSSW on 30 January 2016 where we study longitude-depending GW occurence. The discussion of our results in Section 4 is followed by the Summary and Conclusion (Sec. 5).

## 2  Method and Data

In this section we develop and validate an algorithm to extract wave parameters from gridded three-dimensional data. For local diagnosis of waves, phase-independent estimates of wave amplitudes as well as the wave vector are essential. For this, we employ the Hilbert transform (e.g. Von Storch and Zwiers, 2001). The Hilbert transform shifts any sinusoidal wave structure by a quarter phase, i.e. turning a sine into a cosine. By constructing a new complex-valued data series consisting of the original field as real part and its Hilbert transform as the imaginary part, the absolute value is always the amplitude (square root of squared real and imaginary part). The amplitude is independent of the phase of the wave and the wavelength of the

underlying oscillation and there is no need of any explicit fitting of a particular wave. In addition, the absolute wave number in all three dimensions is determined from the phase gradient. The only requirement of the method to work is that the data contain any harmonic component. Thus, it works universally for any given variable, hence it was named Unified Wave Diagnostics.

## 2.1 Step-by-step outline of the method

In the following we introduce UWaDi by a step-by-step outline. Further, we validate it with a well-defined test wave packet in comparison with other methods. In general, UWaDi is a script package which allows the user to steer data preprocessing, the main wave analysis and data plotting, from a set of namelists.

1. UWaDi requires data from equidistant grids. For the ECMWF analyses on a longitude-latitude grid, the latitude-dependence of grid distance is taken into account by determining the longitudinal grid distance by $dx = 2\pi r_\theta \frac{\gamma}{360}$ where $r_\theta = R\cos(\theta)$ is the latitude-depending earth radius ($R = 6371$km - earth radius; $\theta$ - latitude [°]) and $\gamma$ denotes the resolution of the gridded input data (e.g. $\gamma = 0.36°$).

2. To retrieve vertical equidistant levels firstly, the hybrid levels of the ECMWF data are transformed to pressure levels. Secondly, these pressure levels are assigned to equidistant height levels. For this purpose, we assume hydrostatic conditions and consider the surface geopotential and pressure as well as temperature and humidity. Both steps are performed with the help of common functions provided in the NCAR command language (NCL). This might cause problems in areas of high orography and inside the planetary boundary layer. These areas are not considered in the following analysis.

3. The method can handle any kind of variables. For the present application, we choose the horizontal divergence. While this quantity was available in the ECMWF analyses, other data source might require its calculation from the wind fields. However, the required preprocessing of the target variable is done in this step.

4. The underlying Hilbert transform is implemented with a Discrete Fourier Transform (DFT) which creates a complex spectrum in wave number space from the real valued data in real space (e.g. Smith et al. (1997)). The processing of the three-dimensional data in the $(x, y, z)$-space is begun with the $x-$direction. The mathematics behind the Hilbert transform is described briefly for an one-dimensional function $f_x$ originating in real space:

$$f_k = \text{DFT}(f_x) \tag{1}$$

The index $_k$ denotes the wave number space, $_x$ describes the function $f$ in real space.

5. DFTs can be biased by variance leakage through side lobes in spectral space. Tapering methods abandon this but can smear out nearby wave numbers. A loss of absolute amplitude can be overcome by using normalised weights (e.g. Von Storch and Zwiers, 2001). For the present study, however, the best results were obtained by turning the taper off.

6. In wave number space a rectangular bandpass filter reduces the complex spectrum to the user-predefined wave number limits $k_{min}$ and $k_{max}$. Here, we make sure that only waves of the considered range of wave numbers are used for the following analysis.

$$f_{k,filtered} = F(k_{min}, k_{max})f_k. \tag{2}$$

7. To get back from wave number space an inverse DFT is performed.

$$\hat{f}_x = 2 * \mathrm{DFT}^{-1}(f_{k,filtered}). \tag{3}$$

8. The such constructed complex-valued function $\hat{f}_x$ consists of the input data $f_x$ as the real part and the Hilbert-transformed function $H(f_x)$ as the imaginary part

$$\hat{f}_x = f_x + iH(f_x). \tag{4}$$

It provides the amplitude $a_x$ (Schönwiese, 2013)

$$a_x = |\hat{f}_x| = \sqrt{f_x{}^2 + H(f_x)^2} \tag{5}$$

and the phase estimate $\Phi_x$

$$\Phi_x = \mathrm{atan}\left(\frac{H(f_x)}{f_x}\right). \tag{6}$$

9. The phase gradient is a measure of the wave number modulus

$$k_x = \frac{d\Phi_x}{dx} \approx \frac{\left|\mathrm{DFT}^{-1}(k\mathrm{DFT}\hat{f}_x)\right|}{|\hat{f}_x|}. \tag{7}$$

10. Due to the finite character of the data series it may happen that high-frequency spurious fluctuations appear after the Hilbert transform. We damp them by applying a low-pass filter. We smooth over a number of grid points determined by the lower wave number limit $k_{min}$.

11. Identification of outliers is taken care of by two different quality checks. Firstly, the amplitude and wave number are checked for at least a half undamped wave. Therefore, the packet length $l_x$ is essential. It is calculated by covariance functions $C_{xx}$:

$$l_x = \sum_{x=0}^{x_{max}} \left|\frac{C_{xx}}{C_{00}}\right| \tag{8}$$

with $x_{max} = \frac{N-1}{5}$ (Chatfield, 2016). This method goes back to Zülicke and Peters (2006). The quality check then is defined by the inequality

$$k_x l_x > \pi. \tag{9}$$

Secondly, the retrieved signals are required to lie above the noise level of the input data. An empirical threshold $c$ checks the amplitude for being valid considering the standard deviation of the input function $\sigma(f_x)$

$$a_x > c * \sigma(f_x). \tag{10}$$

Empirically, we use $c = 0.01$. This idea follows Glatt and Wirth (2014).

UWaDi uses a quality flag $q = 1$ which is set to false ($q = 0$) if at least one quality check is rejected.

12. Steps 4 to 9 are repeated for the other dimensions $(y, z)$.

13. Amplitude and absolute wave number are saved on the same grid as the input data to create a full three-dimensional analysis of local wave quantities. The amplitude is combined to a wave number-weighted sum of the three spatial dimensions

$$a_{(x,y,z)} = \left( \frac{\sum_{d=x,y,z} q_d k_d^2 a_d^2}{\sum_{d=x,y,z} q_d k_d^2} \right)^{\frac{1}{2}}. \tag{11}$$

The absolute wave number is determined by

$$k_{(x,y,z)} = \left( \sum_{d=x,y,z} q_d * k_d^2 \right)^{\frac{1}{2}}, \tag{12}$$

with $d$ denoting the spatial index.

The method provides an exact measure of the amplitude in the sense of the sum of squared amplitudes of the wave modes. The dominating wave number is the amplitude weighted sum of all. Spectrally wide dynamics can cause a significant reduction of information (Appendix A). Applying UWaDi with several narrow band-pass limits would provide information on spectrally spread waves. The wave numbers are estimated as moduli, that is: the three-dimensional wave number $(\pm k_x, \pm k_y, \pm k_z)$ allows for eight possible directions. However, the method is recommended for the first guess of the dominant wave packet including the derivation of the intrinsic frequency from the dispersion relation.

## 2.2 Validation of the method

For a comparison of wave characteristics obtained with different methods we choose the test case presented in Zimin et al. (2003) (Fig. 1a). In this exercise, a couple of localised wave packets with the wave numbers 4 and 9 is given in one dimension on the interval $[0, 4\pi]$ by

$$f_x = \exp\left(-(x - 4.5)^2\right) \cos(4x) + \exp\left(-(x - 7.5)^2\right) \cos(9x). \tag{13}$$

Here, the quality check (step 11) requires the amplitudes to exceed half of the sample standard deviation.

UWaDi based on a Hilbert transform is a continuous method working without any box parameter. 3D ST, S3D and DIV need box-width parameters to be adapted to the corresponding scientific case. A compromise between accuracy in space and wave

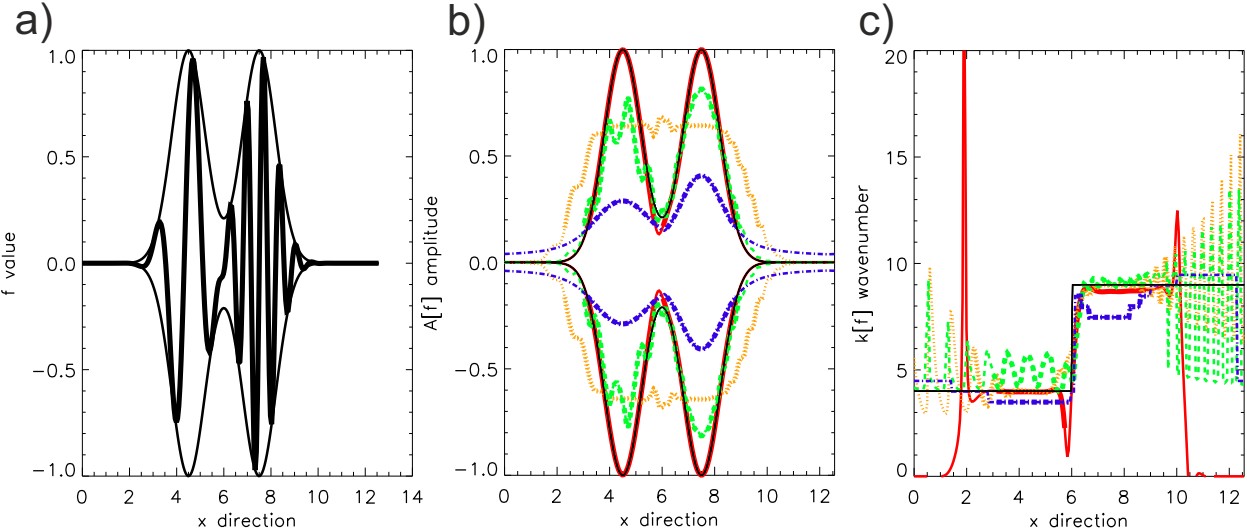

**Figure 1.** One-dimensional test function (a) bold line) adapted from Zimin et al. (2003) and its envelope (a) thin line). Comparison of amplitude (b) and wave number (c) calculated by different methods: UWaDi (solid, red), DIV (dotted, orange), S1D (dashed, green) and 1D ST (dash-dotted, blue). Valid estimates are drawn in bold.

number has to be found. For DIV the box length is set to $L^{DIV} = 8.0$ which covers the largest anticipated wavelength. For 1D ST, which was modified from 3D ST, two steering parameters have to be adapted to the present task. We find a box width factor of $C_L^{1DST} = 0.25$ suiting our requirements. Next to that, 1D ST provides a correction factor for the absolute amplitude value $C_A^{1DST}$. In three-dimensional data analyses enormous amplitude reduction can occur. Artificially changing the amplitude

impacts the variance conservation of the technique. This is why we choose $C_A^{1DST} = 1$ for our example. For S1D a fixed box length has to be determined in advance. We find $L^{S1D} = 1.5$ to give acceptable results. We note that an extension to three dimensions would add information and therefore, accuracy. However, in order to realise comparability we stay with the strictly one-dimensional setup.

The method showing the best agreement with the theoretical value is UWaDi (Fig. 1b). For the amplitude both wave packets are

clearly distinguishable and the maximum peaks are recovered exactly. 1D ST and S1D rebuild the wave packet shape as well. The lack of absolute amplitude value might be adjusted with empirical correction factors provided for 1D ST. Nevertheless, the amplitudes of both wave packets differ from each other. DIV provides a smeared out wave-packet envelope. The wave number calculation is best for UWaDi (Fig. 1c). The high peaks at the beginning and end of the wave packets are sorted out by the quality check. DIV meets the right wave numbers as well, but does not cover the whole spatial range of the two wave

packets. 1D ST and S1D show small deviations from the expected values. Altogether, UWaDi shows the best agreements with the theoretical expectations.

## 2.3 Analysis data

ECMWF data from the IFS operational cycle 41r1 is chosen for this analysis. We performed comparison studies between IFS data on different grid sizes ($0.1°, 0.36°, 1°$). By considering our bandpass filter conditions we found reliable results for the $0.1°$- and $0.36°$-grids. To compromise between computational costs and stability of results we decide that data with a resolution of $0.36°$ (ca. 40 km) meet our requirements. They are retrieved from a resolution of T511. We discuss resolved gravity waves of a horizontal scale between 100 km and 1500 km. In vertical direction we are interested in gravity waves within the wavelength limits of 1 km to 15 km. These scales fulfill the assumption of hydrostatics and cover the range of mid- and low-frequency GWs (Guest et al., 2000).

Vertical propagating GWs are damped in ECMWF IFS products from 10 hPa ($\approx$30 km) upwards (ECMWF, 2016). At 10 hPa the stratospheric sponge starts and a damping of wave propagation is expected (Jablonowski and Williamson, 2011). The mesospheric sponge follows at 1 hPa acting on the divergence and therefore directly on the GW properties. We restrict our analysis to a maximum altitude of 45 km and therefore follow the advice of Yamashita et al. (2010). We interpolate model levels to equidistant height levels between 2 km to 45 km with a distance of 500 m and provide initial scientific analysis for a snapshot on 30 January 2016, 0 UTC corresponding to a minor SSW.

## 2.4 Gravity-wave specific quantities

From the diagnosed fields of amplitude and wave number we calculate the kinematic wave energy $e$ and wave action $A$. In order to find the ageostrophic GW motion we analyse fields of horizontal divergence. The kinematic wave energy is derived from polarisation equations for GWs assuming hydrostatics (Zülicke and Peters, 2006) (Appendix B):

$$e = \frac{\delta^2}{k_h^2}. \tag{14}$$

In this formula we need information on the divergence variance and the horizontal wave number. Both are provided by UWaDi from the three-dimensional divergence field.

$$\delta^2 = \frac{a^2}{2} \tag{15}$$

$$k_h^2 = k_x^2 + k_y^2 \tag{16}$$

The wave action is a conserved quantity as long as the slowly varying wave packet does not interact with the mean flow, e.g. by dissipation, absorption or breaking (Bretherton, 1966). Wave action is defined by putting the kinematic wave energy $e$ in relation to the intrinsic (flow-relative) frequency $\hat{\omega}$:

$$A = \rho \frac{e}{\hat{\omega}}, \tag{17}$$

$\rho$ being the density. The intrinsic frequency $\hat{\omega}$ is calculated with the dispersion relation in mid- and low-frequency approximation: $\hat{\omega}^2 = f^2 + \frac{N^2 \left( k_x^2 + k_y^2 \right)}{k_z^2}$.

From $A = \rho \frac{e}{\hat{\omega}}$ =constant, one can see the following

- density effect: $e \propto \frac{1}{\rho} \propto \exp\left(\frac{z}{H}\right)$. The above derived energy undergoes an exponential increase according to the density with the scale height $H$ in vertical direction $z$.

- wind effect: $e \propto u_h$. This relation holds for a stationary horizontally homogeneous mean flow $(u(z), v(z))$ which implies the invariance of the horizontal wave number ($k_x$ =const and $k_y$ =const) along with the apparent (ground-based) frequency ($\omega = \hat{\omega} + k_x u + k_y v = \hat{\omega} + k_h u_h \cos(\alpha_k - \alpha_u)$ =const). The energy scaling is obtained with the invariance of the wave action for an upwind wave ($\alpha_k \to \alpha_u + \pi$) as $e = \frac{A}{\rho}\hat{\omega} \to \frac{A}{\rho}\omega + k_h u_h$) due to the Doppler shift of the intrinsic frequency (Marks and Eckermann, 1995).

For the following analysis primarily wave action is used.

## 3 Results

A minor SSW occurred on 30 January 2016. Fig. 2a shows the wind speed of the northern hemisphere at 10 hPa at 0 UTC. A vortex displacement from the pole is visible. The displaced vortex causes areas of strongly curved winds. The horizontal divergence as a measure of GWs shows high wave activity above two areas (Fig. 2b). Firstly, above northern Europe horizontal divergence is aligned cross-stream. Secondly, spiral-like patterns appear above eastern Siberia, corresponding to an area of a curved jet streak. UWaDi applied on the field of horizontal divergence provides GW amplitude and wave action (Fig. 2c, d). GW amplitudes show patterns in regions of strongly alternating horizontal divergence. The wave action shows the highest peak above northern Europe and lower values above eastern Siberia. In zonal mean the horizontal wavelength varies between 120 km and 200 km (Fig 3a). In the mid-stratosphere between 18 km and 40 km altitude the horizontal wavelength remains nearly constant. Thus, our assumption of a homogeneous background wind field is approximately valid in the mid-stratosphere. The vertical wavelength scales from 2.2 km to 5.2 km. It increases throughout the whole atmospheric section with a slight change of gradient at the altitude of the tropopause (10 km). The decrease of vertical wavelength above 35 km altitude is dubious. It occurs frequently, also for other temporal snapshots. We suspect an influence of the IFS sponge layer but do not exclude an influence of the decreasing zonal wind, and therefore remain critical on interpretations in these altitudes. The horizontal phase speed in wave direction ($c_h = \frac{\omega}{k_h} = \frac{\hat{\omega}}{k_h} + u_h \cos(\alpha_k - \alpha_u)$) is approximated for an upwind wave ($c_h \to \hat{c}_h - u_h$). It remains unchanged for waves propagating passively through a stationary horizontally homogeneous wind field. The zonal mean remains nearly constant with a value of 5 ms$^{-1}$ below the tropopause and then steadily increases in the stratosphere (Fig. 3b). The indicated variations of horizontal wave number and phase speed contradict assumptions of a passive propagation through a stationary horizontally homogeneous wind field.

We next inspect local profiles in different background wind conditions. Longitude-height sections of zonal wind (Fig. 4a) and wave action (Fig. 4b) at 60° N help to find the location of interesting vertical profiles. Three profiles are chosen that are representative for regions of similar filter conditions. The first profile ① at 7.56° E is chosen to be in a longitudinal range characterised by strong zonal eastward winds and lies in the deceleration area of the jet stream above northern Europe. Profile ② is at 151.92° E, therewith in the area of a strongly curved stratospheric jet associated with the displacement of the polar

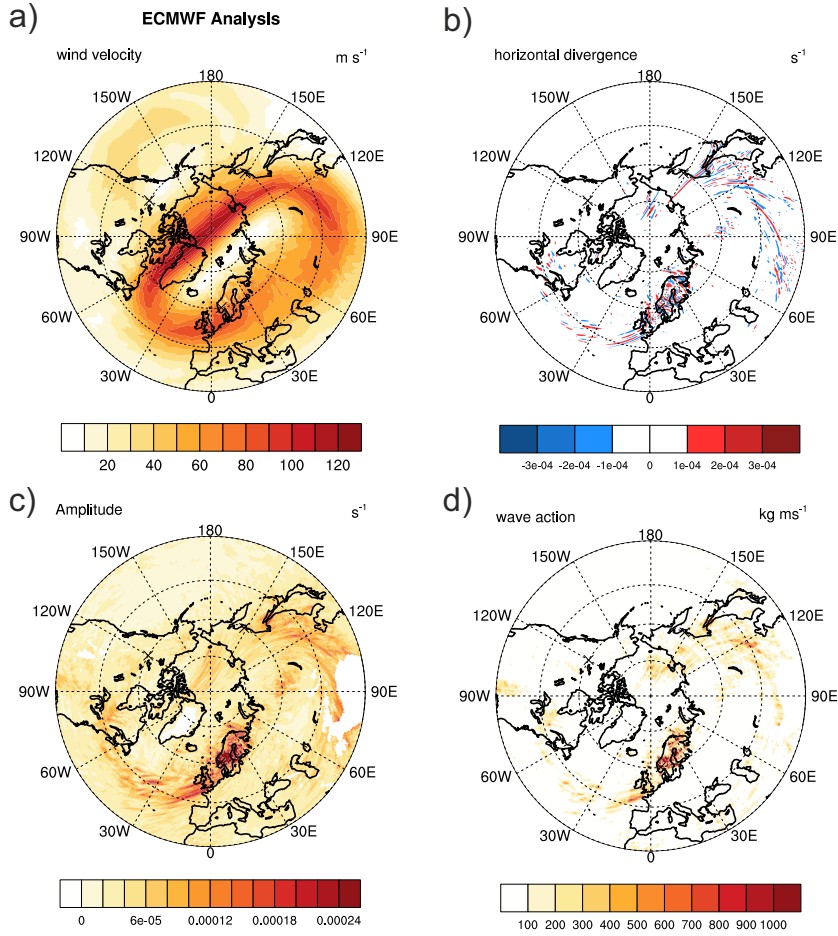

**Figure 2.** Synoptical situation of the northern hemisphere from ECMWF analysis at $10\,\mathrm{hPa}$ on 30 January, 2016. Wind speed (a), horizontal divergence (b). Gravity wave amplitude (c) and wave action (d). Circled numbers along the $60°$ N-latitude indicate positions of three vertical profiles for the later analysis.

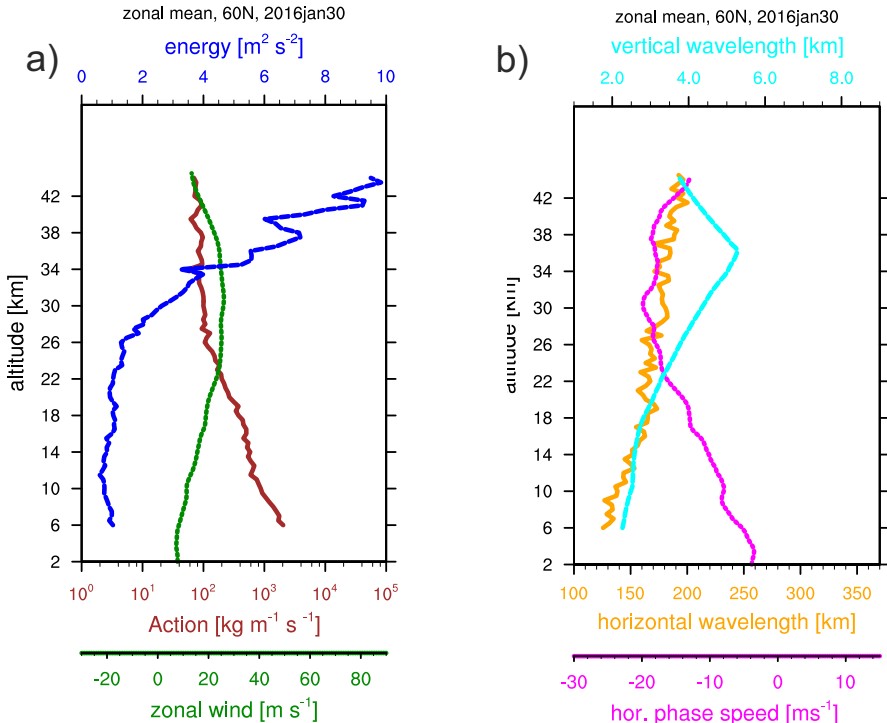

**Figure 3.** Zonal mean profiles at 60 °N with a) zonal wind (green, dotted), energy (dark blue, dashed) and wave action (red, solid) and b) apparent horizontal phase speed (pink, dotted), vertical wavelength (light blue, dashed) and horizontal wavelength (orange, solid).

vortex. In Fig. 4a it is visible as a wind intrusion in the altitude range between 14 km and 34 km. The wave action shows a peak in that height rage (Fig. 4b). For comparison we take a third profile ③ at 240.12° E in a region of low wind speed above Canada, that is: weak tropospheric and weak stratospheric jets.

To highlight the advantage of a local wave analysis we show profiles at selected longitudinal positions (Fig. 5a).

5   During a local increase of wind speed above northern Europe the vertical profiles of ① show that the zonal wind meanders around 50 m s$^{-1}$ (Fig. 5b). In the stratosphere, the vertical wavelength is nearly constant with an average wavelength of 8 km which is higher in the troposphere with 11.5 km (not shown). The wave action shows a high gradient changing from former $10^4$ kg m$^{-1}$ s$^{-1}$ to $10^3$ kg m$^{-1}$ s$^{-1}$ and remains at this high level above 20 km.

Above eastern Siberia a displaced stratospheric jet streak appears, jointly with high wave action (Fig. 4). The zonal wind vertical

10   profile ② shows this in a height range of 14 km to 30 km with in increase from 5 m s$^{-1}$ to maximal 30 m s$^{-1}$ (Fig. 5c). The wave action follows the structure of the zonal wind. Notably, peak GW activity takes place in the lower stratosphere, clearly above the tropospheric jet stream. At these altitude, GWs of vertical wavelength of 2 km is found and the horizontal wavelength is about 350 km (Fig. 6).

The last set of vertical profiles is located in an area of low zonal winds ③ (Fig. 5d). In the troposphere eastward winds and in

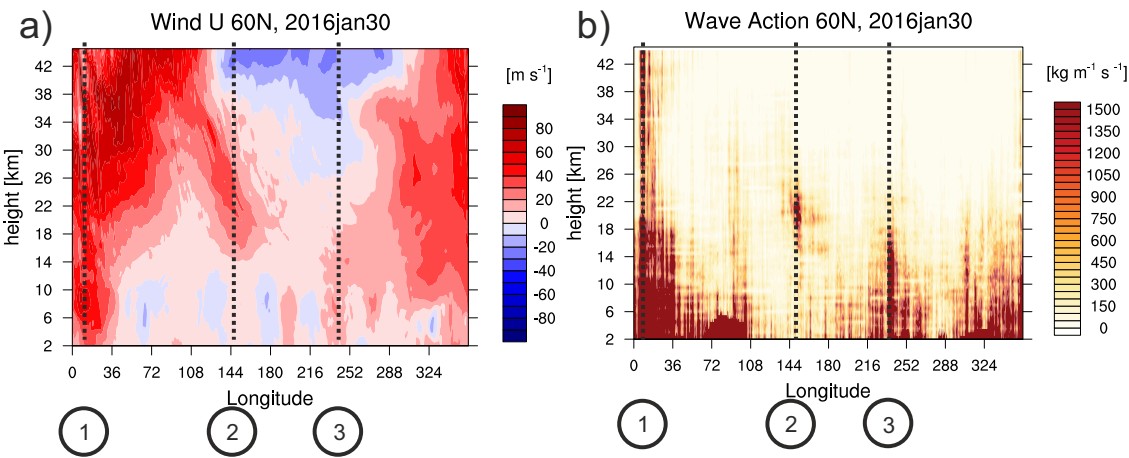

**Figure 4.** Zonal wind (a) and wave action (b) at 60°N, 30 January 2016 in longitude-height section. Numbered vertical profiles for further analysis are highlighted.

the middle stratosphere westward winds occur with magnitudes below 20 $ms^{-1}$. Above the altitude of the wind reversal the wave action remains constant.

## 4 Discussion

The topic of selective wave transmission was first modeled by Dunkerton and Butchart (1984). They highlighted the longitude-dependent gravity wave propagation during an SSW by focussing on the impact on the mesosphere. Ern et al. (2016) further point out that the selective filtering by the anomalous winds during an SSW create heavy impact on GW propagation through the whole atmosphere. They point out theoretically that during the upward propagation of GWs, these waves get attenuated or eliminated by distinct specifications of background flows. Here, we compare local vertical profiles of background wind and GW parameters from analysis data.

Comparing the three cases ①, ② and ③ with respect to their wave action profiles we diagnose at 42 km values ranging over three orders of magnitude. This confirms the high spatial variability of GWs during SSWs. Also the shapes of the profiles differ clearly. One class is characterised with a steady decrease up until 25 km and constance above (such as the zonal mean and cases ① and ③). Another class of profile shows a well-expressed peak in the stratosphere and a steady decrease above (case ②). The detailed analysis related GW dynamics goes beyond the scope of this paper. However, some hypotheses are formulated.

In the high-wind case ①, showing the highest values of wave action and weak in the vertical wave number above northern Europe, we find the longest vertical wavelength of our study (8 km). These steep waves hint on an orographic GW caused by the eastward flow above the Scandinavian mountain ridge Kjølen. This is comparable to findings of Limpasuvan et al. (2011) who showed that during the SSW 2009 westward propagating GW packets emanate from key topographical features around

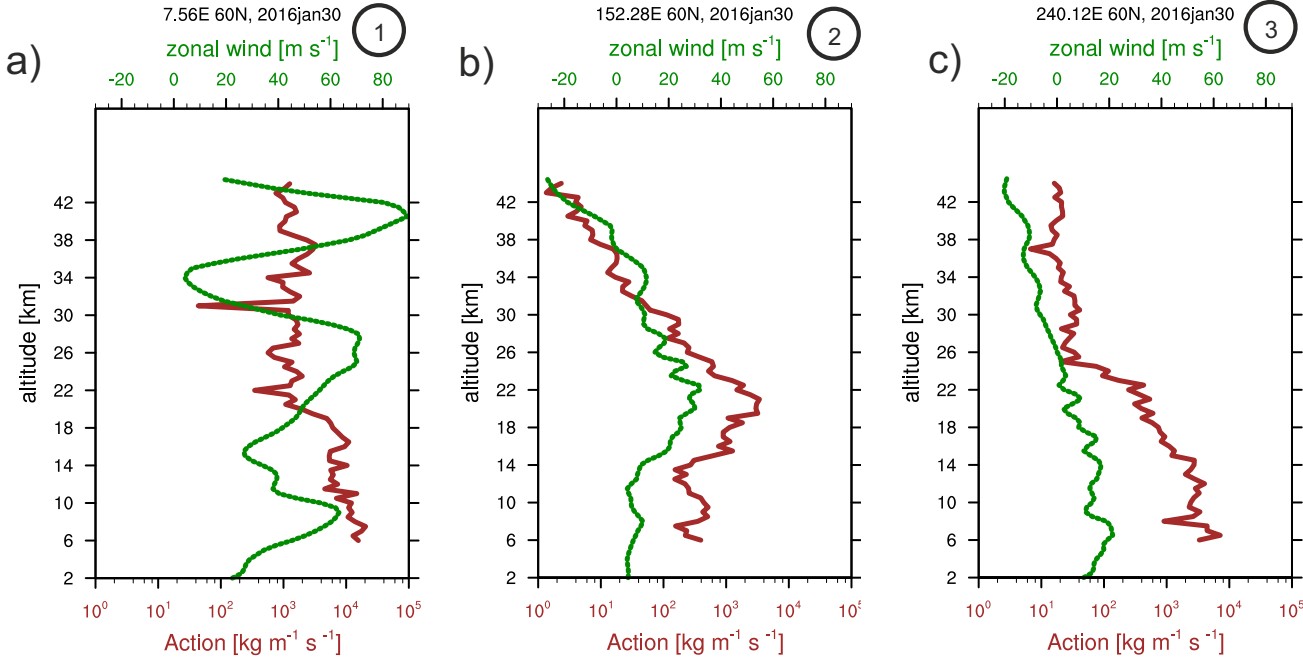

**Figure 5.** Vertical profiles at 60°N, 30 January 2016. Local vertical profiles at 7.56°E (a), 151.92°E (b) and 240.12°E (c) with the wave action (solid, red) and zonal wind (dashed, green). Local profiles according to markers in Fig. 4.

the polar edge and that these wave packets have long vertical wavelengths. Further detailed analysis on this GW packet are expected by upcoming publications according to the joint measurement campaign of ROMIC and MS-GWaves at, amongst others, Kiruna, Sweden (67° N, 20° E).

In the displaced stratospheric jet case ② (Fig. 5c) we find a GW packet triggered off a curved stratospheric jet streak. Firstly
explained by Uccellini and Koch (1987), jet-exit regions in the troposphere are expected to emit GWs. The increase of wave action in the middle stratosphere according to the intrusion of westerlies seen in Fig. 4a and b leads to the assumption that the present feature is associated to the stratospheric jet. The horizontal divergence field supports this hypothesis with GW structures spiraling out of the curved jet above eastern Siberia. These features are comparable to the simulations of the troposphere by Mirzaei et al. (2014) which resolved shallow near-inertia waves in jet-exit regions. Further hints are the higher wave action as
well as the smallest found wavelength (1.9 km) along the largest found horizontal wavelength (350 km).

In the low-wind case ③ the high wave action in the upper troposphere up to the height of the wind reversal of 23 km might be caused by orographic induced GWs due to the position in the lee of the Rocky Mountains. Above that, the overall lowest values of wave action are found, agreeing with measurements in that height range (Thurairajah et al., 2010). It is interesting to note, that the shape of the wave action profile is similar to the zonal-mean profile (compare Fig. 3a and Fig. 5c) while the wind
above 25 km goes into another direction.

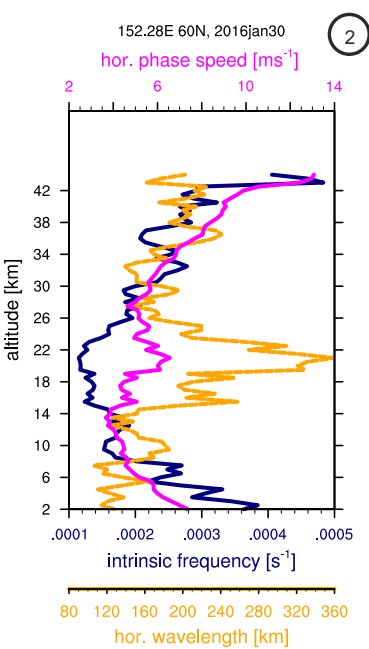

**Figure 6.** Intrinsic frequency (bold, dark blue), horizontal wavelength (dashed, orange) and horizontal phase speed (bold, pink) for profile ② (Fig. 5c).

## 5 Summary and Conclusion

With UWaDi we provide a tool for the analysis of gridded three-dimensional data to estimate amplitude and wave number phase-independently and locally. The method is based on a Hilbert transform with which the use of pre-defined analysis boxes is avoided. It returns an estimate for each data grid point but stays computational cheap. With regard to the locality it clearly

shows its advantages in a method comparison for a synthetic one-dimensional test case. Disadvantages may play a role when the wave spectrum is broad and the attribution of the variance to one dominant harmonic is not justified. The additional estimation of the wave numbers completes the elements of a wave packet description. There is an ambiguity in the sign of the wave number, respectively in the direction of the wave vector, which however is the case for all spatial analysis methods considered in this paper. Still, the method is recommended as a reliable local diagnosis of medium complexity.

For the analysis of gravity waves, we estimated wave energy and wave action from the horizontal divergence. This approach does not require an explicit numerical filtering which is a practical advantage. Other methods for the analysis of unbalanced flow components are available, although more complicated (Mirzaei et al., 2017). While the chosen formulae requires the variance (or squared amplitude) and wave numbers, the Hilbert transform method may also provide local estimates for more complex quantities as included in the combined Rossby wave and gravity wave diagnostics of Kinoshita and Sato (2013).

For our study, which is focused on hydrostatic inertia-GW occurence, the specific approach provides reliable results at low

complexity.

With the demonstrative analysis of the synoptic situation on 30 January 2016 we show the advantages of UWaDi: providing wave quantities on every grid point. Longitude-dependent GW filter processes, known as selective wave transmission, can be diagnosed spatially in detail. Local vertical profiles show selective wave transmissions and generation processes. We found cases with steady decrease of the wave action through the tropopause up to the mid-stratosphere and constant values above in contrast to a case with a strong peak in the lower stratosphere and a steady decrease above. The latter happened in an area where the wind field is effected by the mSSW, characterised with a curved jet stream-exit region in the stratosphere and discuss GW generation by spontaneous emission. The diagnosed long horizontal and short vertical wavelengths support this hypothesis. With the present method we plan to join the closer evaluation of observations and models with respect to local features of GW generation and propagation.

*Code and data availability.* The data from ECMWF is accessible through the archive of www.ecmwf.int provided by the Deutscher Wetterdienst. The code named UWaDi is available through the authors. It is coded in open-source software and a user's manual can be provided.

## Appendix A:  Estimates for two-wave mixture

In this section we illustrate mathematically the amplitude and wave number estimates for a superposition of waves. For simplicity, imagine a mixture of two waves which have amplitudes changing on much larger scales than the lengths of the carrier waves

$$f = a_1 \cos(k_1 x + \phi_1) + a_2 \cos(k_2 x + \phi_2). \tag{A1}$$

The Hilbert transform creates

$$H = a_1 \sin(k_1 x + \phi_1) + a_2 \sin(k_2 x + \phi_2). \tag{A2}$$

The local amplitude is calculated by

$$a^2 = f^2 + H^2. \tag{A3}$$

It contains terms with equal wavelengths (the desired squared sines and cosines) as well as mixed-wavelength terms which are either slow ($\pm(k_1 - k_2)$) or fast ($\pm(k_1 + k_2)$). The application of the low-pass filter (Step 10) is intended to eliminate the fast spurious components which are expected to create the most fuzziness. With this procedure we find from the equal-wave number terms the sum of all squared amplitudes

$$a^2 = a_1^2 + a_2^2. \tag{A4}$$

This means: all variance is included in this estimate. For the wave numbers we find from the definition

$$k^2 = \frac{k_1^2 a_1^2 + k_2^2 a_2^2}{a_1^2 + a_2^2}. \tag{A5}$$

This is the amplitude-weighted sum of squared wave numbers.

The covariance (or squared standard deviation) is the mean of squares:

$$s^2 = \langle f^2 \rangle = \langle a_1^2 \cos^2(k_1 x + \phi_1) + a_2^2 \cos^2(k_2 x + \phi_2) \rangle = \frac{a_1^2}{2} + \frac{a_2^2}{2} = \frac{a^2}{2} \tag{A6}$$

Hence, the ensemble average results in half of the squared amplitude.

## 5 Appendix B: Derivation of total wave energy

The total wave energy is composed of kinetic and potential energy $(e_{tot} = e_{kin} + e_{pot})$. The following considerations are related to linearised equations in Boussinesq approximation in a resting environment (e.g. Gill, 1982). We use the definitions of horizontal divergence $\delta = \mathrm{i}(k_x u + k_y v)$ and vorticity $\zeta = \mathrm{i}(k_x v - k_y u)$ to rewrite the kinetic energy as

$$e_{kin} = \frac{1}{2}(u^2 + v^2) = \frac{1}{2} \frac{\delta^2 + \zeta^2}{k_h^2}. \tag{B1}$$

The potential energy is expressed with the buoyancy tendency $-\mathrm{i}\omega b = -N^2 w$ to yield

$$e_{pot} = \frac{1}{2} \frac{b^2}{N^2} = \frac{1}{2} \frac{N^2 w^2}{\omega^2}. \tag{B2}$$

In order to express the total energy in terms of the divergence, both formulae are combined with the vorticity tendency $-\mathrm{i}\omega\zeta = f\delta$ and the continuity equation $(\delta = -\mathrm{i}k_z w)$

$$e_{tot} = \frac{1}{2}\left( \frac{\delta^2}{k_h^2}\left(1 - \frac{f^2}{\omega^2}\right) - \frac{N^2}{\omega^2} \frac{\delta^2}{k_z^2} \right). \tag{B3}$$

The final result is obtained with incorporation of the dispersion relation for hydrostatic inertia-GWs $\omega^2 = f^2 + N^2 \frac{k_h^2}{k_z^2}$ reading

$$e_{tot} = \frac{\delta^2}{k_h^2}. \tag{B4}$$

*Competing interests.* The authors declare that no competing interests are present.

*Acknowledgements.* We acknowledge the funding of the research unit Multiscale Dynamics of Gravity Waves / project Spontaneous Imbalance by the Deutsche Forschungsgemeinschaft (DFG) through grant ZU 120/2-1. Furthermore the authors want to thank the ECMWF for
data supply. Useful comments on this manuscript were given by Vivien Matthias and Steffen Hien. Special thanks go to one of the reviewers for provision of program code and check values.

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
