# Peer review of "A novel method for the extraction of local gravity wave parameters from gridded three-dimensional data: description, validation and application"

_Atmospheric Chemistry and Physics, 2017_

## Referee Comment (RC1) · Anonymous Referee #1 · 3 Jul 2017

The paper by Lena Schoon and Christoph Zuelicke introduces a new tool for analysing gravity waves (GWs) in regularly gridded data such as from high resolution numerical weather prediction or other similar models able to resolve GWs. The tool has the advantage of being robust and implying only weak constraints on the wavelengths to be analyzed. A stratospheric warming case is discussed as a first application of the tool. The paper is presenting novel work and presents an innovative approach to GWs and thus merits publication in ACP. However, there are some points which are non-conclusive and which I recommend to remove and some details which need to be specified for the analyzed ECMWF case. Also, in my mind, the text could become more lucent and figures should be modified. Therefore I recommend publication after

major revisions.

**Major comments:**

1. Filter and filter response
At page 7, line 10 you introduce that you use a bandpass filter. You state the filter limits in terms of wavelengths. However, most filters have a spectral response rather than a hard limit. For the further interpretation this response is important. In particular, the short horizontal wavelengths cut-off might remove part of the mountain waves and favor waves excited by spontaneous imbalance and the long vertical wavelength cut-off could remove part of the GW spectrum in the high wind case (22 January). The latter would mean that you underestimate GWs for this case. Therefore please include a figure showing the filter response in terms of wavenumber or wavelength. In general, please explain why you need this filter at all.

2. Discuss the advantages and disadvantages of the technique
All techniques to analyze waves need to make a trade off between spectral and spatial resolution. The Hilbert transform is an innovative and elegant concept for high spatial resolution. Since one of the major objects of the paper is to introduce the new technique you should have a paragraph highlighting the properties of the new method. If I understand this correctly, the advantages are:

- The tool is mathematically well defined

- It is applicable to data of any dimension 1D to 4D

- Beside some spectral filter it does not make a preselection of the wavelengths, i.e. it is superior to e.g.

    – Fourier transform, which works on a fixed grid and distributes spectral power from any other wavelengths to that grid

– S3D, which needs to preset the analysis volume and thus either smears out waves with small wavelengths or becomes unreliable at large wavelengths

• With FFT behind, it is fast

The prize you have to pay:

• You can determine only one wave vector per location, i.e. you attribute all the wave energy to a single wave. This does not allow to separate, for instance, the superposition of an upward and a downward propagating wave close to a reflection layer. (maybe that could be the reason for some peaks of wave action below the tropopause)

• With FFT behind some filter issues should apply

3. Introduce the idea

You could make better use of the introductory paragraph of section 2 and motivate the main idea of introducing the Hilbert transform. Perhaps something like: In this section we develop and validate an algorithm to extract wave parameters from equidistant three-dimensional data. For local diagnosis of waves, e.g. inertia gravity waves, phase-independent estimates of wave amplitudes as well as estimates of the wave vector are essential. For this we employ the Hilbert transform. The Hilbert transform shifts any sinusoidal wave structure by a quarter phase, i.e. turning a sine into a cosine. By constructing a new complex number consisting of the original field as real part and its Hilbert transform as the imaginary part, the absolute value is always the amplitude (square-root sum of sine and cosine), independent of the phase, the wavelength of the oscillation and without any explicit fitting of a wave. In addition the phase and, from the phase gradient, the wavenumber are determined. A tool called "Unified Wave Diagnosis" (UWaDi) is developed, which ...

**4. Graphics**

Please use axis scaling which comprise all data. Quite frequently in your figures the curves run out of the selected value range. That is quite unnecessarily hampering the interpretation since often a small extension should suffice.

**5. Selection of individual profiles**

The selection of individual profiles is somewhat arbitrary. With oblique wave propagation and finite vertical group velocity there may be other mechanisms contributing to the vertical structure than you would expect from a single column model. That should be noted in the text. In addition, profiles just in the vicinity seem to be quite different though similar filter arguments would apply. I think it would be more meaningful to select a longitude range of similar filter conditions and show the average profile for that range. Most of your conclusions would still hold and these are the valid ones. For the discussion of these profiles use the actual values (and not as sometimes now average values). For the critical wind filtering discussion you may assume upward propagation and then you should have a horizontal propagation direction and see whether a critical layer is approached.

**6. Remove inconclusive parts**

You compare to radiosonde data and find that they are different. However, there are many reasons why this could be the case and a detailed discussion is beyond the scope of the paper. Similar, there is no reason why wave action should be Gaussian shaped in the altitude profile, so a comparison of peak altitudes is not physically plausible. Please remove these discussions.

**Specific comments:**

P1L1 Why "maintain"? What do you want to say?
Except from a few spectral decomposition methods, the analysis of GWs is based on local methods, and at first reveals local wave phenomena. The calculation of zonal means then is a decision for generating a climatological mean state, but not a question

of the technique. Or are you talking about wave properties? Not all data allow, unlike regular gridded model data, to determine all wave properties.

P1L13 1000km (at the equator zonal wave 40) is more commonly called synoptic scale

P1L23 Complicated sentence

P1L24 "forbidden" is always a matter of the phase speed of the waves. Perhaps: as well as zones where wind reversals inhibit the propagation of GWs.

P1L25 "Models and simulations" That are not two equal terms to be linked by "and"; you need the model to perform a simulation.

P2L14 At altitudes below the sponge. Above about 40km GWs are very strongly damped and not realistic at all

P2L15 Even though the tropical portion of parameterised convective GWs is still too small Not clear what you want to say: ECMWF has a parametrization for convection. This likely results in a misrepresentation of the resolved subtropical/tropical gravity waves. ECMWF does not use a specific parametrization for convective waves, only a non-orographic GW parametrization.

P2L34 Other methods are 3D S-transform (Wright et al., ACP, in press), localized sinusoidal fits (Lehmann et al., AMT, 2012, Preusse et al., ACP, 2014) and 3D wavelets. These are more closely related to your own method and should hence be quoted here. These would be the methods you could delineate your own tool against in a separate paragraph.

P4L1 discrete Fourier transform

P4L4 ... a user-defined ... since you pronounce like "you" and not like "us", i.e. the word as pronounced starts with a consonant

P4L21 As I understand it, d is not the vector of spatial coordinates x,y,z as in the lines before (e.g. a[x,y,z]). Instead it corresponds to the spatial index of e.g. a wavenumber $k_x$ for the x direction, i.e. the sums above are the sums over the three spatial dimensions. Correct? Please use different notations for different things.

P4L24 The noise threshold is essential for understanding the results. How is that calculated? Globally? Locally? Please include the definition.

P4L25 Why is this necessary after you have applied a band-pass filter already above?

P5L4 A one- ...

What happens for two waves of similar size in the same volume?

P5L14 sufficiently monochromatic

P7 Please state precisely which data you are using. Though both Cy41r1 and Cy41r2 use T1279 the effective resolution is different and for Jan 2016 both versions were generated.

P7L4 restricted -> reduced

P7L6 222km / cos(lat) for zonal direction; makes a factor of 2 at 60N and introduces an anisotropy in the cutting frequency

P7L10 These limits are coarse. ECMWF resolves in both relevant model cycles mountain waves with wavelengths shorter than 200km, i.e. you have performed here a preselection in physics.

P7L23 ... but not interacting with the mean flow Is that true? A wave refracted horizontally would conserve its wave action, but change direction and thus transfer momentum to the mean flow.

P7L26 in a mid- and low-frequency approximation:

Say -> From

P7L30 Please use always intrinsic and ground-based, respectively.

[Figure]

P8L1 omit: "one has to accept that"

P8L3 for the following analysis primarily wave action is used.

P8L7 The period 21 January to 21 February 2016 exhibits interesting wind features and is chosen for further analysis.

P8L8 zonal mean?

A change in wave action is supposed to be caused by a variation in the intrinsic frequency hinting at a steepening of GWs You mean relative to energy? Steepening = longer vertical wavelengths

Your analysis in F3 is 2D (in the horizontal plane)? Please highlight this.

P9L1 but not well above the filter!

P9LL1 What is the use of average values. In particular of e.g. average intrinsic phase speeds.

P9L9 Here you do a cross-comparison with four differences: location, time, generic data and analysis method. This is very difficult to interpret. Better keep at least time and space the same.

Figure 4: Please show also plots for wave action from UWADI

P9L24 Where is there any evidence for orographic waves in the figure?

In the stratosphere you can use the rule of thumb: 3km vertical wavelength correspond to 10m/s intrinsic phase speed. With a vertical cut-off of 15km that would mean that at 50m/s background wind speed most slow waves (such as mountain waves) are still in, and at 75m/s background wind speed a considerable part is removed.

How is a vertical wavenumber zero compatible with a long-wavelength filter edge of 15km?

Show the filter response for the respective axes.

Fig 6 Please use the same vertical axis for panels a and b

P11L13 "This finding contributes to our understanding to the density decrease with height which is not considered for the kinematic wave energy." Perhaps instead: The vertical profile results mainly from two competing effects: at increasing altitude density decreases. As the kinematic wave energy does not include density, we expect exponential energy growth for conservative wave propagation and hence a strong increase in regions of weak dissipation. Above 40km the mesospheric sponge of the ECMWF model sets in and cause strong, arteficial dissipation, which results in the decrease of wave energy at larger altitudes. In addition, ...

P11L15 Wave action should decrease above source altitude and there is no reason to assume it to be Gaussian. Please remove the sentence

P13L5 afterwards -> above

P13L6 the slow westward

P13L9 This is mid frequency approximation. If you use further approximations, note in the text

P13LL7 You use a single profile at one fixed time for your argumentation, but wave propagation may be oblique, requires time and the tropopause may cause partial reflection (what happens in the latter case?). Are your conclusions valid the same way for the profile at 40W? It would make much more sense to me to integerate over a small region.

P13L21 GWs are forbidden -> GW propagation is strongly inhibited. Unless $N^2 < 0$ you always have some GWs which may exist

P13L28 A longitudinal profile at $20°$ An altitude profile at $20°$ west ... Where do I see the wavelike structure in the figure?

P13L32 wave guide A wave guide means keeping the wave between two reflection

layers as you would have it e.g. at the tropopause or mesopause for short horizontal wavelength waves. Open-walve region?

P17L4 Split this up: The tool is applicable to ... Here we apply the tool on divergence fields and limit towards long wavelengths thus isolating GWs. The procedure leads to reliable results for synthetic test cases. As a first application we run it on operational analysis data of ECMWF for a stratospheric warming case.

In future, the lack ... For comparing the phases you do not even need to have the Hilbert transform 4D. The most serious limitation is that you need ECMWF data at sufficient dense sampling which you could get from forecast data. For a first step you could assume upward propagation of the wave energy.

P17LL14 You use a pump=source and valve picture. 1.) You should have an introducing sentence that this is a picture for a more complicated process. 2.) That's based on Ron Smith's ideas? Is there any peer-reviewed article to quote? 3.) While the valve summarizes the properties of a wind profile, source is already such a general expression. Is it necessary to introduce a new word? In particular since source could work already in such a hydraulic picture.

---

## Referee Comment (RC2) · Anonymous Referee #2 · 24 Jul 2017

Review of

Diagnosis of Local Gravity Wave Properties during a Sudden Stratospheric Warming

by

Lena Schoon and Christoph Zülicke

This paper diagnoses local gravity wave properties during a series on minor stratospheric sudden warmings (SSWs) which occurred in the Northern hemispheric spring 2016. The authors focus on only a few wave properties as amplitude and wavenumber which are retrieved from gridded 3D horizontal divergence data by applying a well-established methodology to gravity waves. From these quantities, wave action is estimated which is the key quantity of the SSW analysis. Altogether, the paper suffers from too many weaknesses, and I recommend rewriting the paper, maybe even split it in two separate contributions. The main critical points are listed in the following and will be explained in more detail afterwards.

(1) As a first impression, the paper reads as an attempt to combine the presentation of an analysis tool (called "UWaDi") for estimating kinematic gravity wave properties with the discussion of the gravity wave propagation during a prolonged period of minor SSWs. Unfortunately, I've to admit: This attempt totally fails as neither the analysis tool part nor the gravity wave analysis are substantial enough to allow for a combined scientific publication.

(2) The methodology to retrieve gravity wave parameters is not convincingly introduced and clearly outlined for global 3D gridded data. Compared to solid and mathematically exact descriptions, e.g. provided by Zimin et al. (2003), the mathematical part is poor, see comments below. Especially, it is not necessary to repeat that the method is working for synthetic data as this was documented by others already.

It would be much more interesting to see the application of the method to gravity wave packets using 3D IFS analysis fields of horizontal divergence step-by-step. Essential parts are missing in the description: extraction of wave packets (not all regions of non-zero divergence belong to gravity waves) and proof that the extracted wave packets really satisfy the dispersion relationship.

Another point: The horizontal divergence is a quantity which can hardly be observed in the atmosphere. I miss a clear link to observable quantities like temperature fluctuations. There are published attempts, e.g. by Khaykin et al. (2015)[1] to do so. Without such a link, the whole analysis tool is probably handy for gridded data but gives no quantitative relation to observations in the real atmosphere.

(3) The analysis of the minor SSW is totally incomprehensive. It is not clear what the relation between time/space is and which mean values, which locations are considered. There are several hypotheses formulated and statements given in the text which are not proven by results presented in the paper. Is there any progress in knowledge, new
* * *
[1] Khaykin, S. M., Hauchecorne, A., Mzé, N. and Keckhut, P. (2015), Seasonal variation of gravity wave activity at midlatitudes from 7 years of COSMIC GPS and Rayleigh lidar temperature observations. Geophys. Res. Lett., 42: 1251–1258. doi: 10.1002/2014GL062891

understanding compared to the results on selective wave transmission during SSWs published by Dunkerton and Butchart (1984)[2]?

(4) The writing is often very sloppy and not precise. Definitions are modified without discussing the implications, see remarks about wave action. The whole style of the paper is essentially not acceptable for a scientific publication. There is a frequent mix between presenting of results and discussions which blurs the paper and makes reading more than difficult. Below, I give several examples without attempting to edit through the whole text. This would take too much time and effort I cannot spend. I actually stopped reading and commenting after Sec. 3.2. This does not mean, afterwards is all fine. It just means, I see the action by the authors to improve the whole text.

Generally, I noticed a tendency to name, denote facts and processes with new, partly fancy terms (mostly taken from hydromechanics for what reason ever) which are not exactly defined or explained in the text and which leave room for associations. I just want to remind the authors on one principle, scientific publications should follow. It is known as Occam's razor and says "Entities must not be multiplied beyond necessity". It would be great, if the author could follow this principle in future publications. Take as an example the naming of the analysis tool. Why a new name is created for a well-documented methodology which has been obviously used several times before? Well, maybe for other scales and maybe also because an approximated form of wave action is calculated here, but it is absolutely not clear why this minor modification should be named with "Unified Wave Analysis". What does "unified" mean?

The quality and labeling of some of the figures is poor. Examples are given below.

(5) Essential references are missing in the text. The authors focus on the winter 2015/16. They totally ignore papers which are even published from authors of their own institution! Examples are given below.

Last but not least, clear-cut formulated scientific questions are missing for both parts of the paper. So, the suitability of the paper to fit within the scope of ACP cannot be evaluated so far. And maybe, to formulate scientific questions might be a suitable starting point for a new attempt to publish results of the presented study. Thus, at the end, I recommend to proceeds on two routes. First, outline the new facets of the wave analysis clearly and publish these as an independent methodological contribution, e.g. to the GMD. Secondly, conduct a thorough study of the sequence of minor SSWs which occurred in January/February 2016. If the increment of knowledge gain is measurable and constitutes a significant contribution to the understanding, such a paper would fit perfectly to ACP!
* * *
[2] Dunkerton, T.J. and N. Butchart, 1984: Propagation and Selective Transmission of Internal Gravity Waves in a Sudden Warming. *J. Atmos. Sci.*, **41**, 1443–1460, https://doi.org/10.1175/1520-0469(1984)041<1443:PASTOI>2.0.CO;2

Specific Remarks:

**Abstract**

line 1: These two sentences are incomprehensible. What do they mean? Furthermore, Abstract is not a place to argue.

line 2: Reads like a technical task which is the topic of the paper. Formulation and grammar is unclear: What is a "diagnostic tool for studies of wave packets locally"? Do you mean: "retrieve localized wave packets from 3D gridded data"? The following sentence with "UWaDi" confirms the impression of a technical study.

line 4: Be more specific: you use 6 hourly operational analyses of the IFS? Why do you use such a general formulation as " ...is used to perform the analysis"? Write exactly what you do with the data: they are interpolated on a spatially equidistant grid to apply the Hilbert transformation to extract amplitudes and wave numbers at specific times ....

line 5: The first result appears (about the effect of the sponge layer). Is this an essential result of the applied method to be mentioned first? Does it undoubtedly relate to the assumed numerical damping or is there a possibility that the atmospheric state simply didn't supported gravity waves? See remarks to Sec.2.3.

line 7: Second result, however, incomprehensible again. What means " zonal mean wind quantities cannot reveal local 'valves' ...". The usage of not generally accepted terminology or terminology which is not yet introduced in the previous text is dangerous and does not explain anything.  What are "zonal mean wind quantities"?

Line 8: third result: obviously, one event of the mentioned three cases (line 6) is picked randomly which states high gravity wave activity without any relation to location and height. And again a term "local pump" is used which does not explain anything. Why these relations to hydro-machines?

line 9: Why "Accordingly"? What shall the reader re-connect in order to conclude about the advantages which are stated?

At the end: The Abstract is incomprehensive and incomprehensible, and it leaves more questions than answers! It needs a thorough re-write and focus either on methodology or SSW dynamics.

**1 Introduction**

Generally, an Introduction should contain the state-of-the-art knowledge of the topic which is going to be addressed in the paper. It should formulate the challenges and the methods which are applied to answer the scientific questions resulting from the challenges. At the end, the answers are given in the Conclusions where you should clearly state what kind of new knowledge has been generated by the research conducted for the paper. Unfortunately, this Section 1 only partly serves this purpose.

**First paragraph**

*PAGE 1*

line 12: provide evidence by adding essential references

line 12/13: The logic of the sentence goes wrong: Do "the scales of GWs ... create a broad field of interest .."?? I don't think so. Furthermore, do you really claim that atmospheric gravity waves exist at 10 m scale??

line 14/15: What do you mean with "huge changes in GW appearance"? Where? When? Increase? Decrease? Provide evidence by references. Be more specific. For example, mention that you consider the Northern hemisphere only and specify the physical variables you are referring to.

line 15/16: This classification relies on the definition of "normal winter conditions" and "summer-like conditions". Specify what is meant! Which months are you referring to? Early winter, late winter? The use of these terms is an example where the application of the principle of Occam's razor would be beneficial.

Essential references about SSWs are missing, also at lines 18-20. Start with

Butler, A.H., D.J. Seidel, S.C. Hardiman, N. Butchart, T. Birner, and A. Match, 2015: Defining Sudden Stratospheric Warmings. Bull. Amer. Meteor. Soc., 96, 1913–1928, https://doi.org/10.1175/BAMS-D-13-00173.1

and find relevant references therein.

line 17: What are "winder" conditions?

lines 21-23: Very colloquial language! Be specific what the "crucial role in driving …" means

lines 23-25: Be more specific, not so general. Attention by using the term "wave guide": in the cited paper (Dunkerton and Butchart, 1984) this term never appears and, mostly, it refers to horizontal propagation. I think you might refer to the concept of selective wave transmission instead which was introduced by Dunkerton and Butchart (1984). Again: very colloquial language.

line 25: This is a rather general statement. Ask yourself what specific facts, information do we need from the cited papers for introducing your research topic! Just the statement that their data can be analyzed seems to weak!

*PAGE 2*

line 2: Do De Wit et al and the other cited papers really "verify" the momentum fluxes analyzed by the mentioned modeling papers? Be more specific and keep an eye what is needed in your text. As far as I see, momentum flux does not play any role in the paper!

line 4:

- The statements of the Ern et al. (2016) seem to be essential: Describe what is exactly meant with the "zonal average view of GW parameters". Then, get the way to your point of local wave quantities.

- provide evidence of your statements using "mainly extracted" and "misleading"

line 5: the fact that "local GW activity can vary locally" is known and best expressed in the intermittency which was derived from various observations - why such a long chain of arguments before??

line 6: colloquial: "gravity waves slow down" – be more physically exact and refer to vanishing vertical group velocity. Not all gravity waves interact with the critical level, only those whose phase speed is equal to the background wind. Good references are text books on gravity waves as Nappo (2012), Sutherland (2010), Gill (1982), Gossard and Hooke (1975), ..... or the papers of Bretherton (1966, 1969)[3] and Booker and Bretherton (1967)[4].

line 7 and 8: Introduce and explain physically what is meant by the used terms ("valve" and "bottleneck" and "pump") as you are now making the step from background conditions to local flow regimes.

line 8 and 9: statement of the goal of this study, I suppose. Why test case? What is the emphasis of this study? Is it the methodology or the analysis of the minor SSWs? Focus on one or the other. To keep both alive does not work!

Altogether, the **whole first paragraph** contains too many aspects which do not logically lead to a clear goal formulated in terms of scientific questions. Even the last sentence leaves it open what the paper is focusing on. It does not become evident what the scientific problem is nor why it is timely to conduct such an analysis being presented in the paper. There are vague associations that some kind of previous wave analysis is giving results which will be contrasted (improved, complemented??) with the results of this study. But, at the end, the paragraph is not saying this explicitly and remains incomprehensive.

**Second paragraph**

line 10/11: a very general statement that combines too many aspects: Specifiy the data you are going to analyse! What is meant by "local phenomena and their coupling"?  Give evidence for the statement ".. resolve essential parts of GW dynamics .." – in which sense essential?

line 11/12: provide reference, why already? What is meant with "correct GW appearance"??

line 13/14: why the link to the tropics is necessary? Refer specifically to the results of Yamashita et al. if they are relevant for the present study.

lines 14-20: provide evidence for the ".... bigger portion of resolved GWs ....", this is just a statement, are there references? The collected arguments and statements do not convincingly lead to the concluding sentence starting with "Hence, ....". First of all, the requirements were never specified before. Secondly, the term "local valves" is not defined yet.
* * *
[3] Bretherton, F. P. 1966 'The propagation of groups of internal gravity waves in shear flow,' Quart. J. R. Met. Soc., 92, pp. 466–480
Bretherton, F. P. (1969), Momentum transport by gravity waves. Q.J.R. Meteorol. Soc., 95: 213–243. doi:10.1002/qj.49709540402
[4] Booker, J., & Bretherton, F. (1967). The critical layer for internal gravity waves in a shear flow. *Journal of Fluid Mechanics, 27*(3), 513–539. doi:10.1017/S0022112067000515

I'm trying to guess: you claim that the IFS data provide the locations of wave-induced critical levels?? This might be true if one would know of which part of the GW spectrum you are talking about. Essentially, this aspect of resolution dependence should be discussed in detail to provide fair ground for further arguments. The presented arguments are too general. Moreover, there are quite a few case studies of the recent years using high-resolution analyses and forecasts of the IFS to derive local wave parameters, just to name a few:

Zhao, J., et al., 2017: Lidar observations of stratospheric gravity waves from 2011 to 2015 at McMurdo (77.84° S, 166.69° E), Antarctica: Part I. Vertical wavelengths, periods, and frequency and vertical wavenumber spectra. J. Geophys. Res., DOI: 10.1002/2016JD026368

Ehard, B.,et al, 2017: Horizontal propagation of large-amplitude mountain waves in the vicinity of the polar night jet, J. Geophys. Res., Atmos., 122, doi:10.1002/2016JD025621

lines 19-21: It is not convincingly explained why such an analysis is necessary. And what does such an analysis add to the understanding of internal gravity waves? What are the challenges? Why is such an analysis necessary?

Again: also the second paragraph should be much better structured and focused on the needs which lead to the presentation of the presented approach to analyze gravity waves.

**Third paragraph**:

lines 22-line 9(**PAGE 3**):

This paragraph starts with sentences about sources (why not name them as non-orographic sources) and at line 24 it jumps to methods to extract wave properties: I would recommend to separate these both issues.

- what means "varying" in "search for varying GW sources":  different, variable, transient, ...? Regarding the logics in the first sentence: Why is there  "Another issue … because there is some likeliness of ..."? No idea what this means and implies.
- I don't like the formulation " … which may 'pump' them into the middle atmosphere .." Why "pumping"? Why this analogy to hydro-machines? Waves are excited and they propagate in response to the ambient properties (wind, stability) of the medium. Physically, there exists an established terminology: vertical flux of wave energy or wave action (see again: Occam's razor).

line 25: provide evidence by proper references (" .. found in the literature."); the 2nd sentence in this line, and the following one too, remain incomprehensible as nobody knows what are you referring to. Also, the concluding sentence starting with "Hence, .." (line 26) cannot be verified based on the information you provided.

line 27 – 35: Explain why the mentioned methods are relevant for the present study. From reading this part and scanning through the mentioned papers, I've got the impression that essentially all methodology to derive " .. wave amplitudes and wave numbers .." is available. What is the challenge and the need to present another method? I might be misled, but: you as the authors are responsible to make clear what the community is missing in terms of knowledge and/or methodology. And: what are you

going to add with your paper to close this identified gap! This is not obvious from the present text.

*PAGE 3*

lines 1 -9: Again, it would be beneficial if the reader would be provided with more accurate information. For me, it is rather nebulous what is taken from the published methodology and what is missing and will be added here.

lines 10-18:

The two goals are reformulated: (1) a new method is introduced here "to obtain phase-independent wave properties locally"? What specifically is meant? Amplitudes only?  and (2) "local valves" are going to be detected by considering the vertical GW propagation through the varying background conditions during a mSSW (abbreviation not introduced yet).

"valve detection" – explain exactly what you mean.

- Here, you state you use "reanalyses" (line 12) but later I learnt, these are the operational analyses. Consistency in naming required! This also refers to the new terms "prewarming, midwarming, and postwarming" phases (line 17). Are these the same periods as the stages mentioned earlier on page 1, lines 16,17)???

**2 Method and Data**

Line 19-23: In a potential methodological paper, the very short technical description could be expanded by a code description. Otherwise, the hints to "autonomous" processing and plotting and user-defined namelist as elements of the actual code do not make sense here.

Section 2.1:

- about the name "UWaDi", see above
- line 26: give the range of x-values
- lines 25/27: the Hilbert transform does not "provide a new complex series" – the complex values are determined by Eq. (1) by means of the Hilbert transformation
- the mathematical description is poor as the definitions of DFT and F are not given; are these the same formulae as in Zimin etal (2003)? As a matter of fact, the interested reader should be able to code your algorithm solely based on the equations you provide and on references which exactly point to ingredients you used – this is not possible with the provided information.
- are the quantities calculated by Eq. (1) and (4) the same?
- **PAGE 4**, line 9: I don't think "maintain" is the appropriate verb here, the amplitude or magnitude of a complex number is simply defined as written in Eq. (5); I think, the formulation " .. gives an estimate of the local envelope …" is not correct. Shouldn't it be the amplitude of the wave packet?
- line 23: "First" instead of "Fist"
- Generally, the reference to wave packets and the identification of them is missing!!
- What is the physical meaning of the phase (Eq. 6) with respect to the wave groups?
- In Eqs (8) and (9) indices "d" are used. Later, "d" is used as abbreviation for the vector of Cartesian coordinates.

–   The filtering and smoothing, and the quality checks are not explained in a transparent way!

A concluding paragraph about the advantages of the new method would facilitate the understanding and judgment of the presented algorithm.

**Section 2.2:**

To conduct the presented tests was certainly necessary to code the algorithm properly. However, as the results are neither surprising nor new, I would recommend skipping this part. Instead, the application of algorithm to a 1D series of horizontal divergence along a constant latitude circle at some selected altitude (taken from the IFS data) would be a convincing test if the algorithm really retrieves wave packets and leads to a realistic estimate of amplitude and wavenumber.

**Section 2.3:**

*PAGE 6*

line 18: "ca." ???

*PAGE 7*

It appears that the authors only have limited information and knowledge about the physical parametrizations and the additional filtering and damping in numerical weather prediction models, especially, the IFS cycle they have chosen for their analysis. The main part of the damping in the stratosphere is due to the non-orographic wave drag formulation introduced several years ago (Orr et al., 2010)[5]. Terms as "stratospheric sponge" and "mesospheric sponge" do not describe properly what is done in the model integrations. Essential references are missing which describe the older status of filtering and damping (Jablonowski and Williamson, 2010)[6].

As mentioned above, it is simply assumed that the fading of the waves in the upper stratosphere is due to numerical damping alone. However, physical effects and ceasing wind above the polar night jet might be another reason for wave attenuation. Here, wind lidar measurements or the meteor radar winds (see Fig. 2 in Stober et al, 2017) during the SSWs of spring 2016 conducted by colleagues of the home institution of the authors could clarify at least part of the situation during the minor SSWs.

lines 38-42: As far as I know, the pre-processing step of WRF not only interpolates the data on a regular Cartesian grid it also applies some sort of balancing the field to satisfy the WRF equations. There were also scale factors introduced: u and v are multiplied with them to account for the projection used later on. Did this impact the results? Specify
* * *
[5] Orr, A., P. Bechtold, J. Scinocca, M. Ern, and M. Janiskova, 2010: Improved Middle Atmosphere Climate and Forecasts in the ECMWF Model through a Nonorographic Gravity Wave Drag Parameterization. *J. Climate,* **23**, 5905–5926, https://doi.org/10.1175/2010JCLI3490.1

[6] Jablonowski, C. and D. L. Williamson (2011): "The Pros and Cons of Diffusion, Filters and Fixers in Atmospheric General Circulation Models", In: Lauritzen, P. H., C. Jablonowski, M. A. Taylor, R. D. Nair (Eds.), Numerical Techniques for Global Atmospheric Models', Lecture Notes in Computational Science and Engineering, Springer, Vol. 80, 381-493.

exactly which part you have applied to pre-process your data. How was the
horizontal divergence calculated? Did you take the ECMWF values or are
they calculated by means of WRF-pre-processing? Why was band-pass
filtering necessary?

**Section 2.4:**

- Eq. (13): How is s_delta defined? How is Eq (13) derived? Which
  assumption went into the derivation? Unfortunately, also the mentioned
  reference is not very helpful either.
- Can you give a reference to the statement in line 21?
- Eq. (14): I learned that wave action is the mean wave energy
  (E_KIN+E_POT) divided by the intrinsic frequency, for example
  Sutherland (2010) Eq. 3.94 or Gill (1982) Eqs. 8.12.33 and 8.6.1.
  Obviously, Eq. (14) and using "e" as the E_KIN is an approximation.
  Can you comment why you neglect E_POT?
- Line 28-31 and

***PAGE 8***

  Lines 1-3: you should discuss properties of the wave action and how
wave action is changing in s sheared environment!

**3 Results**

**Section 3.1 The stratospheric conditions in winter 2016**

Reading such a headline (I would modify the last part to Arctic winter
2015/16), one would expect that the authors have undertaken a literature
research what has already been published about the winter 2015/2016. And
there are indeed some articles. Just to mention a few:

Matthias, V., A. Dörnbrack, and G. Stober (2016), The extraordinarily
strong and cold polar vortex in the early northern winter 2015/2016,
Geophys. Res. Lett., 43, 12,287–12,294, *doi:*10.1002/2016GL071676.

Manney, G. L. and Lawrence, Z. D.: The major stratospheric final warming
in 2016: dispersal of vortex air and termination of Arctic chemical ozone
loss, Atmos. Chem. Phys., 16, 15371-15396, https://doi.org/10.5194/acp-
16-15371-2016, 2016.

Stober, G., Matthias, V., Jacobi, C., Wilhelm, S., Höffner, J., and Chau,
J. L.: Exceptionally strong summer-like zonal wind reversal in the upper
mesosphere during winter 2015/16, Ann. Geophys., 35, 711-720,
https://doi.org/10.5194/angeo-35-711-2017, 2017.

Dörnbrack, A., S. Gisinger, M.C. Pitts, L.R. Poole, and M. Maturilli,
2017: Multilevel Cloud Structures over Svalbard. *Mon. Wea. Rev.,* **145**,
1149-1159, https://doi.org/10.1175/MWR-D-16-0214.1

All of them deal inter alia with meteorological conditions in the
stratosphere, with planetary wave activity, with SSWs, and, eventually,
with gravity wave activity in the Arctic. So, they are highly relevant
and totally ignored here. As mentioned above, this is not understandable
as two of these publications come from the same institutions as the
authors themselves.

The section 3.1 is not very focused as it mixes the presentation of meteorological results (mean state in terms of U, Z, gravity waves in terms of DIV, and results from the wave analysis) from the Jan/Feb 2016 period with the discussion. So, a strict separation of presenting results and the discussion is highly recommended to enhance the readability of the text. Furthermore, the comparison to so-called long-term observations in Lindenberg and campaigns in Kühlungsborn is not convincing as the link to SSWs is not obvious. The question stated at the end of line 14, **PAGE 9** is either foolish or not necessary as everybody knows that SSWs are large-amplitude PW events deviating the flow from long-term averages.

line 8: Are these zonal mean zonal winds plotted in Fig. 3? Clarify this in the text!

line 9: Specify the exact criteria which are used to determine the dates of the minor SSWs? From Fig. 3, there is only information about U.

line 15: What are you referring to? Which "diagnosed GW properties" do you mean? Do you refer to the mean values presented some lines above?

line 17: The first sentence manifests the dilemma of the approach which is followed in the whole Section 3: The authors assume a (I assume local) relation between zonal wind and gravity wave activity without explicitly considering the conditions for excitation and propagation. They selected special geographical locations (60°N latitude band, some place near Greenland) and consider the conditions there without taking into account the generation of gravity waves at remote places and their horizontal propagation. At the end, this cumulates in the 1D mechanical analog applying "pumps" and "valves" presented in the final Fig. 9.

line 20: there is inconsistency: here and in the Fig. 4 you say: U, Z at 30 km altitude. But how can you plot Z at a fixed altitude? Maybe, the caption is right saying that the plots are at the 10 hPa pressure surface?! Clarify!!

line 21: What "uniformily distributed wind" mean? As the wind consists of a magnitude and direction, a ring vortex can hardly ever have such property.

line 22: How do you define the edge of the polar vortex? Which quantitative measure you are using? There is a huge volume of literature devoted to this topic and I'm not sure what are you referring to.

line 23: A sentence like "They are supposed to .." is ridiculous in a scientific paper! There is no proof, no evidence of "typical orographic features", just a statement. Please, go ahead and show that this statement is true. I guess, it will be another full paper. And most probably, you will be forced to modify or revise your statement.

lines 23-28, also 32-35: the links to published results should be separated into a discussion chapter and not mixed with the presentation of your results here in this Section 3.

Generally: the quantification of wave activity is very sloppy although the authors applied a tool to quantify them. Therefore sentences like those in lines 31 ("In this area increased GW activity can be observed in the horizontal divergence field …") or on **PAGE 10**,line 2 ("The horizontal divergence field shows much more fluctuations .." should be avoided.

line 4: Avoid statement like this in the presentation of results. They belong to the discussion.

**Section 3.2**

**PAGE 10,**line 7: The logic of the sentence is strange: Why is the focus on "vertical wave propagation since… " the horizontal wavenumber is assumed to be constant?

I cannot follow the argument, why a 1D model is sufficient.  You only consider conditions at 60°N! And from them you conclude later on the mechanisms which are involved. I don't think, this pure mechanistic picture is in any way related to processes in the real atmosphere. There, gravity waves are excited over widespread areas due to a number of sources at different levels from the surface to the mesosphere and they contain a broad spectrum of frequencies and wavelengths. The whole section and the following ones are based on this very strong restriction to assume a wave source near the surface and a pure vertical propagation. I think, this type of argumentation and reasoning is a big step backward from the results on selective wave transmission during SSWs published by Dunkerton and Butchart 33 years ago.

**PAGE 11**

line 4: "westerly orientation": first zonal wind are always east-west winds, so the orientation is clear; second, "westerly" is enough to name wind from the west.

line 8: in my understanding "wind reversal" means change of sign in U; so, in Fig. 6c I see no reversal at all; the wind must be zero by definition at the surface. Why do you mention this?

Line 10: the comparison of this statement with well-defined wave packets visible at 10 hPa (~30 km) in Fig. 4a (divergence) south of the considered band at 60°N evidently show the limited conclusiveness of the analysis. The limited stratospheric wave activity is certainly related to the respective positions with respect to the polar night jet. By the way, this finding is known since years, see the publication of Whiteway et al. (1997)[7] and papers citing his work!

On the other hand, such experimental studies could guide you to adapt your analysis strategy to available knowledge.

**PAGE 13**

Last two paragraphs of Section 3.2: Here, again, you pick a arbitrary location (50°W, 60°N) and build a 1D model out of it which leads to the left schematic in Fig. 9. This is not to accept as you assume that waves are exited near the surface. First of all, you should show that this is really the case. Second, what frequencies, wavelengths, phase velocities do they have? Third, even assuming that all works out fine for our reasoning: What is so different, so new in your conclusions and in the schematic from the common knowledge about critical level filtering??
* * *
[7]Whiteway, J. A.,T. J. Duck, D. P. Donovan, J. C. Bird, S. R. Pal, A. I. Carswell, Measurements of gravity wave activity within and around the Arctic stratospheric vortex, Geophys. Res. Lett., 24, 1387–1390, 1997

You mention the link to PW activity. Nothing (!!) is shown this respect which gives evidence that the statement is true. Again: what is the progress to the paper of Dunkerton and Butchart (1984)??

I stop here.

**FIGURES**

**Fig 1:** Units are missing at the axes. The mentioned crosses are not visible. Or are these the elements of the bold lines?

**Fig 2:** Numbers and units are missing at both of the axes in all panels.

**Fig 3:** It is not clear what exactly is plotted. Zonal mean quantities? Specify! Are the graphs really at 30 km altitude? See Remark to Figure 4 in the text above.

**Fig 4**: Remove the irritating "30 km" label from the figures. It would be helpful not to show the horizontal divergence field alone but also the retrieved wave packets from the algorithm. The scaling of the divergence is too detailed; select a lower absolute value (e.g. $2 \cdot 10^{-4}$ $s^{-1}$) for plotting.

---

## Author Comment (AC2) · 4 Oct 2017

**Reply on Review Process of acp-2017-472 Version 1**

L. Schoon and Ch. Zülicke

October 4, 2017

First of all, we like to thank the two anonymous referees for their time expenses to comment on our manuscript acp-2017-472 published in the discussion part of the special issue of Atmospheric Chemistry and Physics "Sources, propagation, dissipation and impact of gravity waves" on 3 July 2017. In the following we first give an overview of the main changes of the manuscript, adressing both referees and the editor (Sec. 1). This is followed by the respondence to the statements of anonymous Referee #2 (Sec. 2).

**1 General Comments of the Authors**

- Regarding the suggestion of Referee #2 to "improve the whole text" the authors decided to rewrite the whole manuscript. Therefore, the attached file including the highlighted changes looked very complex and we omitted it.

- Now, we attempt to guide the reader to the impact of our manuscript by highlighting more intensively its novel characters in the introductionary part. We expanded the literature research massively.

- As Referee #2 had concerns regarding the reliability of our data (preprocessed with the WRF Preprocessing System (WPS)) we thoroughly investigated the analysis data of the European Centre for Medium-Range Weather Forecasts (ECMWF) to find the best fitting data set and resolution of data during the last month. All calculations were redone and restricted to altitudes below $45\,\mathrm{km}$ to avoid the strong sponge layer in ECMWF data starting at $1\,\mathrm{hPa}$, following the suggestion not just

of Referee #1 but also published findings in literature (Sec. 2.3). We avoid horizontal interpolation by keeping the data on the original latitude-longitude grid, adjusting our algorithm accordingly. The discussion on ECMWF data is shortended appreciably in favour of a brief literature review.

- We provide a step-by-step outline of the methods because Referee #2 doubts that the former explanation was sufficient (Sec. 2.1). We also add some calculations in the Appendix.

- Now, the application of the method is clearer arranged and trimmed to the analysis of three profiles from one time step (Sec.3).

- The concerns of Referee #2 regarding our pictoral schemes of hydromechanics, namely "valves and pumps" are taken care of. We erased this literal description of the analysed mechanisms from the manuscript.

We want to highlight again, that this manuscript focuses on the introduction of our novel method called "Unified Wave Diagnostics" (UWaDi). The application on the minor Sudden Stratospheric Warming on 30 January 2016 acts as a demonstrative application to show the advantage of this method. We plan to join the closer analysis of observations and models with respect to local features of GW generation and propagation. The authors highly recommend, that the introduction and the application of UWaDi should not be seperated and published in different journals as we prefer to join the special issue (SI) "Sources, propagation, dissipation and impact of gravity waves". All four issues named in the title of this SI are specifically addressed in the discussion part of our manuscript. Furthermore, we hope by belonging to this SI, that other scientists interested in this topic can find simple access to our method and cooperation.

**2 Comments to the Referee #2**

*(1) As a first impression, the paper reads as an attempt to combine the presentation of an analysis tool (called "UWaDi") for estimating kinematic gravity wave properties with the discussion of the gravity wave propagation during a prolonged period of minor SSWs. Unfortunately, I've to admit: This attempt totally fails as neither the analysis tool part nor the gravity wave analysis are substantial enough to allow for a combined scientific publication.*

We are sorry for this impression. We revised the whole manuscript to better point out the base of our method as well as the result of our application on the minor SSWs on 30 January 2016.

*(2) The methodology to retrieve gravity wave parameters is not convincingly introduced and clearly outlined for global 3D gridded data. Compared to solid and mathematically exact descriptions, e.g. provided by Zimin et al. (2003), the mathematical part is poor, see comments below. Especially, it is not necessary to repeat that the method is working for synthetic data as this was documented by others already.*

*It would be much more interesting to see the application of the method to gravity wave packets using 3D IFS analysis fields of horizontal divergence step-by-step. Essential parts are missing in the description: extraction of wave packets (not all regions of non-zero divergence belong to gravity waves) and proof that the extracted wave packets really satisfy the dispersion relationship.*

*Another point: The horizontal divergence is a quantity which can hardly be observed in the atmosphere. I miss a clear link to observable quantities like temperature fluctuations. There are published attempts, e.g. by Khaykin et al. (2015) 1 to do so. Without such a link, the whole analysis tool is probably handy for gridded data but gives no quantitative relation to observations in the real atmosphere.*

We thank the reviewer to raise this issue. In response, we added a step-by-step outline of the method. The improvements compared to Zimin et al. (2003) are highlighted in the Introduction as well as in the method part (Sec.2-2.2). We see the necessity of showing that the method works for synthetic data because with that we can point out clearly, that not just the envelope of this wave packet is estimated correctly (like Zimin et al. (2003) showed) but also the wave number calculation at every grid point (which is novel work in UWaDi) works well. This only could be done by an example where the wave number is known in advance. Furthermore, this synthetic wave packet works well as test case for the comparison of several methods (Sec. 2.2).

As described above, we prefer the synthetic test case from Zimin et al. (2003) because with that we can truely show the gain of UWaDi. The discussion of wave quantities in Sec. 3 and 4 should make sure, that we deal with GWs that fulfill the dispersion relation.

With these specifications, we used the advantage of availability of the divergence in the analysis data, which directly made accessible the wavy ageostrophic motion without the need of filtering out the geostrophic modes.

As we point out, this method is developped for gridded data and not primarily for observations. Nevertheless, we added in Sec. 2.1, Step 1 that the method works for every variable on gridded data, if numerical or dynamical filters are approved to provide the fluctuations of the background flow. We choose the horizontal divergence to overcome the use of a numerical filter. Several studies, including the named Khaykin et al. (2015) (Plougonven et al., 2003; Zülicke and Peters, 2006; Limpasuvan et al., 2011; Dörnbrack et al., 2012, 2017) use the horizontal divergence as a dynamical indicator for GWs and so do we.

*(3) The analysis of the minor SSW is totally incomprehensive. It is not clear what the relation between time/space is and which mean values, which locations are considered. There are several hypotheses formulated and statements given in the text which are not proven by results presented in the paper. Is there any progress in knowledge, new understanding compared to the results on selective wave transmission during SSWs published by Dunkerton and Butchart (1984) 2 ?*

We are again sorry for this impression. We changed the whole analysis and hope that Referee #2 sees the connection of our results to the discussion, now. The differences to Dunkerton and Butchart (1984) are pointed out in the Introduction and discussion part of the paper. Shortly: Dunkerton and Butchart (1984) investigated parameterised GWs of a different range of wave length than we do. We concentrate on resolved GWs in analysis data and its vertical propagation through the middle atmosphere. This was not done in Dunkerton and Butchart (1984). In particular, we provide such a local analysis on every longitude which is not possible with such an accuracy with other methods. This has been demonstrated in Sec. 2.2 with the Zimin test case.

*(4) The writing is often very sloppy and not precise. Definitions are modified without discussing the implications, see remarks about wave action. The whole style of the paper is essentially not acceptable for a scientific publication. There is a frequent mix between presenting of results and discussions which blurs the paper and makes reading more than difficult. Below, I give several examples without attempting to edit through the whole*

*text. This would take too much time and effort I cannot spend. I actually stopped read-*
*ing and commenting after Sec. 3.2. This does not mean, afterwards is all fine. It just*
*means, I see the action by the authors to improve the whole text.*

*Generally, I noticed a tendency to name, denote facts and processes with new, partly*
*fancy terms (mostly taken from hydromechanics for what reason ever) which are not ex-*
*actly defined or explained in the text and which leave room for associations. I just want*
*to remind the authors on one principle, scientific publications should follow. It is known*
*as Occam's razor and says "Entities must not be multiplied beyond necessity". It would*
*be great, if the author could follow this principle in future publications. Take as an ex-*
*ample the naming of the analysis tool. Why a new name is created for a well-documented*
*methodology which has been obviously used several times before? Well, maybe for other*
*scales and maybe also because an approximated form of wave action is calculated here,*
*but it is absolutely not clear why this minor modification should be named with "Unified*
*Wave Analysis". What does "unified" mean?*

*The quality and labeling of some of the figures is poor. Examples are given below.*

We take care of this remark and rewrote the whole text. Results (Sec. 3) and Discussion
(Sec. 4) are clearly seperated, now. We hope that by reading the whole manuscript, the
Referee will see our effort of answering the questions asked in the Introduction, analysed
and discussed in Sec. 3 and 4 and summed up in Sec. 5. We carefully took care to keep
the golden threat.

We removed the terms "valve and pump" because they seem to take away the attention
from our scientific goals which is to point out the longitude-dependent vertical propaga-
tion of GWs. The name of the tool is not disputable. The unified character comes from
several issues. First, the method is applicable to several different parts of wave types,
e.g. GWs or Rossby Waves. Furthermore, any kind of variable can be analysed, as long
as it contains wave-like structures. By choosing narrow band bandpass limits one can
even analyse different kinds of one wave type. Hence, it can be used for any kind of
gridded data. It is applicable on one-dimensional data and up to four dimensional data.
We obtain phase-independent wave quantities which makes it easy to calculate wave
energy measures locally. Again, our method is based on that one introduced by Zimin
et al. (2003) but comes with an extra wave number estimation in all three dimensions

(which is the major novelty) and combines the three dimensional amplitude and wave number estimates on the same grid as the input data.

The Figures are new. The labeling is taken care of.

*5) Essential references are missing in the text. The authors focus on the winter 2015/16. They totally ignore papers which are even published from authors of their own institution! Examples are given below.*

*Last but not least, clear-cut formulated scientific questions are missing for both parts of the paper. So, the suitability of the paper to fit within the scope of ACP cannot be evaluated so far. And maybe, to formulate scientific questions might be a suitable starting point for a new attempt to publish results of the presented study. Thus, at the end, I recommend to proceeds on two routes. First, outline the new facets of the wave analysis clearly and publish these as an independent methodological contribution, e.g. to the GMD. Secondly, conduct a thorough study of the sequence of minor SSWs which occurred in January/February 2016. If the increment of knowledge gain is measurable and constitutes a significant contribution to the understanding, such a paper would fit perfectly to ACP!*

We extended the list of references to several publications regarding the Winter 2015/16.

As mentioned above, we reformulated the introduction to find scientific questions and tuned the whole text to answer those. Our comment on the seperation of the method and the application into two journals can be found above (Sec. 1).

*Specific Remarks:*
*Abstract*
*line 1: These two sentences are incomprehensible. What do they mean? Furthermore, Abstract is not a place to argue.*
*line 2: Reads like a technical task which is the topic of the paper. Formulation and grammar is unclear: What is a "diagnostic tool for studies of wave packets locally"? Do you mean: "retrieve localized wave packets from 3D gridded data"? The following sentence with "UWaDi" confirms the impression of a technical study.*
We hope the new formulation is clearer.
*line 4: Be more specific: you use 6 hourly operational analyses of the IFS? Why do you*

*use such a general formulation as " ...is used to perform the analysis"? Write exactly what you do with the data: they are interpolated on a spatially equidistant grid to apply the Hilbert transformation to extract amplitudes and wave numbers at specific times ....*
*line 5: The first result appears (about the effect of the sponge layer). Is this an essential result of the applied method to be mentioned first? Does it undoubtedly relate to the assumed numerical damping or is there a possibility that the atmospheric state simply didn't supported gravity waves? See remarks to Sec.2.3.*
*line 7: Second result, however, incomprehensible again. What means "zonal mean wind quantities cannot reveal local 'valves' ...". The usage of not generally accepted terminology or terminology which is not yet introduced in the previous text is dangerous and does not explain anything. What are "zonal mean wind quantities"?*
*Line 8: third result: obviously, one event of the mentioned three cases (line 6) is picked randomly which states high gravity wave activity without any relation to location and height. And again a term "local pump" is used which does not explain anything. Why these relations to hydro-machines?*
*line 9: Why "Accordingly"? What shall the reader re-connect in order to conclude about the advantages which are stated?*
*At the end: The Abstract is incomprehensive and incomprehensible, and it leaves more questions than answers! It needs a thorough re-write and focus either on methodology or SSW dynamics.*

Regarding the last suggestion of the Referee, we rewrote the abstract completely. There, all these comments were taken care of.

*1 Introduction*
*Generally, an Introduction should contain the state-of-the-art knowledge of the topic which is going to be addressed in the paper. It should formulate the challenges and the methods which are applied to answer the scientific questions resulting from the challenges. At the end, the answers are given in the Conclusions where you should clearly state what kind of new knowledge has been generated by the research conducted for the paper. Unfortunately, this Section 1 only partly serves this purpose.*

We extended the Introduction, including more information on other methods and analyses of the winter 2015/16. We resorted it and took care of rising questions in the individual paragraphs and answering them in the corresponding paragraph of the Conclusion.

*First paragraph*
*PAGE 1*
*line 12: provide evidence by adding essential references*

The authors clearly point out, that an overall overview on the different scales of GWs can be found in the given reference (Fritts and Alexander, 2003).

*line 12/13: The logic of the sentence goes wrong: Do "the scales of GWs ... create a broad field of interest .."?? I don't think so. Furthermore, do you really claim that atmospheric gravity waves exist at 10 m scale??*

No, we do not claim that. We reformulated the sentence to make its point clearer.

*line 14/15: What do you mean with "huge changes in GW appearance"? Where? When? Increase? Decrease? Provide evidence by references. Be more specific. For example, mention that you consider the Northern hemisphere only and specify the physical variables you are referring to.*

The new Introduction is clearer.

*line 15/16: This classification relies on the definition of "normal winter conditions" and "summer-like conditions". Specify what is meant! Which months are you referring to? Early winter, late winter? The use of these terms is an example where the application of the principle of Occam's razor would be beneficial.*

We are more specific now.

*Essential references about SSWs are missing, also at lines 18-20. Start with*
*Butler, A.H., D.J. Seidel, S.C. Hardiman, N. Butchart, T. Birner, and A. Match, 2015:*
*Defining Sudden Stratospheric Warmings. Bull. Amer. Meteor. Soc., 96, 1913–1928,*
*https://doi.org/10.1175/BAMS-D-13-00173.1*
*and find relevant references therein.*

We included more references on SSWs, especially those dealing with GWs. See paragraph 6 of the Introduction.

*line 17: What are "winder" conditions?*

This typo is removed.

*lines 21-23: Very colloquial language! Be specific what the "crucial role in driving ..." means*

Rewritten.

*lines 23-25: Be more specific, not so general. Attention by using the term "wave guide": in the cited paper (Dunkerton and Butchart, 1984) this term never appears and, mostly, it refers to horizontal propagation. I think you might refer to the concept of selective wave transmission instead which was introduced by Dunkerton and Butchart (1984). Again: very colloquial language.*

Rewritten

*line 25: This is a rather general statement. Ask yourself what specific facts, information do we need from the cited papers for introducing your research topic! Just the statement that their data can be analyzed seems to weak!*

We extended this.

*PAGE 2*
*line 2: Do De Wit et al and the other cited papers really "verify" the momentum fluxes analyzed by the mentioned modeling papers? Be more specific and keep an eye what is needed in your text. As far as I see, momentum flux does not play any role in the paper!*

We removed the references regarding the momentum flux.

*line 4:*
*- The statements of the Ern et al. (2016) seem to be essential: Describe what is exactly*

*meant with the "zonal average view of GW parameters". Then, get the way to your point of local wave quantities.*

*- provide evidence of your statements using "mainly extracted" and "misleading"*

Now, we point out clearly the advantage of our local method in the Introduction.

*line 5: the fact that "local GW activity can vary locally" is known and best expressed in the intermittency which was derived from various observations - why such a long chain of arguments before??*

This is rewritten.

*line 6: colloquial: "gravity waves slow down" – be more physically exact and refer to vanishing vertical group velocity. Not all gravity waves interact with the critical level, only those whose phase speed is equal to the background wind. Good references are text books on gravity waves as Nappo (2012), Sutherland (2010), Gill (1982), Gossard and Hooke (1975), ..... or the papers of Bretherton (1966, 1969) 3 and Booker and Bretherton (1967) 4 .*

A discussion on critical layer absorption can be found in Sec. 4.

*line 7 and 8: Introduce and explain physically what is meant by the used terms ("valve" and "bottleneck" and "pump") as you are now making the step from background conditions to local flow regimes.*

For above mentioned reasons we removed these terms.

*line 8 and 9: statement of the goal of this study, I suppose. Why test case? What is the emphasis of this study? Is it the methodology or the analysis of the minor SSWs? Focus on one or the other. To keep both alive does not work!*

We clearly state our goals in the Introduction, now. More comments on that can be found in Sec. 1 above.

*Altogether, the whole first paragraph contains too many aspects which do not logically lead to a clear goal formulated in terms of scientific questions. Even the last sentence*

*leaves it open what the paper is focusing on. It does not become evident what the scientific problem is nor why it is timely to conduct such an analysis being presented in the paper. There are vague associations that some kind of previous wave analysis is giving results which will be contrasted (improved, complemented??) with the results of this study. But, at the end, the paragraph is not saying this explicitly and remains incomprehensive.*

We are sure that the Introduction is clearer to the reader, now.

*Second paragraph*
*line 10/11: a very general statement that combines too many aspects: Specifiy the data you are going to analyse! What is meant by "local phenomena and their coupling"? Give evidence for the statement ".. resolve essential parts of GW dynamics .." - in which sense essential?*

A detailed description on the data can be found in Sec. 2.3.

*line 11/12: provide reference, why already? What is meant with "correct GW appearance"??*

The reliability of the data is discussed in the Introduction, as well as in the Sec. 2.3.

*line 13/14: why the link to the tropics is necessary? Refer specifically to the results of Yamashita et al. if they are relevant for the present study.*

We refer to Yamashita et al. (2010) and removed the link to the tropics.

*lines 14-20: provide evidence for the ".... bigger portion of resolved GWs ....", this is just a statement, are there references? The collected arguments and statements do not convincingly lead to the concluding sentence starting with "Hence, ....". First of all, the requirements were never specified before. Secondly, the term "local valves" is not defined yet.*

This is rewritten.

*I'm trying to guess: you claim that the IFS data provide the locations of wave-induced*

*critical levels?? This might be true if one would know of which part of the GW spectrum you are talking about. Essentially, this aspect of resolution dependence should be discussed in detail to provide fair ground for further arguments. The presented arguments are too general. Moreover, there are quite a few case studies of the recent years using high-resolution analyses and forecasts of the IFS to derive local wave parameters, just to name a few:*

*Zhao, J., et al., 2017: Lidar observations of stratospheric gravity waves from 2011 to 2015 at McMurdo (77.84 S, 166.69 E), Antarctica: Part I. Vertical wavelengths, periods, and frequency and vertical wavenumber spectra. J. Geophys. Res., DOI: 10.1002/2016JD026368*
*Ehard, B.,et al, 2017: Horizontal propagation of large-amplitude mountain waves in the vicinity of the polar night jet, J. Geophys. Res., Atmos., 122, doi:10.1002/2016JD025621*

We are aware of these publications and decided to add also several other studies that highlight local GW features. Especially in the last but one paragraph of the Introduction we deal with the ECMWF data. Furthermore, we made several case studies with respect to resolution and filters to find the best fitting data to our analysis (See Sec. 2.3). There the restrictions of the data are discussed, too. In particular, we found the same results using the $100 \, \text{km}$ to $1500 \, \text{km}$ filter for $0.36°$ and $0.1°$ grid size data. We interprete this finding with a GW spectrum which is rapidly decaying for horizontal wavelengths above $200 \, \text{km}$.

Zhao et al. (2017) used ECMWF model data as background wind information to interprete vertical wavelengths from their lidar observations at McMurdo, Antarctica. For their spectral GW analysis the height range of 30 - 50 km was used. With regard to method, we quote some more complicated approaches, while for the application we focus on the SSW. Hence, for the sake of brevity we do not include this paper.

Ehard et al. (2017) concentrate on the GW behaviour above NZ and traced horizontally refracted GW signals in IFS data. However, in order to better focus the introduction to the considered SSW case, we quote this paper without further details in the introduction as an example for horizontal propagation.

*lines 19-21: It is not convincingly explained why such an analysis is necessary. And what does such an analysis add to the understanding of internal gravity waves? What are the challenges? Why is such an analysis necessary?*

We point out the impact of our analysis at the end of the most paragraphs in the introduction, now.

*Again: also the second paragraph should be much better structured and focused on the needs which lead to the presentation of the presented approach to analyze gravity waves*

We did that.

*Third paragraph:*
*lines 22-line 9(PAGE 3):*
*This paragraph starts with sentences about sources (why not name them as non-orographic sources) and at line 24 it jumps to methods to extract wave properties: I would recommend to separate these both issues.*
*- what means "varying" in "search for varying GW sources": different, variable, transient, ...? Regarding the logics in the first sentence: Why is there "Another issue ... because there is some likeliness of ..."? No idea what this means and implies*
*- I don't like the formulation " ... which may 'pump' them into the middle atmosphere .." Why "pumping"? Why this analogy to hydro-machines? Waves are excited and they propagate in response to the ambient properties (wind, stability) of the medium. Physically, there exists an established terminology: vertical flux of wave energy or wave action (see again: Occam's razor).*

This was rewritten, taken care of this comments.

*line 25: provide evidence by proper references (" .. found in the literature."); the 2nd sentence in this line, and the following one too, remain incomprehensible as nobody knows what are you referring to. Also, the concluding sentence starting with "Hence, .." (line 26) cannot be verified based on the information you provided.*

Again, by rewriting we hope to clear up this part.

*line 27 - 35: Explain why the mentioned methods are relevant for the present study. From reading this part and scanning through the mentioned papers, I've got the impression that essentially all methodology to derive " .. wave amplitudes and wave numbers .." is available. What is the challenge and the need to present another method? I might be misled, but: you as the authors are responsible to make clear what the community is*

*missing in terms of knowledge and/or methodology. And: what are you going to add with your paper to close this identified gap! This is not obvious from the present text.*

The novelty of our method was already mentioned above and is pointed out much clearer in the manuscript now.

*PAGE 3*
*lines 1 -9: Again, it would be beneficial if the reader would be provided with more accurate information. For me, it is rather nebulous what is taken from the published methodology and what is missing and will be added here.*

These issues are now included in the Introduction and Methodology section.

*lines 10-18:*
*The two goals are reformulated: (1) a new method is introduced here "to obtain phase-independent wave properties locally"? What specifically is meant? Amplitudes only? and (2) "local valves" are going to be detected by considering the vertical GW propagation through the varying background conditions during a mSSW (abbreviation not introduced yet).*
*"valve detection" – explain exactly what you mean.*
*- Here, you state you use "reanalyses" (line 12) but later I learnt, these are the operational analyses. Consistency in naming required! This also refers to the new terms "prewarming, midwarming, and postwarming" phases (line 17). Are these the same periods as the stages mentioned earlier on page 1, lines 16,17)???*

Most of the issues stated are removed from the text. The data is explained in detail. Abbreviations are introduced correctly.

*2 Method and Data*
*Line 19-23: In a potential methodological paper, the very short technical description could be expanded by a code description. Otherwise, the hints to "autonomous" processing and plotting and user-defined namelist as elements of the actual code do not make sense here.*

We extended the section by a step-by-step explanation of the method.

*Section 2.1:*

*about the name "UWaDi", see above*

Discussed above.

*line 26: give the range of x-values*

Done

*lines 25/27: the Hilbert transform does not "provide a new complex series" – the complex values are determined by Eq. (1) by means of the Hilbert transformation*

Changed.

*the mathematical description is poor as the definitions of DFT and F are not given; are these the same formulae as in Zimin etal (2003)? As a matter of fact, the interested reader should be able to code your algorithm solely based on the equations you provide and on references which exactly point to ingredients you used – this is not possible with the provided information.*

As mentioned above, we provide a step-by-step outline, now. As mentioned in the manuscript, the authors may provide the code to interested readers if this is wanted. Again, the agreement to Zimin et al. (2003) is solely restricted to the mathematical background, namely the Hilbert transform.

*are the quantities calculated by Eq. (1) and (4) the same?*

Yes, they are. However, Eq. (1) shows the idea of the Hilbert transform. Eq. (4) belongs to the stepwise implementation of the Hilbert transform.

*PAGE 4, line 9: I don't think "maintain" is the appropriate verb here, the amplitude or magnitude of a complex number is simply defined as written in Eq. (5); I think, the formulation " .. gives an estimate of the local envelope ..." is not correct. Shouldn't it be the amplitude of the wave packet?*

This is rewritten for better understanding.

*line 23: "First" instead of "Fist"*

Changed

*Generally, the reference to wave packets and the identification of them is missing!!*

We provide the synthetic test case as a simple application of the method. There we indentify wave packets. (Sec.2 to 2.2)

*What is the physical meaning of the phase (Eq. 6) with respect to the wave groups?*

The real and (Hilbert-derived) imaginary part of the function are used to change to an amplitude-phase representation. While the amplitude takes the maximum elongation of oscillations, the phase describes the changes in between. How often the phase is changing, this is proportional to the frequency (in time) or wave number (in space). Respective differentiation brings it up. When the wave group consists of many freuquencies, our estimate returns the amplitude-weighted mean of all (see appendix A).

*In Eqs (8) and (9) indices "d" are used. Later, "d" is used as abbreviation for the vector of Cartesian coordinates.*

This is changed.

*The filtering and smoothing, and the quality checks are not explained in a transparent way!*

We provide this in the step-by-step manual, now.

*A concluding paragraph about the advantages of the new method would facilitate the understanding and judgment of the presented algorithm.*

First advantages are mentioned in the Introduction, now. Further we added a Section where we validate our methods with other methods. This clearly showed the locally

precise estimation of amplitude and wave number.

*Section 2.2:*

*To conduct the presented tests was certainly necessary to code the algorithm properly. However, as the results are neither surprising nor new, I would recommend skipping this part. Instead, the application of algorithm to a 1D series of horizontal divergence along a constant latitude circle at some selected altitude (taken from the IFS data) would be a convincing test if the algorithm really retrieves wave packets and leads to a realistic estimate of amplitude and wavenumber.*

We discussed this above. Only a synthetic test case with a-priori known "truth" can be used to validate a method for itself and to conduct a qualified comparison to other methods.

*Section 2.3:*
*PAGE 6*
*line 18: "ca." ???*

Removed

*PAGE 7*

*It appears that the authors only have limited information and knowledge about the physical parametrizations and the additional filtering and damping in numerical weather prediction models, especially, the IFS cycle they have chosen for their analysis. The main part of the damping in the stratosphere is due to the non-orographic wave drag formulation introduced several years ago (Orr et al., 2010) 5 . Terms as "stratospheric sponge" and "mesospheric sponge" do not describe properly what is done in the model integrations. Essential references are missing which describe the older status of filtering and damping (Jablonowski and Williamson, 2010) 6 .*

The authors took the explanation of the sponge layers from ECMWF (2016). We shortened the discussion on the sponge layer issue massively. We rather focussed on vertical propagation issues of well-resolved GWs in the troposphere and middle stratosphere during a SSW event. Hence, we decided to not add a discussion on GW parameterization and damping but to quote the Jablonowski and Williamson paper in the introduction.

Orr et al. (2010) discuss the improvements in ECMWF data by changing from Rayleigh Friction to the Scinocca Scheme. Nevertheless, a sponge-specific discussion which would support our statements on resolved GWs in the new manuscript is lacking. Therefore, we dit not include this in our list of references.

*As mentioned above, it is simply assumed that the fading of the waves in the upper stratosphere is due to numerical damping alone. However, physical effects and ceasing wind above the polar night jet might be another reason for wave attenuation. Here, wind lidar measurements or the meteor radar winds (see Fig. 2 in Stober et al, 2017) during the SSWs of spring 2016 conducted by colleagues of the home institution of the authors could clarify at least part of the situation during the minor SSWs.*

The issues have been discussed in-house before. Our findings found agreements, in general, incuding the intercomparison of unpublished data material. We restrict our method application to a region without massive damping up to the mid-stratosphere and therefore follow the advice of Referee #1 and e.g. Yamashita et al. (2010).

*lines 38-42: As far as I know, the pre-processing step of WRF not only interpolates the data on a regular Cartesian grid it also applies some sort of balancing the field to satisfy the WRF equations. There were also scale factors introduced: u and v are multiplied with them to account for the projection used later on. Did this impact the results? Specify exactly which part you have applied to pre-process your data. How was the horizontal divergence calculated? Did you take the ECMWF values or are they calculated by means of WRF-pre-processing? Why was band-pass filtering necessary?*

Regarding this concerns, we changed the data preprocessing as described above in Sec. 1. The horizontal divergence is directly taken from ECMWF. Bandpass filter is needed to restrict the analysis on wavelengths that we are intersted in, e.g. intertia gravity waves. Clarification on the sampled GW spectrum can be found in Sec. 2.3 in the new manuscript.

*Section 2.4:*
*- Eq. (13): How is s_delta defined? How is Eq (13) derived? Which assumption went into the derivation? Unfortunately, also the mentioned reference is not very helpful either.*

We made it more clearer, now.

*Can you give a reference to the statement in line 21?*

The relation between amplitude and standard deviation is general for harmonic functions as can be verified with a sine. We added an explanation to Appendix A to show that.

*q. (14): I learned that wave action is the mean wave energy (E_KIN+E_POT) divided by the intrinsic frequency, for example Sutherland (2010) Eq. 3.94 or Gill (1982) Eqs. 8.12.33 and 8.6.1.Obviously, Eq. (14) and using "e" as the E_KIN is an approximation. Can you comment why you neglect E_POT?*

A derivation of our formulae can be found in Appendix B.

*Line 28-31 and PAGE 8 Lines 1-3: you should discuss properties of the wave action and how wave action is changing in s sheared environment!*

With this items we want to point out the difference between wave energy and wave action and why we prefere the wave action. A discussion on wave action, especially in varying background winds is part of the discussion, Sec. 4.

*3 Results*
*Section 3.1 The stratospheric conditions in winter 2016 Reading such a headline (I would modify the last part to Arctic winter 2015/16), one would expect that the authors have undertaken a literature research what has already been published about the winter 2015/2016. And there are indeed some articles. Just to mention a few:*
*Matthias, V., A. Dörnbrack, and G. Stober (2016), The extraordinarily strong and cold polar vortex in the early northern winter 2015/2016, Geophys. Res. Lett., 43, 12,287–12,294, doi:10.1002/2016GL071676.*
*Manney, G. L. and Lawrence, Z. D.: The major stratospheric final warming in 2016: dispersal of vortex air and termination of Arctic chemical ozone loss, Atmos. Chem. Phys., 16, 15371-15396, https://doi.org/10.5194/acp-16-15371-2016, 2016.*
*Stober, G., Matthias, V., Jacobi, C., Wilhelm, S., Höffner, J., and Chau, J. L.: Exceptionally strong summer-like zonal wind reversal in the upper mesosphere during win-*

*ter 2015/16, Ann. Geophys., 35, 711-720, https://doi.org/10.5194/angeo-35-711-2017, 2017.*

*Dörnbrack, A., S. Gisinger, M.C. Pitts, L.R. Poole, and M. Maturilli, 2017: Multilevel Cloud Structures over Svalbard. Mon. Wea. Rev., 145,1149 159, https://doi.org/10.1175/MWR-D-16-0214.1*

*All of them deal inter alia with meteorological conditions in the stratosphere, with planetary wave activity, with SSWs, and, eventually, with gravity wave activity in the Arctic. So, they are highly relevant and totally ignored here. As mentioned above, this is not understandable as two of these publications come from the same institutions as the authors themselves.*

We have included the named publications in our Introduction.

*The section 3.1 is not very focused as it mixes the presentation of meteorological results (mean state in terms of U, Z, gravity waves in terms of DIV, and results from the wave analysis) from the Jan/Feb 2016 period with the discussion. So, a strict separation of presenting results and the discussion is highly recommended to enhance the readability of the text. Furthermore, the comparison to so-called long-term observations in Lindenberg and campaigns in Kühlungsborn is not convincing as the link to SSWs is not obvious. The question stated at the end of line 14, PAGE 9 is either foolish or not necessary as everybody knows that SSWs are large-amplitude PW events deviating the flow from long-term averages.*

We seperated Results and Discussion. The comparison with observations from Lindenberg and Kühlungsborn are removed. It was not our aim to sound foolish, so we removed this part, too.

*line 8: Are these zonal mean zonal winds plotted in Fig. 3? Clarify this in the text!*

This Figure was erased.

*line 9: Specify the exact criteria which are used to determine the dates of the minor SSWs? From Fig. 3, there is only information about U.*

Not relevant any more.

*line 15: What are you referring to? Which "diagnosed GW properties" do you mean? Do you refer to the mean values presented some lines above?*

Not relevant any more.

*line 17: The first sentence manifests the dilemma of the approach which is followed in the whole Section 3: The authors assume a (I assume local) relation between zonal wind and gravity wave activity without explicitly considering the conditions for excitation and propagation. They selected special geographical locations (60N latitude band, some place near Greenland) and consider the conditions there without taking into account the generation of gravity waves at remote places and their horizontal propagation. At the end, this cumulates in the 1D mechanical analog applying "pumps" and "valves" presented in the final Fig. 9.*

In the new manuscript we point out the restrictions on vertical propagation only. We compared our local findings to spatial averages over similiar background wind conditions and found no striking deviations. Therefore, we concentrate on local wave propagation, as it is an advantage of our technique to obtain local wave quantities. Furthermore, we highlight the position of our local GWs.

*line 20: there is inconsistency: here and in the Fig. 4 you say: U, Z at 30 km altitude. But how can you plot Z at a fixed altitude? Maybe, the caption is right saying that the plots are at the 10 hPa pressure surface?! Clarify!!*

The way how we obtain equidistant height levels from the model levels is described in Sec. 2.1, Step 1. As we use new data now, the polarstereographic maps are redone.

*line 21: What "uniformily distributed wind" mean? As the wind consists of a magnitude and direction, a ring vortex can hardly ever have such property.*

This is reformulated.

*line 22: How do you define the edge of the polar vortex? Which quantitative measure you are using? There is a huge volume of literature devoted to this topic and I'm not*

*sure what are you referring to.*

The authors are aware of the difficult definition of the edge of the polar vortex. Clearly, this goes beyond the scope of the paper. What is meant is that the bright reader should be capable of combining the wind field (Fig. 2a) with the polar vortex and then sees from the horizontal divergence (Fig. 2b) that anomalies tend to come up at the places where the ring vortex is sharply deformed.

*line 23: A sentence like "They are supposed to .." is ridiculous in a scientific paper! There is no proof, no evidence of "typical orographic features", just a statement. Please, go ahead and show that this statement is true. I guess, it will be another full paper. And most probably, you will be forced to modify or revise your statement.*

Changed.

*lines 23-28, also 32-35: the links to published results should be separated into a discussion chapter and not mixed with the presentation of your results here in this Section 3.*

Done.

*Generally: the quantification of wave activity is very sloppy although the authors applied a tool to quantify them. Therefore sentences like those in lines 31 ("In this area increased GW activity can be observed in the horizontal divergence field ...") or on PAGE 10,line 2 ("The horizontal divergence field shows much more fluctuations .." should be avoided.*

Done

*line 4: Avoid statement like this in the presentation of results. They belong to the discussion.*

Done.

*Section 3.2*
*PAGE 10,line 7: The logic of the sentence is strange: Why is the focus on "vertical wave propagation since... " the horizontal wavenumber is assumed to be constant?*

Changed.

*I cannot follow the argument, why a 1D model is sufficient. You only consider conditions at 60N! And from them you conclude later on the mechanisms which are involved. I don't think, this pure mechanistic picture is in any way related to processes in the real atmosphere. There, gravity waves are excited over widespread areas due to a number of sources at different levels from the surface to the mesosphere and they contain a broad spectrum of frequencies and wavelengths. The whole section and the following ones are based on this very strong restriction to assume a wave source near the surface and a pure vertical propagation. I think, this type of argumentation and reasoning is a big step backward from the results on selective wave transmission during SSWs published by Dunkerton and Butchart 33 years ago.*

The vertical column modell for GWs is well approved. We are aware that horizontal alignment to strong winds or horizontal propagation play a role but this did not play a leading role by comparing our local profiles to spatial averaged profiles, see discussions above. We now show GWs not only arise from sources from the troposphere, but also from stratospheric jets. We also demonstrate that UWaDi may detect locally very different GW activities in different wind conditions.

*PAGE 11*
*line 4: "westerly orientation": first zonal wind are always east-west winds, so the orientation is clear; second, "westerly" is enough to name wind from the west.*

Taken care of.

*line 8: in my understanding "wind reversal" means change of sign in U; so, in Fig. 6c I see no reversal at all; the wind must be zero by definition at the surface. Why do you mention this?*

This Figure was removed.

*Line 10: the comparison of this statement with well-defined wave packets visible at 10 hPa ( 30 km) in Fig. 4a (divergence) south of the considered band at 60N evidently*

*show the limited conclusiveness of the analysis. The limited stratospheric wave activity is certainly related to the respective positions with respect to the polar night jet. By the way, this finding is known since years, see the publication of Whiteway et al. (1997) 7 and papers citing his work!*

In the new manuscript we study three different locations at one time step, showing and discussing wind and divergence together with GW parameters. Insofar, we take the relative position with respect to the polar vortex into account. In the Introduction and discussion we mention several more recent publications and their restrictions due to the necessity of spatial or temporal averaging (Yamashita et al., 2010, 2013; Limpasuvan et al., 2011; Ern et al., 2016). We do not claim, that we find results heavily differing from Whiteway (1997) but with this publication we want wo point out the advantages of our method, beneath others we provide snapshots of vertical profiles of local GW propagation without the necessity of e.g. temporal averaging, which was done in Whiteway (1997). We can give local GW properties in faster changing background winds. To keep this manuscript clear, we restricted the list of references to the already listed publications above which support the messsage of our manuscript equally.

*On the other hand, such experimental studies could guide you to adapt your analysis strategy to available knowledge.*

*PAGE 13*
*Last two paragraphs of Section 3.2: Here, again, you pick a arbitrary location (50W, 60N) and build a 1D model out of it which leads to the left schematic in Fig. 9. This is not to accept as you assume that waves are exited near the surface. First of all, you should show that this is really the case. Second, what frequencies, wavelengths, phase velocities do they have? Third, even assuming that all works out fine for our reasoning: What is so different, so new in your conclusions and in the schematic from the common knowledge about critical level filtering??*

The issue of critical level filtering is a good case to show the advantages of the method. Only with a precise estimate for any height the critical level can be identified. Other box-like methods smear out the results, as shown in the test case.

*You mention the link to PW activity. Nothing (!!) is shown this respect which gives evidence that the statement is true. Again: what is the progress to the paper of Dunkerton*

*and Butchart (1984)??*
*I stop here.*

We removed the discussion regarding PWs. The improvements regarding Dunkerton and Butchart (1984) are already discussed above.

*FIGURES*
*Fig 1: Units are missing at the axes. The mentioned crosses are not visible. Or are these the elements of the bold lines?*

The figure is redone.

*Fig 2: Numbers and units are missing at both of the axes in all panels.*

The figure is removed.

*Fig 3: It is not clear what exactly is plotted. Zonal mean quantities? Specify! Are the graphs really at 30 km altitude? See Remark to Figure 4 in the text above.*

The figure is removed.

*Fig 4: Remove the irritating "30 km" label from the figures. It would be helpful not to show the horizontal divergence field alone but also the retrieved wave packets from the algorithm. The scaling of the divergence is too detailed; select a lower absolute value (e.g. 2 10 -4 s -1 ) for plotting.*

The figures are changed according to these comments.

**References**

Dörnbrack, A., Gisinger, S., Pitts, M. C., Poole, L. R., and Maturilli, M. (2017). Multilevel cloud structures over svalbard. *Monthly Weather Review*, 145(4):1149–1159.

Dörnbrack, A., Pitts, M. C., Poole, L. R., Orsolini, Y. J., Nishii, K., and Nakamura, H. (2012). The 2009-2010 arctic stratospheric winter-general evolution, mountain waves

and predictability of an operational weather forecast model. *Atmospheric Chemistry and Physics*, 12(8):3659.

Dunkerton, T. J. and Butchart, N. (1984). Propagation and selective transmission of internal gravity waves in a sudden warming. *Journal of the Atmospheric Sciences*, 41(8):1443–1460.

ECMWF (2016). *Part III: Dynamics and Numerical Procedures.* IFS Documentation. ECMWF.

Ehard, B., Kaifler, B., Dörnbrack, A., Preusse, P., Eckermann, S. D., Bramberger, M., Gisinger, S., Kaifler, N., Liley, B., Wagner, J., et al. (2017). Horizontal propagation of large-amplitude mountain waves into the polar night jet. *Journal of Geophysical Research: Atmospheres*, 122(3):1423–1436.

Ern, M., Thai, Q., John, C., Martin, G., James, M., and Michael, J. (2016). Satellite observations of middle atmosphere gravity wave absolute momentum flux and of its vertical gradient during recent stratospheric warmings. *Atmos. Chem. Phys*, 1680:7324.

Fritts, D. C. and Alexander, M. J. (2003). Gravity wave dynamics and effects in the middle atmosphere. *Reviews of Geophysics*, 41(1). 1003.

Khaykin, S., Hauchecorne, A., Mzé, N., and Keckhut, P. (2015). Seasonal variation of gravity wave activity at midlatitudes from 7 years of cosmic gps and rayleigh lidar temperature observations. *Geophysical Research Letters*, 42(4):1251–1258.

Limpasuvan, V., Alexander, M. J., Orsolini, Y. J., Wu, D. L., Xue, M., Richter, J. H., and Yamashita, C. (2011). Mesoscale simulations of gravity waves during the 2008–2009 major stratospheric sudden warming. *Journal of Geophysical Research: Atmospheres*, 116(D17).

Orr, A., Bechtold, P., Scinocca, J., Ern, M., and Janiskova, M. (2010). Improved middle atmosphere climate and forecasts in the ecmwf model through a nonorographic gravity wave drag parameterization. *Journal of Climate*, 23(22):5905–5926.

Plougonven, R., Teitelbaum, H., and Zeitlin, V. (2003). Inertia gravity wave generation by the tropospheric midlatitude jet as given by the fronts and atlantic storm-track experiment radio soundings. *Journal of Geophysical Research: Atmospheres*, 108(D21). 4686.

Yamashita, C., England, S. L., Immel, T. J., and Chang, L. C. (2013). Gravity wave variations during elevated stratopause events using saber observations. *Journal of Geophysical Research: Atmospheres*, 118(11):5287–5303.

Yamashita, C., Liu, H.-L., and Chu, X. (2010). Gravity wave variations during the 2009 stratospheric sudden warming as revealed by ecmwf-t799 and observations. *Geophysical Research Letters*, 37(22). L22806.

Zhao, J., Chu, X., Chen, C., Lu, X., Fong, W., Yu, Z., Michael Jones, R., Roberts, B. R., and Dörnbrack, A. (2017). Lidar observations of stratospheric gravity waves from 2011 to 2015 at mcmurdo (77.84 s, 166.69 e), antarctica: 1. vertical wavelengths, periods, and frequency and vertical wave number spectra. *Journal of Geophysical Research: Atmospheres*, 122(10):5041–5062.

Zimin, A. V., Szunyogh, I., Patil, D., Hunt, B. R., and Ott, E. (2003). Extracting envelopes of Rossby wave packets. *Monthly Weather Review*, 131(5):1011–1017.

Zülicke, C. and Peters, D. (2006). Simulation of inertia-gravity waves in a poleward-breaking Rossby wave. *Journal of the Atmospheric Sciences*, 63(12):3253–3276.

---

## Author Comment (AC3) · 4 Oct 2017

Here we provide the new manuscript as a single file.

Please also note the supplement to this comment:
https://www.atmos-chem-phys-discuss.net/acp-2017-472/acp-2017-472-AC3-supplement.pdf
* * *

---

## Author Response (AR1)

**Reply on Review Process of acp-2017-472 Version 1**

L. Schoon and Ch. Zülicke

October 4, 2017

First of all, we like to thank the two anonymous referees for their time expenses to comment on our manuscript acp-2017-472 published in the discussion part of the special issue of Atmospheric Chemistry and Physics "Sources, propagation, dissipation and impact of gravity waves" on 3 July 2017. In the following we first give an overview of the main changes of the manuscript, adressing both referees and the editor (Sec. 1). After that, we reply in detail on the constructive comments of Referee #1 (Sec. 2), followed by the respondence to the statements of anonymous Referee #2 (Sec. 3).

**1 General Comments of the Authors**

- Regarding the suggestion of Referee #2 to "improve the whole text" the authors decided to rewrite the whole manuscript. Therefore, the attached file including the highlighted changes looked very complex and we omitted it.
- Now, we attempt to guide the reader to the impact of our manuscript by highlighting more intensively its novel characters in the introductionary part. We expanded the literature research massively.
- As Referee #2 had concerns regarding the reliability of our data (preprocessed with the WRF Preprocessing System (WPS)) we thoroughly investigated the analysis data of the European Centre for Medium-Range Weather Forecasts (ECMWF) to find the best fitting data set and resolution of data during the last month. All calculations were redone and restricted to altitudes below 45 km to avoid the strong

sponge layer in ECMWF data starting at 1 hPa, following the suggestion not just of Referee #1 but also published findings in literature (Sec. 2.3). We avoid horizontal interpolation by keeping the data on the original latitude-longitude grid, adjusting our algorithm accordingly. The discussion on ECMWF data is shortended appreciably in favour of a brief literature review.

- We provide a step-by-step outline of the methods because Referee #2 doubts that the former explanation was sufficient (Sec. 2.1). We also add some calculations in the Appendix.
- Now, the application of the method is clearer arranged and trimmed to the analysis of three profiles from one time step (Sec.3).
- The concerns of Referee #2 regarding our pictoral schemes of hydromechanics, namely "valves and pumps" are taken care of. We erased this literal description of the analysed mechanisms from the manuscript.

We want to highlight again, that this manuscript focuses on the introduction of our novel method called "Unified Wave Diagnostics" (UWaDi). The application on the minor Sudden Stratospheric Warming on 30 January 2016 acts as a demonstrative application to show the advantage of this method. We plan to join the closer analysis of observations and models with respect to local features of GW generation and propagation. The authors highly recommend, that the introduction and the application of UWaDi should not be seperated and published in different journals as we prefer to join the special issue (SI) "Sources, propagation, dissipation and impact of gravity waves". All four issues named in the title of this SI are specifically addressed in the discussion part of our manuscript. Furthermore, we hope by belonging to this SI, that other scientists interested in this topic can find simple access to our method and cooperation.

**2 Comments to the Referee #1**

**1. Filter and filter response**

At page 7, line 10 you introduce that you use a bandpass filter. You state the filter limits in terms of wavelengths. However, most filters have a spectral response rather than a hard limit. For the further interpretation this response is important. In particular, the short horizontal wavelengths cut-off might remove part of the mountain waves and favor waves excited by spontaneous imbalance and the long vertical wavelength cut-off could remove part of the GW spectrum in the high wind case (22 January). The latter would mean that you underestimate GWs for this case. Therefore please include a figure showing the filter response in terms of wavenumber or wavelength. In general, please explain why you need this filter at all.

The bandpass filter acts in spectral space, where we sort out waves that are not important for our analysis. Here, we use a rectangular filter with hard limits of  $k_{min}$  and  $k_{max}$ . UWaDi can be run with a gaussian shape bandpass filter, which does not have sharp limits. However, we find best results with the rectangular filter in this case.

We now choose a range of wavelengths between 100 km and 1500 km horizontally and 1 km and 15 km vertically. We find inertia GWs from spontaneous imbalance and flow over orography, as we discuss in Sec. 4 of the new manuscript. Insofar, the filter is wide enough. The bandpass filter is described a bit more in detail in Sec. 2.1, Step 4. We also performed numerous tests with the sensitivity of the results to the filter and resolution of data (grid sizes of 1°, 0.36° and 0.1°). It turned out, the characteristic wavelength mentioned in the manuscript does not depend on grid size and filter width. However, we did not dwell on these details in the manuscript, for the sake of brevity.

**2. Discuss the advantages and disadvantages of the technique**

All techniques to analyze waves need to make a trade off between spectral and spatial resolution. The Hilbert transform is an innovative and elegant concept for high spatial resolution. Since one of the major objects of the paper is to introduce the new technique you should have a paragraph highlighting the properties of the new method. If I understand this correctly, the advantages are:

- The tool is mathematically well defined
- It is applicable to data of any dimension 1D to 4D
- Beside some spectral filter it does not make a preselection of the wavelengths, i.e. it is superior to e.g. Fourier transform, which works on a fixed grid and distributes spectral power from any other wavelengths to that grid, which needs to preset the analysis volume and thus either smears out waves with small wavelengths or becomes unreliable at large wavelengths
- With FFT behind, it is fast The prize you have to pay:

- You can determine only one wave vector per location, i.e. you attribute all the wave energy to a single wave. This does not allow to separate, for instance, the superposition of an upward and a downward propagating wave close to a reflection layer. (maybe that could be the reason for some peaks of wave action below the tropopause)
- With FFT behind some filter issues should apply

According to this comment, we extended the method part in the introductary part as well as the method section itself (Sec. 2 to 2.2) where we discuss the above listed issues.

**3. Introduce the idea**

You could make better use of the introductory paragraph of section 2 and motivate the main idea of introducing the Hilbert transform. Perhaps something like: In this section we develop and validate an algorithm to extract wave parameters from equidistant three-dimensional data. For local diagnosis of waves, e.g. inertia gravity waves, phase-independent estimates of wave amplitudes as well as estimates of the wave vector are essential. For this we employ the Hilbert transform. The Hilbert transform shifts any sinusoidal wave structure by a quarter phase, i.e. turning a sine into a cosine. By constructing a new complex number consisting of the original field as real part and its Hilbert transform as the imaginary part, the absolute value is always the amplitude (square-root sum of sine and cosine), independent of the phase, the wavelength of the oscillation and without any explicit fitting of a wave. In addition the phase and, from the phase gradient, the wavenumber are determined. A tool called "Unified Wave Diagnosis" (UWaDi) is developed, which ..

Exceptionally minor changes, we have made use of this suggestion at the beginning of Sec. 2.

**4. Graphics**

Please use axis scaling which comprise all data. Quite frequently in your figures the curves run out of the selected value range. That is quite unnecessarily hampering the interpretation since often a small extension should suffice.

Because of the different data that we use now, we adapted the scaling of the axis and all corresponding figures are comparable.

**5. Selection of individual profiles**

The selection of individual profiles is somewhat arbitrary. With oblique wave propagation and finite vertical group velocity there may be other mechanisms contributing to the vertical structure than you would expect from a single column model. That should be noted in the text. In addition, profiles just in the vicinity seem to be quite different though similar filter arguments would apply. I think it would be more meaningful to select a longitude range of similar filter conditions and show the average profile for that range. Most of your conclusions would still hold and these are the valid ones. For the discussion of these profiles use the actual values (and not as sometimes now average values). For the critical wind filtering discussion you may assume upward propagation and then you should have a horizontal propagation direction and see whether a critical layer is approached.

We inserted the restriction of a vertical-only columnar propagation analysis in the introduction. Further, we checked if spatial averaging over a longitude range of similar wind filtering conditions affect our vertical profile approach. This was not the case so we want to keep our approach of local profiles to point out that we are able to find reliable wave quantities on every grid point without the necessity of spatial averaging. In detail, instead of the local profiles at  $7.56^{\circ}$  E,  $151.92^{\circ}$  E and  $240.12^{\circ}$  E we spatially averaged over  $340-30^{\circ}$  E,  $125-180^{\circ}$  E and  $190-270^{\circ}$  E and found no change in the overall results compared to our local analysis.

**6. Remove inconclusive parts**

You compare to radiosonde data and find that they are different. However, there are many reasons why this could be the case and a detailed discussion is beyond the scope of the paper. Similar, there is no reason why wave action should be Gaussian shaped in the altitude profile, so a comparison of peak altitudes is not physically plausible. Please remove these discussions.

We removed this parts from the manuscript. Furthermore, we added results from other publications which are more comparable to ours (e.g. Krisch et al. (2017)).

Specific comments: P1L1 Why "maintain"? What do you want to say? Except from a few spectral decomposition methods, the analysis of GWs is based on local methods, and at first reveals local wave phenomena. The calculation of zonal means then is a decision for generating a climatological mean state, but not a question

The abstract was rewritten. We distinguish our methods from other methods now clearer in the Introduction and the method part (Sec. 2-2.2): We want to have phase-independent local wave quantities.

P1L13 1000km (at the equator zonal wave 40) is more commonly called synoptic scale

We removed this.

P1L23 Complicated sentence

Changed

P1L24 "forbidden" is always a matter of the phase speed of the waves. Perhaps: as well as zones where wind reversals inhibit the propagation of GWs.

**Changed**

P1L25 "Models and simulations" That are not two equal terms to be linked by "and"; you need the model to perform a simulation.

It is removed.

P2L14 At altitudes below the sponge. Above about 40km GWs are very strongly damped and not realistic at all

We restrict our analysis to an altitude up to 45 km now. A discussion on the impact of the stratospheric sponge layer is given in Sec. 2.3 and Sec. 4.

P2L15 Even though the tropical portion of parameterised convective GWs is still too small Not clear what you want to say: ECMWF has a parametrization for convection. This likely results in a misrepresentation of the resolved subtropical/tropical gravity waves. ECMWF does not use a specific parametrization for convective waves, only a nonorographic GW parametrization.

This missunderstanding was removed.

P2L34 Other methods are 3D S-transform (Wright et al., ACP, in press), localized sinusoidal fits (Lehmann et al., AMT, 2012, Preusse et al., ACP, 2014) and 3D wavelets. These are more closely related to your own method and should hence be quoted here. These would be the methods you could delineate your own tool against in a separate paragraph.

We followed this suggestion in Sec. 2.2 and included a careful comparison for a test case. It revealed clearly the differences between the methods. We are very grateful to the Reviewer #1 for this particular suggestion.

P4L1 discrete Fourier transform

Changed

P4L4 ... a user-defined ... since you pronounce like "you" and not like "us", i.e. the word as pronounced starts with a consonant

**Changed**

P4L21 As I understand it, d is not the vector of spatial coordinates x, y, z as in the lines before (e.g. a[x, y, z]). Instead it corresponds to the spatial index of e.g. a wavenumber  $k_x$ for the x direction, i.e. the sums above are the sums over the three spatial dimensions. Correct? Please use different notations for different things.

Yes, you are correct. It is changed.

P4L24 The noise threshold is essential for understanding the results. How is that calculated? Globally? Locally? Please include the definition.

Now, the definition of the quality checks can be found in Sec. 2.1, Step 9.

P4L25 Why is this necessary after you have applied a band-pass filter already above?

The necessity of the low-pass filtering is now explained in more detail in Sec. 2.1, Step 8. Furthermore we provide a short explanation in Appendix A on that topic.

P5L4 A one- ...

The typo has been changed.

What happens for two waves of similar size in the same volume?

We now discuss the impact of a two-wave mixture in Appendix A.

P5L14 sufficiently monochromatic

This exact formulatin was rewritten in the new manuscript. We refer on the method sensitivity on spectral properties of the data mainly in the discussion part of the new manuscript. It is an important aspect, so we come back to it in several parts of the manuscript. In the step-by-step outline we mention that all variance is considered independent on the spectral properties. Problems may arise with the calculation of the wave number for wide spectra because for that the amplitude-weighted mean is taken. Special care is taken of this issue in the two-wave mixture calculation in the Appendix. In the Conclusion we give references regarding this issue.

P7 Please state precisely which data you are using. Though both Cy41r1 and Cy41r2 use T1279 the effective resolution is different and for Jan 2016 both versions were generated.

A precise description can be found in Sec. 2.3 now.

 $P7L4 \ restricted \rightarrow reduced$

Not relevant any more.

P7L6 222km / cos(lat) for zonal direction; makes a factor of 2 at 60N and introduces

an anisotropy in the cutting frequency

After a couple of tests with grids of  $1^{\circ}$ ,  $0.36^{\circ}$  and  $0.1^{\circ}$ , now, we use data with a resolution of about 40 km horizontal grid distance  $(0.36^{\circ})$ . With our lower bandpass limit of 100 km we make sure that we find waves that are resolved in the data. In order to acknowledge the latitude-dependence of the longitudinal distance, we first take the meridional sectoin for which, from the lat-lon grid, we calculate the distance in this direction and apply the filter, FFT, etc. Because we operate separately with the three dimensions and respective filtering, we take this anisotropy into account.

P7L10 These limits are coarse. ECMWF resolves in both relevant model cycles mountain waves with wavelengths shorter than 200km, i.e. you have performed here a preselection in physics.

The lower limit is reduced to 100 km horizontally.

P7L23... but not interacting with the mean flow Is that true? A wave refracted horizontally would conserve its wave action, but change direction and thus transfer momentum to the mean flow.

We rewrote this part. The wave action is a conserved quantity describing waves in an inhomogeneous background wind field. It does not change for upward propagating waves, as long as they do not interact with the mean flow.

P7L26 in a mid- and low-frequency approximation:

Inserted

 $Say \rightarrow From$

Changed.

P7L30 Please use always intrinsic and ground-based, respectively.

With first appearance of intrinsic and apparent we added the terms (flow-relative) and

(ground-based) to clear this up.

P8L1 omit: "one has to accept that"

Yes. Done.

P8L3 for the following analysis primarily wave action is used.

Changed

P8L7 The period 21 January to 21 February 2016 exhibits interesting wind features and is chosen for further analysis.

Not relevant any more.

P8L8 zonal mean?

Not relevant any more.

A change in wave action is supposed to be caused by a variation in the intrinsic frequency hinting at a steepening of GWs You mean relative to energy? Steepening = longer vertical wavelengths

The steepening of waves regarding the vertical wave lengths is explained more in detail, now, in Sec.4.

Your analysis in F3 is 2D (in the horizontal plane)? Please highlight this.

Former Fig.3 has been removed.

P9L1 but not well above the filter!

Yes, this does not happen in this new analysis with different data. The largest wavelength, found in the mountain-wave case  $\bigcirc$  is well inside the vertical filter of 1 km to 15 km.

*P9LL1* What is the use of average values. In particular of e.g. average intrinsic phase speeds.

This discussion was removed.

P9L9 Here you do a cross-comparison with four differences: location, time, generic data and analysis method. This is very difficult to interpret. Better keep at least time and space the same.

Mentioned above, this discussion was replaced by a comparison to observations made during a comparable synoptic situation.

Figure 4: Please show also plots for wave action from UWADI

This can be seen in Fig. 2, now.

P9L24 Where is there any evidence for orographic waves in the figure?

This was removed.

In the stratosphere you can use the rule of thumb: 3km vertical wavelength correspond to 10m/s intrinsic phase speed. With a vertical cut-off of 15km that would mean that at 50m/s background wind speed most slow waves (such as mountain waves) are still in, and at 75m/s background wind speed a considerable part is removed.

Yes, we also did similar thumbs for any of our profiles to be sure we do not cut the GWs. Actually, the wind was not as large in the considered cases so we do not run into trouble.

How is a vertical wavenumber zero compatible with a long-wavelength filter edge of 15km?

Sorry, this was a bit loose writing. The algorithm does not return a Zero wave number. Now, we find the smallest wavelength (highest vertical wave number) in the stratospheric jet case (2) with 2 km. This is well in the limites of our filter (1 km to 15 km).

Show the filter response for the respective axes.

We experimented with overplotting the filter response over these already rather detailed plots. Unfortunately, we did not arrive at a satisfactory solution without causing confusion. So, we rather left it out.

**Fig 6 Please use the same vertical axis for panels a and b**

We do provide different profiles with similar axis in the new manuscript.

P11L13 "This finding contributes to our understanding to the density decrease with height which is not considered for the kinematic wave energy." Perhaps instead: The vertical profile results mainly from two competing effects: at increasing altitude density decreases. As the kinematic wave energy does not include density, we expect exponential energy growth for conservative wave propagation and hence a strong increase in regions of weak dissipation. Above 40km the mesospheric sponge of the ECMWF model sets in and cause strong, arteficial dissipation, which results in the decrease of wave energy at larger altitudes. In addition, ...

This was rewritten.

P11L15 Wave action should decrease above source altitude and there is no reason to assume it to be Gaussian. Please remove the sentence

With this little calculation we wanted to show that the energy maximum is always above the corresponding action maximum. Therefore, the Gaussian shape was taken as an arbitrary example for a function with maximum. However, as this calculation achieved more questions than clarity, we removed it.

 $P13L5 \ afterwards$  -> above

Not relevant any more.

P13L6 the slow westward

This was rewritten.

P13L9 This is mid frequency approximation. If you use further approximations, note in the text

This was added.

P13LL7 You use a single profile at one fixed time for your argumentation, but wave propagation may be oblique, requires time and the tropopause may cause partial reflection (what happens in the latter case?). Are your conclusions valid the same way for the profile at 40W? It would make much more sense to me to integerate over a small region.

We discussed the issue of local profiles vs. spatial averaging already above. Because we want to show the advantages of local estimates, we do not average over regions. For a rough interpretation of profiles, the columnar (vertical-only) thinking was helpful. We are aware of the more complicated horizontal and vertical propagation issues and mention this in the text.

P13L21 GWs are forbidden -> GW propagation is strongly inhibited. Unless  $N^2

1Leibniz-Institute of Atmospheric Physics, Schlossstrasse 6, 18225 Kühlungsborn, Germany *Correspondence to:* Lena Schoon (schoon@iap-kborn.de)

Abstract. Commonly, wave quantities are maintained in zonal mean averages. Hence, local wave phenomena remain unclear. Here, we introduce a diagnostic tool for studies of wave packets locally. The "Unified Wave Diagnosis" (UWaDi) uses the Hilbert Transform to obtain a complex signal from a real-valued function and estimates the amplitude and wave number locally. Operational data from the European Centre for Medium-Range Weather Forecasts is used to perform the analysis.

- 5 Restrictions on gravity wave propagation due to model sponge layers are identified well above the 10 hPa altitude. From a minor stratospheric warming in January 2016 three cases for vertical gravity wave propagation in different background wind conditions are selected. It is shown that zonal mean wind quantities cannot reveal local "valves" allowing gravity waves to propagate into the mid-stratosphere. The unexpected finding of high gravity wave activity at the minor warming of 30 January 2016 is related to strong planetary wave activity and a strong local "pump". Accordingly, the advantages of a local wave packet
- 10 analysis are demonstrated for profiles up to the model sponge layer. The selective transmission of gravity waves through an inhomogenous mean flow is investigated. For the local diagnosis of wave properties we develop, validate and apply a novel method which is based on the Hilbert transform and is named "Unified Wave Diagnostics" (UWaDi). Thus, it provides wave properties at any grid point for any wave-containing data. UWaDi is validated for a synthetic test case comprising two different wave packets. In comparison with other methods, the perfomance of UWaDi is very good with respect to wave
- 15 properties and their location. For a practical application of UWaDi, a minor sudden stratospheric warming on 30 January 2016 is chosen. Specifying the diagnostics on hydrostatic gravity waves in analyses from the European Centre for Medium-Range Weather Forecasts, we confirm locally different transmission through the middle atmosphere. These are interpreted in terms of columnar vertical propagation using the additionally diagnosed local wave numbers. We also note some hint on local gravity wave generation by the stratospheric jet.

**20 1 Introduction**

25

Gravity waves (GWs) have been subject of intense research during the past decades. The scales of GWs reaching from planetary scales ( $\approx 1000 \text{ km}$ ) to turbulent microscales ( $\approx 10\text{m}$ ) create a broad field of interest in their role in coupling processes of atmospheric dynamics (Fritts and Alexander, 2003). A phenomenon associated with huge changes in GW appearance is a sudden stratospheric warming (SSW). During a SSW event the middle atmosphere is characterised by three different background wind conditions during a short period of time. Starting with normal winter conditions, followed up by summer-like

conditions and a slowly transfer back to winder conditions. Respectively, GW variability is associated with changing background wind conditions. Defined by the World Meteorological Organization SSWs are characterised by a reversal of the 60° N to 90° N-temperature gradient. Major warmings are, additionally, associated with a wind reversal at 10 hPa and 60° N, minor SSWs (mSSWs) with a wind deceleration at 10 hPa and 60° N, where the prevailing westerlies are not turned into easterlies. Planetary

- 5 waves (PWs) play a crucial role in the driving of a SSW event (Andrews et al., 1987). Nevertheless, on the one hand GWs are affected by different background wind conditions that are characteristic during SSWs, on the other hand they are suspected to take, even though a minor, part in the modification of the polar vortex prior to SSWs (Albers and Birner, 2014). Variations in background wind conditions establish zones with wave guides where GWs can propagate easily to higher altitude, as well as forbidden zones of GW propagation (Dunkerton and Butchart, 1984). Models and simulations give the opportunity to analyse
- 10 the behaviour of GWs and PWs during SSWs up to the mesosphere and lower thermosphere (Liu and Roble, 2002; Limpasuvan et al., 2011, 2012; Hitchcock and Sheperd, 2013; Miller et al., 2013; Albers and Birner, 2014). These studies dealing with GW momentum fluxes are verified by radar measurements (De Wit et al., 2015) as well as satellite observations (Limpasuvan et al., 2011; Yamashita et al., 2013; Thurairajah et al., 2014; Jia et al., 2015; Ern et al., 2016). Ern et al., 2016 point out that the zonal average view of GW parameters that are mainly extracted of models is misleading and local GW activity can vary strongly,
- 15 especially during SSWs. Next to critical levels, where GWs slow down and eventually dissipate, local regimes of "valves" or "bottlenecks" occur, where the transmission of GWs into the middle stratosphere depends very sensitively on the local wind (Zülicke and Peters, 2008; Kruse et al., 2016). Here we study a minor warming in January 2016 as a test case for local specific propagation conditions for GWs. 
[revised manuscript text omitted]
 GWs from the tropics are not realistic (Preusse et al., 2014), but mid- and high-latitude GWs are captured well by ECMWF analysis (Yamashita et al., 2010). The recently introduced and improved T1279 resolution yields to a bigger portion of resolved GWs in ECWMF data. Even though the tropical portion of parameterised convective GWs is still too small, mid-latitude GWs are captured well being driven by orographic and jet-stream associated sources (Shutts and Vosper, 2011). A comparison with balloon-borne measurements on the southern hemisphere shows an underestimation of momentum fluxes by
- 35 the factor of 5 of ECMWF GWs in mid-latitudes but the overall appearance and propagation of GWs are realistic (Jewtoukoff et

al., 2015). Hence, ECMWF resolved GWs meet our requirements to identify local valves. In particular, we want to investigate local quantities of resolved GWs in the ECMWF analysis for different background wind conditions. In order to perform such an analysis, the GW amplitudes and wave numbers need to be locally diagnosed in three dimensions.

Another issue is the search for varying GW sources because there is some likeliness that strong PWs govern jet streaks.

- 5 From their exit regions GWs may be radiated by spontaneous imbalance which may "pump" them into the middle atmosphere (Uccelini and Koch, 1987; O'Sullivan and Dunkerton, 1995; Pluogonven et al., 2003). Several methods to extract wave properties can be found in literature. Most of these methods are linked to a limited range of wave frequencies and/or special observation techniques. Hence, most of these methods provide wave properties by spatial averaging and therefore accept a loss of information. Starting with the analysis of PWs, two-dimensional zonal-mean effects can be described by the Eliassen-Palm
- 10 flux (Andrews et al., 1987). Extended studies on three-dimensional wave propagation yield to the wave activity flux (WAF) (Plumb, 1985, 1986) with an analogue for GWs, the gravity wave flux (Bretherton, 1966). The unification of these fluxes lead to a three-dimensional WAF describing wave propagation both for inertia GWs and PWs (Kinoshita and Sato, 2013a). Further studies provide corrections for more accurate magnitudes and directions of wave propagation (Kinoshita and Sato, 2013b) as well as a separate formulation for equatorial waves (Kinoshita and Sato, 2014). While these approaches are designed
- 15 to estimate the pseudo-momentum flux, a more kinetically oriented approach asks for wave amplitudes and wave numbers. A common approach to obtain vertical wave numbers and GW frequency of high-passed filtered data are Stokes parameters (Vincent and Fritts, 1987). This method works for single vertical measurements and provides the wave properties in several vertical height sections. Another method that is capable to derive wave numbers in all three dimensions uses auto-covariance functions defined over spatial-averaged boxes (Zülicke and Peters, 2006). Therein, it is utilised that the use of the horizontal
- 20 divergence simplifies the harmonic analysis by neglecting geostrophic flow by definition and redundantises filtering processes. Local wave properties are obtained by an one-dimensional signal-processing technique named the Hilbert Transform. Zimin et al. (2003) introduce the method by providing the amplitude of a wave packet for an arbitrary set of waves. Rewriting the formulation to work on streamlines makes it more adaptable to various quasi one-dimensional propagations of wave packets (Zimin etz al., 2006). An adaption to Rossby wave trains shows good results and recommends this approach (Glatt and Wirth,
- 25 2014). Furthermore, it is generalised for different directions addressing multi-dimensional problems by Kinoshita and Sato (2013).

In particular, UWaDi requires regular gridded data. Assimilated data products from ECMWF are suitable to analyse local phenomena and their coupling as they resolve essential parts of GW dynamics in the stratosphere. Even the T799 resolution gives proof of correct GW appearance in the stratosphere. Validation with satellite measurements point out that ECMWF

- 30 captures GWs well in the mid- and high-latitudes (Yamashita et al., 2010; Preusse et al., 2014). The improved T1279 resolution yields to a bigger portion of resolved GWs in ECMWF data. Validation studies with measurements show that mid-latitude GWs are captured well being driven by orographic and jet-stream associated sources (Shutts and Vosper, 2011; Jewtoukoff et al., 2015). Our approach concentrates on fields of horizontal divergence of ECMWF IFS data. The horizontal divergence counts for a dynamical indicator for GWs (Plougonven et al., 2003; Zülicke and Peters, 2006). Its magnitude was found to correlate
- 35 with temperature anomalies induced by mountain waves (Dörnbrack et al., 2012; Khaykin et al., 2015). We concentrate on

vertical propagation only, highlighting selective transmission. Studies arguing the restrictions on vertical-only propagation can be found in Yamashita et al. (2013), Kalisch et al. (2014) and Ehard et al. (2017). We point out that meridional propagation of GWs can play an important role for the analysis of the deposition of GW drag in the mesosphere. As we give an idea of GW propagation in the upper troposphere and stratosphere we concentrate on vertical propagation and are aware of the possibility

5 of GW entrainment of strong winds.

With this paper we introduce a new method to obtain phase-independent wave properties locally. The vertical propagation of GWs during different background wind conditions induced by a mSSW serves as an example to detect local valves in vertical GW transmission. ECMWF reanalysis data are chosen to meet our requirements. The paper is organised as follows. After

- 10 providing an overview of the newly developed method to obtain wave properties locally in Sect. 2, two synthetic examples of wave packets are given to validate and demonstrate the advantages of the new algorithm. This section is followed by a description of the used ECMWF data and the wave quantities needed for the analysis of GW properties during a SSW. In Sect. 3 three time phases around a mSSWs in January 2016 are investigated for valves of vertical wave transmissions: the prewarming, midwarming and postwarming phase. A summary, discussion and outlook are found in Sect. 4. The paper is
- 15 organised as follows. After providing a step-by-step introduction and validation of the novel method in Section 2, we give a short overview of the estimation of wave quantities for synthetic data and describe the analysis data. In Section 3 we show our results for the mSSW on 30 January 2016 where we study local GW generation and propagation. The discussion of our results in Section 4 is followed by the Summary and Conclusion (Sec. 5).

**2 Method and data**

- 20 Hereafter, we develop and validate an algorithm to extract wave parameters from an equidistant three-dimensional data. It reads, processes and plots autonomously, accepting user-defined flags in an attached namelist. For local diagnosis of waves, e.g. inertia gravity waves, phase-independent estimates are essential. Therefore, complex quantities are constructed for an amplitude-phase presentation by a Hilbert Transform (Von Storch and Zwiers, 2001; Zimin et al., 2003, Sato et al., 2013). Additionally, wave numbers are estimated. We call this procedure "Unified Wave Diagnosis" (UWaDi). The method is sketched
- 25 briefly. In this section we develop and validate an algorithm to extract wave parameters from three-dimensional data. For local diagnosis of waves, phase-independent estimates of wave amplitudes as well as the wave vector are essential. For this, we employ the Hilbert transform (Von Storch and Zwiers, 2001). The Hilbert transform shifts any sinusoidal wave structure by a quarter phase, i.e. turning a sine into a cosine. By constructing a new complex number consisting of the original field as real part and its Hilbert transform as the imaginary part, the absolute value is always the amplitude (square root of squared real and
- 30 imaginary part). The amplitude is independent of the phase and the wavelength of the oscillation and there is no need of any explicit fitting of a wave. In addition, the absolute wave number in all three dimensions is determined from the phase gradient.

**Unified Wave Diagnosis**

**2.1 Step-by-step outline of the method**

Complex values of a function f[x] are found with the Hilbert Transform in x-direction:

 $\hat{f}[x] = f[x] + iH(f[x]).$

5 The Hilbert Transform provides a new complex series  $\hat{f}[x]$  compounded of the sum of the original function f[x] as the real part and the Hilbert-transformed series H(f[x]) as the imaginary part. Literally, the Hilbert Transform shifts f[x] a quarter to the right  $\left(-\frac{\pi}{2}\right)$  turning a cosine into a sine and a sine into a minus cosine.

The Hilbert Transform itself is composed of three steps (Zimin et al., 2003). First, a discrete Fourier Transform (DFT) is conducted

$$[k] = \mathrm{DFT}(f[x])$$

followed by an user-defined bandpass filtering process  $(0 < k_{min} < k_{max})$

 $f_{filtered}[k] = F(k_{min}, k_{max})f[k],$

and an inverse DFT

15
$$\hat{f}[x] = 2 * \mathrm{DFT}^{-1}(f_{filtered}[k]).$$

Using this complex series, we estimate the amplitude and wave number of a wave packet. The amplitude is maintained by (Schönwiese, 2013)

 $a[x] = |\hat{f}[x]| = \sqrt{f[x]^2 + H(f[x])^2}$

and gives an estimate of the local envelope of an oscillating function.

20 The phase  $\Phi$

$$\Phi[x] = \operatorname{atan}\left(\frac{H(f[x])}{f[x]}\right)$$

. . . .

is used to derive the absolute wave number

$$k_x[x] = \frac{d\Phi[x]}{dx}.$$

By highlighting the use of UWaDi for three-dimensional data the wave number-weighted three-dimensional quantities are the

25 main gain

$$a_{final}[x,y,z] = \sqrt{\frac{\sum_{d} q_{d} k_{d}^{2} a_{d}^{2}}{\sum_{d} q_{d} k_{d}^{2}}}$$

and

$$k_{final}[x, y, z] = \sqrt{\sum_{d} q_d * k_d^2}$$

with d = [x, y, z]. By using this method, wave properties for every dimension d are obtained separately and get combined in the last step of the algorithm to a three-dimensional field of local wave properties. q denotes the quality flag. Included are

- 5 different quality checks. Fist, the amplitude and wave number are checked for at least a half undamped wave considering the packet length l ( $k \times l > \pi$ ). Second, noise suppression is considered by taking into account the standard deviation of the data and creating a threshold under which results are rejected (Glatt and Wirth, 2014). Third, high frequency fluctuations in amplitude and wave number are smoothed by a running mean over a number of grid points, respecting the minimum wave number  $k_{min}$  of the bandpass filter.
- 10 This method does not cover the temporal propagation of a wave packet and therefore the wave number misses sign (k = ±k[x]). In the following we introduce UWaDi by a step-by-step outline. Further, we validate it with a well-defined test wave packet in comparison with other methods. In general, UWaDi is a script package which allows the user to steer data preprocessing, the main wave analysis and data plotting, from a set of namelists. This package is coded in open source software such as NCL and Fortran. Its multi-purpose applicability on a set of arbitrary waves, e.g. gravity waves or planetary waves, defines its unified

15 character.

[revised manuscript text omitted]

**15 Analytical test case**

**2.2 Validation of the method**

Synthetical one- and two-dimensional test cases are considered to proof the reliability of UWaDi.

**One-dimensional wave packet**

An one-dimensional example is adapted from Zimin et al. (2003). Two consecutive wave packets with the wave numbers 4 and

20 9 are given by For a comparison of available methods that obtain wave quantities we choose the test case presented in Zimin et al. (2003) (Fig. 2a). A couple of localized wave packets with the wave numbers 4 and 9 is given in one dimension on the interval  $[0, 4\pi]$  by

$$f[x] = \exp\left[-(x-4.5)^2\right]\cos(4x) + \exp\left[-(x-7.5)^2\right]\cos(9x).$$
(13)

UWaDi is performed for bandpass limits of wave numbers of 1 and 10. The amplitude peaks with 1.0 and envelopes the
 consecutive wave packets well (Fig. 1). Crosses mark the calculations passing the quality checks. The red lines show the threshold under which calculated values are treated as noise. This threshold can be chosen by the user to suit the individual

Figure 1. One-dimensional test function (bold line, left figure) adapted from Zimin et al. (2003). UWaDi-calculated amplitude enveloping the test function (left) with crosses marking valid values according to the quality check. Additionally, the red lines specify thresholds belonging to the quality check which suppresses noisy signals. The calculated wave number (right) uses the same markers.

---

## Referee Report (RR2)

Review

Diagnosis of Local Gravity Wave Properties during a Sudden Stratospheric Warming

by

Lena Schoon and Christoph Zülicke

The paper's intention is to diagnose local properties of gravity waves (essentially, amplitude and wavenumber vector) from 3D gridded data by a tool named "UWaDi". The paper essentially consists of two portions. One part is the presentation of the wave diagnosis tool and the second part describes an application to 3D IFS analyses for one selected day in January 2016. Unfortunately, both parts are half-baked. And both parts would gain immensely if they were written such as they were individual scientific contributions to this journal.

**Part 1**: The presentation of a new software tool (line 13, page 1: "Here, we want to introduce a new method named ″Unified Wave Diagnosis" (UWaDi).") which is coded with open source software should be happily resolved by offering it also as open source code. Currently, the standard is to publish the open source software under a certain license and to allow access to the code via a repository. Requests to the authors (line 24, page 15) are simply not up-to-date as authors might leave institutions, for example. There are other remarks concerning the outline and scope of the "UWaDi"-tool which are listed below.

**Part 2**: At some places in the text, the authors try to state their scientific goals which shall be tackled with the "UWaDI"-tool: line 1, page 1 ("The selective transmission of gravity waves through an inhomogeneous mean flow is investigated.", line 7, page 3: "We are interested in the longitude-dependent transmission of GWs during a SSW."). This is a relevant topic; however, the results presented shed only some spotlight on the whole dynamical processes during the selected period. Only one selected time is considered and both the temporal process of propagating waves (transmission) and the specification of the wave sources remain rather speculative. As above: the paper would greatly benefit if this part is separated from the paper and investigated in an own, full-length scientific contribution.

Specific Remarks

**Abstract**:

- it should be mentioned right at the beginning that the method is only applicable for gridded 3D data

- line 3: "wave properties": they should be listed here for completeness

- line 3: "wave-containing data" should be specified; later, this turns out to be a crucial point of the analysis where a preselection of wave modes is made by the choice of the horizontal divergence as "wave containing data"; furthermore, the retrieval of this field in a specified spectral resolution certainly impacts the results which is not discussed in the paper at all

- line 7: scientific result: "confirm locally different transmission"; rather weak statement: as no propagation is investigated but only still images are presented you can only refer to "appearance" of gravity waves under the given background conditions (see your own statement in line 14, page 3)

- line 8: I thought, the local wavenumbers are the output of the tool; why "additionally ..."?

- line 9: very speculative statement

**Introduction**:

- it is certainly an advantage to discuss and compare the different methods for the retrieval of wave properties; there are two points which should be added. First: make clear that "UWaDi" needs regularly gridded data from the very first beginning to point out the difference to other methods. Second, I miss the state-of-the-art review of the application of Hilbert transforms to atmospheric data in 3D. For example, the sentences in the paragraph about the Hilbert transform (page 2, 3rd para) can lead to a misunderstanding: "Kinoshita and Sato (2013) provide a three-dimensional application on Rossby and GWs. Our method comes up with an enhancement for three dimensions and the additionally provision of the wave number in every dimension which was not presented before." as they give the impression that K&S2013 use the Hilbert transform. Unfortunately, the word "Hilbert transform" does not appear in that paper. Maybe, I didn't read the paper carefully enough but I suggest to check this statement and to add appropriate applications of Hilbert transforms in atmospheric 3D data discussing their advantages and disadvantages (Glatt and Wirth, you know these papers).

- the scientific focus on the characterization of gravity waves appearing in high-resolution IFS data is good; the authors have all means at hand to provide a substantial contribution; however, this would be a full paper and not only an addition to the presentation of "UWaDI". See part 2 above!

- line 16, page 3: the full set of output of "UWaDi" is unclear to me: Is " ...we will use "UWaDi" with a GW-specific diagnostic." an addition or the standard set?

- the intention to study a period which is " ... very well sampled with observations of GW properties" (line 29, page 3) is very good; unfortunately, the authors do not use them; furthermore, they use a quantity to identify waves which is hardly measurable ("Our approach concentrates on fields of horizontal divergence of ECMWF IFS data", line 2, page 4); for atmospheric physicists, temperature fluctuations would be a much better choice!

- line 32/33, page 3: I would recommend to avoid such strong statements as "Even the T799 resolution gives proof of correct GW appearance in the stratosphere."

lines 5 -9, page 4: As mentioned above, I have problems to see a discussion about propagation (vertical or horizontal) when you only provide one time snapshots. This must be speculative.

**Method and Data**

(a) The description of the first step (page 4 and 5) is totally incomprehensible:

"Horizontally, the grids are equidistant if they are provided on a regular latitude-longitude grid." I don't think, this statement can be true if distance is measured in meters. Please, provide the correct formulas how you compute the distances for the regular lat-lon grid. This should be specified in a step-by-step outline of the method!

"Vertical interpolation from model levels to equidistant height levels is performed by associating constant heights with pressure levels."

I don't understand this. First of all: are you using IFS data on pressure or model levels? How do you determine the height on these levels? And how do you interpolate the data on an equidistant altitude grid?

"Consider to first separate the fluctuations from the background with appropriate numerical or dynamical filters."

I have no idea what the sentence means and what you are referring to.

(b) You outline the method for 1D fields. Could you specify what $f_x$ means?! Is this $f(x, y_f, z_f, t_f)$ ?? Subscripts "_f" mean at fixed positions or time?

(c) Is Eq (10) correct? delta denotes horizontal divergence, right?! Shouldn't be the "delta" in the formulae a "sigma" as $f_x$ is the divergence in your applications, correct??

(d) Figure 1: Why do you plot negative amplitudes when Eq. (11) takes the positive square-root? What are the thin black lines? The choice of colors could be changed to improve readability.

(e) The subsection about the IFS data must be improved. Please, provide the following information and avoid discussions about other cycles here:

- IFS cycle 41r1 is used; (if you really want to refer to differences of 41r1 to other cycles, see Ehard et al, 2018: Comparing ECMWF high resolution analyses to lidar temperature measurements in the middle atmosphere. Q.J.R. Meteorol. Soc. doi:10.1002/qj.3206;)

For which spectral resolution you retrieve the data from the archive?

Maybe, I have to mention that the retrieval of data goes along in a two-step procedure. You can retrieve data using the full set of spectral coefficients (in MARS retrieval RESOL=AV) or you can specify a zonal wavenumber (e.g., RESOL=309) you want. If you do not specify anything, default values are chosen, see: https://software.ecmwf.int/wiki). After this first step, the data are interpolated on a given grid (specification of the GRID parameter in the retrieval).

- on which regular lat/lon grid are the data interpolated to?

- do you use data on pressure or model levels?

- how do you compute the altitude on these levels?

(f) Section 2.4 and Appendix B

Equation (14) cannot be derived by the relations provided in appendix B.

Appendix B: The Appendix is about the "Derivation of the TOTAL wave energy" not only about the kinetic one. The presented equations cannot lead to the final result (B4). There is a mistake (I think, omega is missing) in the provided formula for vorticity tendency (line 23, page 16). Actually, zeta is the default symbol for relative vorticity not xi. Referring to two text books (Vallis, Eq 4.69, Gill, Eq. 7.10.7), the formula for the absolute vorticity in a rotating frame of reference is $D(zeta+f)/Dt = -(f+zeta)*delta$. Why do you obviously use $d\ zeta/dt=-f*delta$ only? Discuss this approximation! Furthermore, I obtain different signs in (B3) when I use the provided relations between vorticity and divergence and between divergence and vertical velocity. Please, check!!

line 8, page 9: What is Omega?

**Results**

- for which time the analysis is performed?

- line 14, page 9: Change formulation "The jet streak above northern Europe is decelerating" if you just refer to the wind inside the jet. If you refer to the propagation of the jet streak itself and its deceleration, you should provide evidence.

- line 16: "Equal patterns appear above eastern Siberia .." I cannot see EQUAL patterns.

- Fig 2b is not mentioned in the text, but reference to it fits in line 16.

- The details in Figure 2 are hardly visible. Maybe, different line increments might increase the readability.

- line 20: If it is "more convenient in terms of wavelengths" you should plot them instead of wavenumbers in the respective Figures 3 and 5

- line 21: I found an upper limit at 196 km (2 pi/3.2 $10^{-5}$ m$^{-1}$) not 165 km.

- line 20 and line 21: the horizontal wavenumber changes by about 66%, the vertical only by about 45% over the height region. Thus, the statement: "In the zonal mean the horizontal wave number remains nearly constant with increasing altitude" cannot be supported by the data provided in Fig. 3.

- very speculative and not provided by evidence: "independent from the overall synoptic situation and is therefore expected to be an artefact of artificial wave damping from the IFS sponge layer." (line 24, page 9)

- I do not get the meaning of "We did not find significant differences between spatial averaging over areas of some longitudes extension and the local profiles (not shown)." What mean areas? Do you also average zonally?

- line 29, page 9: "The low-pass filter applied in Step 8 helps to overcome massive grid-point to grid-point fluctuations." The documentation of this step could be a nice addition to a more substantial documentation of Part 1 (above).

- line 1, page 10: do you really mean "descented"?

- line 9, page 10: "The not trustworthy areas are excluded." See above for Part 1.

- line 12, page 12: "Altogether, GW emissions seems to take place in the .." and line 16, same page "..Mountains, hence, mountain waves are most likely." are very speculative and not fully supported by the data provided. This issue would belong to the Part 2 of the a possible separated paper.

**Discussion**

first paragraph: "These findings were obtained with the box-based S-3D algorithm. We add some spatially more refined analysis with UWaDi." What about to really apply the other methods used for the comparision with the synthetic data to the real case considered here? This would add substance!

line 27, page 12: " ... and agree with findings of Limpasuvan et al. (2011)." and line 29: " ... This is close to the findings of Krisch et al. (2017), who .." If you really intent a quantitative comparision, I would recommend to avoid statements like those. The Limpasuvan case lacks comparability in the background wind field, I guess. And, it is not clear what exactly you compare and refer to. The Krisch case is even more dangerous as they point at the dominant horizontal propagation which is omitted here.

line 30, page 12: there is no evidence of data to provide a proof of the statement " ... from the 25 January to 30 January the overall approaching flow direction did not change above northern Europe and comparable GW characteristics can be expected."

line 4, page 14: again, the hypothesis of a wave source at the stratospheric jet remains hypothetical unless more facts of evidence are provided. The geographical association of large wave action with the nose of the jet does not mean undoubtedly that the jet is the source.

line 15..., page 14: the identification of a critical level by detecting a local maximum in $k_z$ is a possible choice which alone is not sufficient to proof the critical level absorption. As you have all tools at hand, why you do not derive the group and phase velocities of wave packets and identify their sources? This would give much stronger evidence.

line 19/20, page 14: I don't understand "The near-inertial GWs are not subject of absorption." Why?? There exists so-called Jones critical levels.

**Summary and Conclusions**

I would have expected here NEW insights into the scientific topics which were mentioned in the Introduction and in the Abstract: "The selective transmission of gravity waves through an inhomogeneous mean flow is investigated." One finds a summary of "UWaDI" and potential future developments "UWaDi may also provide local estimates for more complex tools such as the combined Rossby wave and gravity wave diagnostics of Kinoshita and Sato (2013)." (line 8, page 16). Scientific conclusions are essentially absent. Again, it must be stressed that the results of the paper do not allow any conclusions with respect to propagation as only one time is considered and very simplifying assumptions are made.

Typos:

inhomogenous -> inhomogeneous (line 1 in Abstract)

artifical    -> artificial (page 9, line 24)

". wave action .." -> ". Wave action …" (page 8, line 28)

imhogeneous  -> inhomogeneous (page 8, line 27)

---

## Author Response (AR2)

**Reply to comments on acp-2017-472**

L. Schoon and Ch. Zülicke

March 23, 2018

**1. Reply to Co-Editor**

We appreciate the comments and recommendations of the Co-Editor to the revised manuscript and answer the risen up issues point-by-point. With Appendix A we provide details of our revised validation excersise. The revised manuscript with the highlighted changes can be found in Appendix B.

1) Reviewer 1 appreciates the addition of a methodology section to the manuscript. However, he/she has identified flaws and actually provides both IDL-code as well as resultplots achieved with this code to check your results. In your revised manuscript you will need to make sure that the methodology section is corrected.

We are very grateful for the hints of Reviewer 1 to ambiguities according the method comparison section. We revised it carefully (see Appendix A). Details can be found in the point-by-point-answer to the comments of Reviewer 1. Most relevant change was the use of a narrower box width which improved the results of the sinusoidal fit. However, the Hilbert transform performed best without any a-priori assumption. The provision of source code from the FZ Jülich helped considerably with this issue.

2) Reviewer 2 still insists that the manuscript should rather be split in two parts, one addressing the new methodology and one addressing the application of the methodology to obtain new scientific results. I understand that it is your approach to introduce the method and that you try to convince the reader of its usefulness by showing some initial, to some extent preliminary, results. This needs to be made clear from the text and also from the title.

I therefore request that you change the title of this manuscript to reflect this. A potential title could be: A novel method for the extraction of gravity wave parameters from gridded 3d data: method description and initial scientific results.

We agree with the Co-Editor that the focus of this publication is on the introduction of the new method and the scientific results are used to convince the reader on its usefulness. Thus, we kept the atmospheric dynamic discussion part short. We changed the title as suggested and added explicit formulation on this into the text.

3) Ideally, in order to improve the scientific quality of the second part of the manuscript, the analysis should be extended to time dependent data. If this proves to be beyond the scope of the present study I request that you state this explicitly and explain what steps will need to be taken.

We had a look at the temporal extend of the analysed mSSW and decided that this will be far beyond the scope of this paper. Indeed, we found several interesting transient features in this time. Their reasonable presentation, however, would require the presentation of wind and wave fields including indicators for their sources. Because this would add too much information to the present paper including the introduction of a new source diagnostics, we decided to leave this issue for a forthcoming paper. For now, we stay with the snapshots of horizontal and vertical sections of wave packets and focus on their interpretation, improve the quality of figures and add some more parameters to the interpretation. We are sure that our initial scientific discussion provided here shows the reader the unique usefulness of the method.

4) Besides the points that I have emphasized above, it goes without saying that also all other specific comments from both reviewers will need to be addressed as usual. Detailed responses to each of the reviewers can be found below, including additional information in the Appendix.

**2. Reply to Reviewer 1**

The revised version of the paper is much improved. There is now a good motivation, the method description is stringent and the examples shown for the application are well chosen. However, in the method comparison S3D is not correctly applied. That is detailed in the major comment below and results for a correct application shown in the attached Figure. In addition, there are some inaccuracies in the description of the other methods.

You should also point out the basic assumption of attributing all variance to a single wave together with the method overview. Some additional specific comments are given below. If the method intercomparison will be corrected and the other comments taken into account I recommend publication of the manuscript in ACP.

Major comment:

You are using an 1D data set. At this you cannot apply S3D or 3DST. There are two ways attacking the problem:

a) You can expand the data set to a 3D data set.

b) You can use an 1D sinusoidal fit in a sliding window

These two possibilities are discussed here for S3D, code for reproduction is attached and can be run with the general IDL package of the Juelich remote sensing group.

a) In order to produce a data set comparable to normal S3D application, expansion of the test function to 3D should include a wave structure in the two other spatial dimensions as well. We can choose two ways of doing this:

a.1) Use the same wavelength for both wave packets

a.2) Use different wavelengths for bot wave packets

Employing a cube-volume of the size of 0.5 in the direction of the variation, a.1 (red) is very close to the solution of the Hilbert transform, a.2 (blue) can separate between the wavelengths and has a step-wise transition in wavenumber without the spurious wavenumbers between the two solutions. The results are shown in the attached figure, panels a (amplitude) and b (wavenumber). However, for real-world problems we probably would apply a size of 1.0 or 1.5 in order to get more confidence on not ideally-shaped wave patterns. In the case of box-size 1.5 (panels c and d) we find a notable smearing of the envelope (15% underestimation of the peak-amplitude), still closer to the reference than the Stockwell transform. All these fits draw information from the two additional directions which we have chosen to be homogeneous in amplitude and wavelength.

b) A mere 1D sinusoidal fit of window length 1.5 produces somewhat smeared amplitudes and some spurious oscillations (panels e and f). That can also be emulated by S3D and prescribing no wavelength information in the two additional directions (A.3).

For the paper you could follow one of the following two approaches:

1. Use the 1D solution. Do not call it S3D. Mention that the additional dimensions will add to the quality of the results

2. Discuss a.1/a.2 for a realistic window length of 1.5. Note in the text that the window

**length has an influence.**

We follow 1. And stay with one dimension according to the definition of the test case. In our view, the quality improvement from adding more dimensions is due to a certain phase average effect. With regard to the window width we use a size of 1.5 for the sinusoidal fit and found an acceptable amplitude. A further decrease of window width leads to a quality decrease for the wave number estimate. Numerous tests are documented in a technical report which we attach to the attention of the reviewer (Appendix A). In the text of the paper, we extend the documentation on the setup for the alternative diagnosis methods including statements on the influence of the window size.

**Anyway, none of this approaches performs as lousy as the so-called S3D solution in the method intercomparison.**

In the paper, we used a window of 7.85 for all the box-based methods (sinusoidal fit and auto-correlation function analysis). This was oriented on a-priori reasoning that the largest wavelength should be at least five times covered by the window. We had in mind the statistical significance of the wave number estimate, and not the best-possible spatial resolution. However, in order to be practical, we performed a number of tests and finally adopted those a-priori windows which lead to the best results in both amplitude and wave number. See attached technical report (Appendix A).

Similarly, the Stockwell transform could perform better than in this test. There is a tuning factor c to reduce the width of the Gaussian envelope and also for the ST the third dimension adds information. One could thus also tune ST for a closer match in the 3D case. Also this should shine up in the discussion

We hesitate to tune the Stockwell transform because the advantageous variance conservation would be violated (see Appendix A). This reasoning is now included in the text.

**Summarizing: the idea of a simple method intercomparison adds value to the paper, but it must be done right!**

The width of the box window is the major responsible for the quality of the box-based methods. Now, we have decreased the window width which improved the quality, and we have completed the documentation of the method intercomparison.

**Minor comments:**

P4L7 The importance of oblique propagation is not restricted to the mesosphere. Ehard

et al show this for the mid stratosphere but it may be quite substantial already in the UTLS (e.g. Krisch et al). Oblique propagation hence just redistributes selective transmission and makes the analysis more complicated. It is definitely a limiting factor for the interpretation of vertical profiles and some discussion should be added.

We add the fact that oblique propagation plays an important role from UTLS to the mesosphere and point out clearer that we focus on vertical propagation.

P3L4 In principle you should be able to calculate time-space spectra of u and w and calculate cospectra for the respective wave modes (e.g. Alexander et al. 2004). This is not limited by the size of the analysis volume and can be used to apply zonal mean quantities. Given that one reviewer asked for the motivation to use horizontal divergence, that is still not sufficiently motivated. One motivation could be the correction terms which need to be applied for the influence of Coriolis force on pseudomomentum flux.

The main motivation of using the horizontal divergence is that we get ageostrophic wave components without the necessity of filtering data beforehand. This is given on P2. In the analysis part of the paper we derive the wave action from the horizontal divergence and choose this quantity to be the centre of our discussion.

Generally, with the help of dispersion relations several quantities can be estimated with the results of UWaDi, like the zonal vertical pseudo momentum flux, etc. We waive these possibilities to not blow up this paper.

Figure 5c and interpretation: The vertical wavelength is shortest in the wind maximum and then increases(!) towards where you assume the critical level is. There is only a very small peak at that altitude (much longer wavelengths than at the assumed excitation altitudes), comparable in size to other local variations. There are a number of explanations beside critical level which could explain your finding of a maximum in the jet, for instance oblique propagation, partial reflection ... So, if you want to retain this discussion, you should offer more proof and infer the phase speed for the considered waves considering the phase shift between two consecutive time steps of ECMWF.

We added Figure 6 to perform this discussion more in detail. We show the intrinsic frequency, horizontal wavelength and horizontal phase speed. We find the critical level by a rotation of the wave vector and showing that the horizontal phase speed equals the background wind speed.

Specific comments:

P1L25 S3D without hyphen

We changed the notation from S-3D to S3D.

P2L1 Ern and Preusse, 2012 and Ern et al. 2014 use different methods  $\rightarrow$  omit We omitted those.

P2L2 to P2L18 What you say is all correct, but I think you are missing the chance to set this in a broader context. The fundamental problem of all wave analyses methods is given by the fact that you have to compromise between spatial and spectral resolution (cf. the uncertainty relation). A second point is to which extent the sampling and total volume will influence the result. Going along such lines you can order along the spatial resolution:

Fourier Transform: Assumes homogeneity of amplitudes and phases through hole considered volume. Wavelengths determined by volume size. Covers whole spectrum continously. No information loss.

3DST: Assumes homogeneity inside a volume corresponding to one wavelength (or fraction, cf. scaling factor). Wavelengths limited by size of total analysis volume. Selecting largest events implies loss of variance (information).

S3D: Assumes homogeneity inside an predefined analysis volume. Restriction to wavelength by cube size (sensitivity study: reliable for wavelength < 2.5\*cube size). Outcome depends on pre-selected cube size. Selecting leading (not largest) components implies loss of variance (information).

Clearer argumentation considering the suggestions of Reviewer 1 on S3D and 3D ST was added.

Hilbert Transform: Does not assume homogeneity. No limitation on wavelengths except Nyquist. Gives information of amplitude and phase, wavelength information inferred from local phase gradient -> all variance is attributed to one wave mode. This prize you have to pay is important and should be mentioned for a fair assessment! We edited this part considering this suggestions.

Anyway, if you present these different compromises between wave resolution and spatial resolution you can claim that Hilbert transform provides the one we are missing. UWADI thus provides the user with a complementary tool for investigating wave events. All localized methods can be shifted by steps smaller than the analysis method, so one is, in principle, able to calculate wave parameters for each original point for 3DST and S3D. A large asset of the Hilbert transform is that it is computationally cheap. Some more discussion on that was added to the validation of the method section 2.2.

**More specific:**

P2L2 "two" is most often chosen, but not a general limitation. However, as you say in your next sentence, always a small number is taken. I would just omit the "two".
We just omitted the "two".

**P2L8 3DST, please use notations as in the reference papers**

In Lehmann et al. (2012) the notation "S3D" is used. We adopted that. In Wright et al. (2017) the notations "3-D ST" and "S3-D" occur. To compromise we use for the Stockwell method "3D ST" now.

P2L11 The only assumption made is that of upward propagating waves. This ambiguity between e.g. eastward-upward and westward-downward cannot be decided from temperature observations alone and is not a restriction of the method. We omitted this mistakable statement.

P2L12 S3D uses the largest described variance, i.e. minimum chi-square to pick the wave, 3DST the largest amplitudes. In both cases it can be more than one wave. Anyway, that should not be relevant for what you want to say, perhaps: Both S3D and 3DST use a small number of the most-prominent waves. This leaves some variance unattributed and hence means a loss of information.

We follow the suggestion and added that statement.

What perhaps is more important in this context: Both methods implicitly assume homogeneity of amplitude and wave vector, S3D inside one analysis volume, 3DST inside a volume corresponding to one wavelength. This is a loss of spatial information. As mentioned before, we edited the method introduction part as well as the comparison section and are sure that the pros and cons of the different methods are clearer now.

P2L27 to make clear it is not flow over convection: by flow over orography, by convection ...
The "by" is added.

P4L18 use e.g. for the reference The "e.g." is added.

*P4L22 The phase of the wave ...* Edited.

P6L4 Alienation of outliers is taken care of by two different quality. Please reformulate The word "checks" was missing.

P8L11 You can focus on these scales. However, a vertical limit of 15km will lead to shift part of the spectrum in and out the so-chosen visibility filter (Alexander, JGR, 1998 and later literature on this topic).

The authors added some discussion on observational filter limitations and provide for one of the vertical profiles additionally horizontal wavelength, horizontal phase speed and intrinsic frequency to make sure to not misinterpret the wave behaviour due to e.g. refraction of the wave outside of our filter limits.

P8L27 do not interact with the mean flow.  $*W^*ave$  action In order to avoid confusion we reformulated this passage.

P9L8 In order to make the paper easier to read you could indicate by the notation whether a quantity is intrinsic or ground based; e.g. use  $\hat{\omega}$  and  $c_g$ . The notation was changed.

P9L9 The wind also needs to be constant in time. Perhaps better turn around: If the wind ... Quote a ray-tracing paper, e.g. Lighthill or Marks and Eckermann.

Our discussion is based on a stationary homogeneous background wind which only varies in vertical direction. This is now stated in the text. Any deeper discussions on wave propagation in a time-depented mean flow will be part of an upcoming paper. We make this clearer in some reformulations throughout the revised manuscript.

*P9L14 swap the two sentences: The displaced ... The jet streak ...* Swapped. P9L16 I am wondering: is eastern Siberia really a comparable situation to Europe? The GW fronts seem more aligned than across the winds and there is less deceleration but strong curvature.

The formulation was misleading. We reformulated that parts considering your comment.

P9L23 However, it is counter-intuitive. Both a stronger damping and a coarser vertical grid would let expect the shorter waves to be more strongly damped. Did you try what happens if you increase the upper limit of your analysis range?

After the first round of revision we concentrated on finding the right data and focussed on comparing different data resolutions with the same setting of our method. This is discussed in Sec.2.3. In so far, the indicated decreasing wavelength above 50 km did not depend on the horizontal resolution. It appeared for different wind conditions (compare the zonal mean profile and case ③). Finally, we could not find a proofable explanation of this behaviour. In order to concentrate the paper on the methodology, we eliminated all kinds of interpretations of the topic of sponge layer and damping to not run into too much speculation. This is not a topic that can be discussed satisfactorily as we follow the editor's advice to concentrate on the method.

*Fig 2: Omit \*\* on the exponents.* Omitted

Fig 3: Please generate a second panel with zonal mean wind and N. At constant  $c_g$ , k vertical wavelength is inversely proportional to N, so you should see that as well.

A second panel is created which completes the diagnosed wave numbers with the apparent (ground-based) phase speed and allows for the evaluation of passive upward propagation. We did not plot N because it is a parameter in the estimation of intrinsic frequency and phase speed. The latter would enter a relation like  $\lambda_z = \frac{2\pi}{N}\hat{c}$  and insofar we could not isolate the impact of N. Such an investigation would require the diagnosis of the ground-based (apparent) frequency or phase in seldom situations with changing N over constant u.

Fig 4: It is evident, still: Give the units of the shown quantities, please. The units are added.

P9L30 ? height range ?

Meant was longitudinal. Changed.

P10L1 descented -> descended ? Yes, changed.

It would be nice to indicate the profile locations in Fig 2a. We agree and added the circled profile numbers to this figure.

P10L7/L9 2\* Overall Reformulated.

P10L10 ... remains constant. Puzzling, that's in the sponge.

The wave action remains constant above 25km altitude. The first sponge starts at 30 km. We reformulated this sentence, but the main statement remains true.

P12L7 This could be one of the major assets of a local method - that one sees the scaling of the wave properties with the varying wind ...

Yes, and we show this asset with the other local profiles. In profile (1) the scaling is restricted by the smoothing inside our method. We find it important to mention this restriction.

P12L13 At these altitudes, GWs of vertical wavelength ... We added your suggestions.

P12L19 an SSW (i.e. an EsEsDoubleU) The whole text was searched and all "n"s added.

P12L21 They point out theoretically that during the upward propagation of GWs these waves (comma, upward) Comma and "s" are deleted.

P12L22 Actually that was profile-based MEM/HA (Preusse et al, JGR, 2002) with a 10km running window. For current day limb sounders we get only an estimate of the absolute horizontal wavelength and do not estimate wave action. We omitted this statement.

P12L25 the longest vertical wave length with 7 km If that were really the longest existing wavelengths AIRS should not see any GWs at all. There should be also cases of substantially longer wavelengths.

We do not state that this is the longest possible wavelength for GWs. We find this wavelength with the limits of our vertical bandpass filter of 1 km to 15 km because we are interested in GWs emitted from jets that are supposed to have wavelengths in this limit. Nevertheless, we reformulated that statement to make clearer that it is the longest wavelengths according to our method settings.

P12L26 Scandinavian Changed to capital "S".

 $P14L12 \ put \rightarrow exerted$ Changed, the whole paragraph was reformulated.

P14L17 jet-generated GWs tend to have such phase speeds. (convection would be much faster).

Changed, the whole paragraph was reformulated.

P14L19 relatively high

Changed, the whole paragraph was reformulated.

P14L27 Low wind speed and low wave action, maybe. In cases of larger waves these may still dominate the statistical effects leading to the higher wavenumbers.

We deleted the discussion part on the sponge layer according to Reviewer 2 to avoid an incidentally discussion of a topic that could fill another whole paper.

 $P14L32 \text{ such } \rightarrow thus ? or omit it Omitted.$

*P15L2 nomination -> assumption or attribution of the variance to* Changed accordingly.

**3. Reply to Reviewer 2**

**Review**

Diagnosis of Local Gravity Wave Properties during a Sudden Stratospheric Warming by

**Lena Schoon and Christoph Zülicke**

The paper's intention is to diagnose local properties of gravity waves (essentially, amplitude and wavenumber vector) from 3D gridded data by a tool named "UWaDi". The paper essentially consists of two portions. One part is the presentation of the wave diagnosis tool and the second part describes an application to 3D IFS analyses for one selected day in January 2016. Unfortunately, both parts are half-baked. And both parts would gain immensely if they were written such as they were individual scientific contributions to this journal. Part 1: The presentation of a new software tool (line 13, page 1: "Here, we want to introduce a new method named "Unified Wave Diagnosis" (UWaDi).") which is coded with open source software should be happily resolved by offering it also as open source code. Currently, the standard is to publish the open source software under a certain license and to allow access to the code via a repository. Requests to the authors (line 24, page 15) are simply not up-to-date as authors might leave institutions, for example. There are other remarks concerning the outline and scope of the "UWaDi"-tool which are listed below.

With regard to the publication of source code and any used data we follow the guidelines of the journal. Next to that, we will provide the manual of the method as well as a list of publications containing results obtained by UWaDi on a web page of the authors institute, the Leibniz-Institute of Atmospheric Physics Kühlungsborn. The source code of the method will be provided upon request until this first method-introducing publications is still not finally published regularly to make sure that if the source code is used by others, correct citation is possible.

Part 2: At some places in the text, the authors try to state their scientific goals which shall be tackled with the "UWaDI"-tool: line 1, page 1 ("The selective transmission of gravity waves through an inhomogeneous mean flow is investigated.", line 7, page 3: "We are interested in the longitude-dependent transmission of GWs during a SSW."). This is a relevant topic; however, the results presented shed only some spotlight on the whole dynamical processes during the selected period. Only one selected time is considered and both the temporal process of propagating waves (transmission) and the specification of the wave sources remain rather speculative. As above: the paper would greatly benefit if this part is separated from the paper and investigated in an own, full-length scientific contribution.

We follow the suggestions of the editor and reformulated the title as well as some specific statements in the text concerning our scientific goals for this paper. We point out more clearly that the focus on the paper lies on the introduction of the method. The scientific discussion is kept short, restrictions are named and further analyses containing temporal evolution and source discussion are mentioned to be subject of future studies. The application case here is used to show the reader that the method can be used to find local spatial wave packets in a dynamically inhomogeneous situation. See also the reply to the editor (Sec. 1).

Specific Remarks Abstract: - it should be mentioned right at the beginning that the method is only applicable for gridded 3D data Now it is mentioned in the title, by suggestion of the editor.

- line 3: "wave properties": they should be listed here for completeness They are listed now.

- line 3: "wave-containing data" should be specified; later, this turns out to be a crucial point of the analysis where a preselection of wave modes is made by the choice of the horizontal divergence as "wave containing data"; furthermore, the retrieval of this field in a specified spectral resolution certainly impacts the results which is not discussed in the paper at all

The separation of wave and background from each other is not topic of this paper. The method needs a wave signal at the beginning. By choosing the horizontal divergence we count for the dynamical filtering of the geostrophic and ageostrophic flow. In the ongoing paper we derive the wave action from the horizontal divergence as the discussed quantity.

The method includes a bandpass filter. The method attributes all variance to one wave mode, therefore this filter indicates the finding of the waves the user is interested in. Information on the resolution of the used data is added. - line 7: scientific result: "confirm locally different transmission"; rather weak statement: as no propagation is investigated but only still images are presented you can only refer to "appearance" of gravity waves under the given background conditions (see your own statement in line 14, page 3)

As mentioned above, we keep the scientific discussion short to keep the introduction of the new method in the focus of the paper. We changed this statement to "appearance" according your suggestion.

- line 8: I thought, the local wavenumbers are the output of the tool; why "additionally ...."?

This is a misunderstanding. The method gives the amplitude and the three-dimensional wave number. Thus, the wave number is an additional output. Nevertheless, we omit it.

**- line 9: very speculative statement**

If this refers to the hint on a GW generated by the stratospheric jet, we give more hints on that in the discussion part of the paper now. It is also noted further scientific investigation is intended. This is adequately addressed in the abstract. This issue will be studied in a forthcoming paper.

**Introduction:**

- it is certainly an advantage to discuss and compare the different methods for the retrieval of wave properties; there are two points which should be added. First: make clear that "UWaDi" needs regularly gridded data from the very first beginning to point out the difference to other methods. Second, I miss the state-of-the-art review of the application of Hilbert transforms to atmospheric data in 3D. For example, the sentences in the paragraph about the Hilbert transform (page 2, 3rd para) can lead to a misunderstanding: "Kinoshita and Sato (2013) provide a three-dimensional application on Rossby and GWs. Our method comes up with an enhancement for three dimensions and the additionally provision of the wave number in every dimension which was not presented before." as they give the impression that K&S2013 use the Hilbert transform. Unfortunately, the word "Hilbert transform" does not appear in that paper. Maybe, I didn't read the paper carefully enough but I suggest to check this statement and to add appropriate applications of Hilbert transforms in atmospheric 3D data discussing their advantages and disadvantages (Glatt and Wirth, you know these papers). Referring to your point 1: The title and abstract were changed according to that. Referring to your point 2: We apologize and corrected the citation. It has to be Sato et al. (2013). Glatt and Wirth (2014) do not use the Hilbert transform in three dimensions. They operate on a longitude-plane and perform the Hilbert Transform for every latitude seperately. We added that.

- the scientific focus on the characterization of gravity waves appearing in high-resolution IFS data is good; the authors have all means at hand to provide a substantial contribution; however, this would be a full paper and not only an addition to the presentation of "UWaDI". See part 2 above!

Indeed, this kind of paper is in preparation which builds on the first scientific results which are given in this publication.

- line 16, page 3: the full set of output of "UWaDi" is unclear to me: Is "...we will use "UWaDi" with a GW-specific diagnostic." an addition or the standard set?

We further extended the method-description part to make that clearer. This specific statement is omitted now, to not rise up unclarities at this point of the paper. What was meant is that UWaDi can be used to obtain waves of any kind of wavelengths. By choosing the limits of the implemented bandpass filter one restricts the results to a specific wave set. We are interested in GWs, therefore we adapt the UWaDi algorithm to that.

- the intention to study a period which is "... very well sampled with observations of GW properties" (line 29, page 3) is very good; unfortunately, the authors do not use them; furthermore, they use a quantity to identify waves which is hardly measurable ("Our approach concentrates on fields of horizontal divergence of ECMWF IFS data", line 2, page 4); for atmospheric physicists, temperature fluctuations would be a much better choice!

Throughout the paper we mention several times that we chose a period of time where measurement campaigns took place. Therefore, these period is very well sampled with observations. We further mention, that it is our aim to contribute to the results of these measurement campaigns with our findings.

The second point of using the horizontal divergence was already discussed above. We choose the horizontal divergence to avoid the separation of wave and background. We use the horizontal divergence to derive the wave action as the analysed quantity. We mention that UWaDi can be used with any other input quantity which contains a wave

signal. If someone wants to use wind or temperature fluctuations this is truly possible. (No 3)

- line 32/33, page 3: I would recommend to avoid such strong statements as "Even the T799 resolution gives proof of correct GW appearance in the stratosphere." As recommended, we avoid such statements.

lines 5 -9, page 4: As mentioned above, I have problems to see a discussion about propagation (vertical or horizontal) when you only provide one time snapshots. This must be speculative.

We reformulated this part to make clear what our purpose is. "We concentrate on vertical profiles of GW appearance to give a first impression of the functionality of the this method."

**$Method \ and \ Data$**

(a) The description of the first step (page 4 and 5) is totally incomprehensible: "Horizontally, the grids are equidistant if they are provided on a regular latitude-longitude grid." I don't think, this statement can be true if distance is measured in meters. Please, provide the correct formulas how you compute the distances for the regular lat-lon grid. This should be specified in a step-by-step outline of the method!

We added more information on that in the description part. Point 1 describes the calculation of grid points.

"Vertical interpolation from model levels to equidistant height levels is performed by associating constant heights with pressure levels."

I don't understand this. First of all: are you using IFS data on pressure or model levels? How do you determine the height on these levels? And how do you interpolate the data on an equidistant altitude grid?

Point No 2 in the step-by-step outline answers these questions now.

"Consider to first separate the fluctuations from the background with appropriate numerical or dynamical filters." I have no idea what the sentence means and what you are referring to.

Point No 3 of the step-by-step outline was reformulated.

(b) You outline the method for 1D fields. Could you specify what  $f_x$  means?! Is this  $f(x, y\_f, z\_f, t\_f)$ ?? Subscripts "\_f" mean at fixed positions or time? We added a sentence to explain the indices in Point No 4.

(c) Is Eq (10) correct? delta denotes horizontal divergence, right?! Shouldn't be the "delta" in the formulae a "sigma" as  $f_x$  is the divergence in your applications, correct??

Yes, you are correct. The sigma was added, as well as a symbol description. No 11.

(d) Figure 1: Why do you plot negative amplitudes when Eq. (11) takes the positive square-root? What are the thin black lines? The choice of colors could be changed to improve readability.

The negative amplitude is shown to include positive and negative variations of the wave packet amplitude. A description on the thin lines (the envelope) is added. Colours were changed.

(e) The subsection about the IFS data must be improved. Please, provide the following information and avoid discussions about other cycles here:

- IFS cycle 41r1 is used; (if you really want to refer to differences of 41r1 to other cycles, see Ehard et al, 2018: Comparing ECMWF high resolution analyses to lidar temperature measurements in the middle atmosphere. Q.J.R. Meteorol. Soc. doi:10.1002/qj.3206;) The discussion part on the other cycles is omitted and the reference included.

For which spectral resolution you retrieve the data from the archive? T511. Information added on P9L??

Maybe, I have to mention that the retrieval of data goes along in a two-step procedure. You can retrieve data using the full set of spectral coefficients (in MARS retrieval RE-SOL=AV) or you can specify a zonal wavenumber (e.g., RESOL=309) you want. If you do not specify anything, default values are chosen, see: https://software.ecmwf.int/wiki). After this first step, the data are interpolated on a given grid (specification of the GRID parameter in the retrieval).

- on which regular lat/lon grid are the data interpolated to?

As mentioned in the text the grid size we find adequate is 0.36.

- do you use data on pressure or model levels? Already answered. Model levels.

- how do you compute the altitude on these levels? Already answered.

(f) Section 2.4 and Appendix B Equation (14) cannot be derived by the relations provided in appendix B.

Appendix B: The Appendix is about the "Derivation of the TOTAL wave energy" not only about the kinetic one. The presented equations cannot lead to the final result (B4). There is a mistake (I think, omega is missing) in the provided formula for vorticity tendency (line 23, page 16). Actually, zeta is the default symbol for relative vorticity not xi. Referring to two text books (Vallis, Eq 4.69, Gill, Eq. 7.10.7), the formula for the absolute vorticity in a rotating frame of reference is  $D(zeta+f)/Dt = -(f+zeta)^*$ delta. Why do you obviously use d zeta/dt=-f\*delta only? Discuss this approximation! Furthermore, I obtain different signs in (B3) when I use the provided relations between vorticity and divergence and between divergence and vertical velocity. Please, check!!

We are grateful to the reviewer to have inspected the mathematical section. The typos have been eliminated, we use  $\zeta$  for the vertical vorticity component and specify the kind of approximation used. Actually, these are the primitive equations in f plane Boussinesq approximation linearised around a resting environment as appropriate for hydrostatic inertia-gravity waves (see Gill, section 8.4., for example).

line 8, page 9: What is Omega? Description added.

Results - for which time the analysis is performed? 0UTC. Added to the text.

- line 14, page 9: Change formulation "The jet streak above northern Europe is decelerating" if you just refer to the wind inside the jet. If you refer to the propagation of the jet streak itself and its deceleration, you should provide evidence. The statement is reformulated. - line 16: "Equal patterns appear above eastern Siberia ..." I cannot see EQUAL patterns. The statement is reformulated.

- Fig 2b is not mentioned in the text, but reference to it fits in line 16. Reference added.

- The details in Figure 2 are hardly visible. Maybe, different line increments might increase the readability. Line thicknesses, color scales and line increments are changed. Fig2

- line 20: If it is "more convenient in terms of wavelengths" you should plot them instead of wavenumbers in the respective Figures 3 and 5 All plots show wavelengths now. Fig3, Fig5, Fig6.

- line 21: I found an upper limit at 196 km (2 pi/3.2 10-5 m-1) not 165 km. With the new plots this paragraph was rewritten.

- line 20 and line 21: the horizontal wavenumber changes by about 66%, the vertical only by about 45% over the height region. Thus, the statement: "In the zonal mean the horizontal wave number remains nearly constant with increasing altitude" cannot be supported by the data provided in Fig. 3.

The part was reformulated including additional information on the horizontal phase speed.

- very speculative and not provided by evidence: "independent from the overall synoptic situation and is therefore expected to be an artefact of artificial wave damping from the IFS sponge layer." (line 24, page 9)

To avoid any misinterpretations we omit the discussion on damping.

- I do not get the meaning of "We did not find significant differences between spatial averaging over areas of some longitudes extension and the local profiles (not shown)." What mean areas? Do you also average zonally?

This was added in the course of revision process 1 because one reviewer had doubts that local profiles show reliable results compared to zonal averaged profiles. We did several case studies and found that the local profiles show as good results as the regional averaged ones. As our purpose is to show the advantages of local wave analysis we mention here, what we have found. Nevertheless, to not cause any misunderstandings, we again omit this statement.

- line 29, page 9: "The low-pass filter applied in Step 8 helps to overcome massive gridpoint to grid-point fluctuations." The documentation of this step could be a nice addition to a more substantial documentation of Part 1 (above).

In the course of deleting the sentence before, this statement was omitted as well. See above.

- line 1, page 10: do you really mean "descented"? Typo. Corrected to descended.

- line 9, page 10: "The not trustworthy areas are excluded." See above for Part 1. It is mentioned in no 2 that areas of high orography provide misleading results. We refer to that here. The statement was reformulated.

- line 12, page 12: "Altogether, GW emissions seems to take place in the ..." and line 16, same page "...Mountains, hence, mountain waves are most likely." are very speculative and not fully supported by the data provided. This issue would belong to the Part 2 of the a possible separated paper.

The statement was reformulated to GW activity.

**Discussion**

first paragraph: "These findings were obtained with the box-based S-3D algorithm. We add some spatially more refined analysis with UWaDi." What about to really apply the other methods used for the comparison with the synthetic data to the real case considered here? This would add substance!

A proven statement on the quality of a method can be made only if the results are known beforehand. Therefore, we choose the Zimin test case. Otherwise you can not decide which number is nearer to "truth" than others.

line 27, page 12: "... and agree with findings of Limpasuvan et al. (2011)." and line 29: "... This is close to the findings of Krisch et al. (2017), who ..." If you really intent a quantitative comparison, I would recommend to avoid statements like those. The Limpasuvan case lacks comparability in the background wind field, I guess. And, it is not clear what exactly you compare and refer to. The Krisch case is even more dangerous as they point at the dominant horizontal propagation which is omitted here.

According to your suggestion we omit the comparison to Krisch et al. (2017). The comparison to Limpasuvan et al. (2011) was reformulated. They did show that during an SSW westward propagating GWs emanate from key topographic features around the polar edge. They also find that these GWs have long vertical wavelengths. This fits to our findings and is worth to be mentioned in the discussion.

line 30, page 12: there is no evidence of data to provide a proof of the statement "... from the 25 January to 30 January the overall approaching flow direction did not change above northern Europe and comparable GW characteristics can be expected." Statement omitted.

line 4, page 14: again, the hypothesis of a wave source at the stratospheric jet remains hypothetical unless more facts of evidence are provided. The geographical association of large wave action with the nose of the jet does not mean undoubtedly that the jet is the source.

See answer below, as this topics belong together.

line 15..., page 14: the identification of a critical level by detecting a local maximum in  $k\_z$  is a possible choice which alone is not sufficient to proof the critical level absorption. As you have all tools at hand, why you do not derive the group and phase velocities of wave packets and identify their sources? This would give much stronger evidence. We adopted the suggestion to include more diagnosed wave characteristics for a more detailed documentation. We decided to do so for the zonal mean profiles and case 2. In these profiles there are no critical (Jones) levels present, and thats why we did not mention them any more. Instead we used the additional data for argumentation against passive propagation and for jet-related sources.

line 19/20, page 14: I don't understand "The near-inertial GWs are not subject of absorption." Why?? There exists so-called Jones critical levels. This discussion was revised. See above.

Summary and Conclusions

I would have expected here NEW insights into the scientific topics which were mentioned in the Introduction and in the Abstract: "The selective transmission of gravity waves through an inhomogeneous mean flow is investigated." One finds a summary of "UWaDI" and potential future developments "UWaDi may also provide local estimates for more complex tools such as the combined Rossby wave and gravity wave diagnostics of Kinoshita and Sato (2013)." (line 8, page 16). Scientific conclusions are essentially absent. Again, it must be stressed that the results of the paper do not allow any conclusions with respect to propagation as only one time is considered and very simplifying assumptions are made.

We carefully revised the title, abstract and discussion considering the reviewer's suggestions. We concentrate in this paper on the diagnosis of GW appearance and hint on an upcoming publication building up on this initial results for further studies on propagation and generation.

Typos:

inhomogenous -> inhomogeneous (line 1 in Abstract)
artifical -> artificial (page 9, line 24)
". wave action .." -> ". Wave action ę" (page 8, line 28)
imhogeneous -> inhomogeneous (page 8, line 27)
All typos corrected.

**A. Diagnosis of the Zimin Test Case**

**Diagnosis of the Zimin test case**

V1R1 (25 Jan 2018 CZ) first draft

This is a technical report on the setup and application of "uwadi3figs.prj" which diagnoses the Zimin test case. This includes

- HIL (UWaDi): Hilbert transform as used with UWaDi
- STW (3DS): Stockwell transform which is also used in three dimensions
- SIN (S3D): Sinusoidal fit as is the base for three-dimensional sin-fit
- ACF (DIV, HDA): Auto-correlation function (so-called divergence method or Harmonic Divergence Analysis)

**1 Setup for the second submission (October 2017)**

For all the methods, we have used a box length of

L = 7.85 (1)

The reason for this choice was oriented on experiences with ACF: For the maximum lag we allow for one fifth of the total length, and this should cover one wavelength

$$\lambda \le \frac{L}{5} \tag{2}$$

and hence

$$L \ge 5\lambda$$
 (3)

For the example we have used wavenumbers 4 and 9, which corresponds to a longest wavelength of

 $\lambda \le 1.6 \tag{4}$

and hence

$$L > 8.0 \tag{5}$$

While the wavepackets are placed at x = 4.5 and x = 7.5 their distance is 3 which is obviously smeared out with such a wide window. This is the major reason for the relative bad performance of the box-using methods SIN and ACF.

The choice of the box length is a matter of compromise between accuracy in space and wavenumber: While the spatial uncertainty is given by  $\delta x = L$ , the basic wavenumber is about  $\delta k = 2 \pi/L$  and hence

$$\delta x \ \delta k = 2 \ \pi \tag{6}$$

Insofar the optimal box width is a matter of compromise and needs to be decided from case to case.

In the following we reconsider each diagnosis method and report on several sensitivity experiments.

**2 Hilbert transform (HIL / UWaDI)**

The Hilbert transform is a continuous method working without any box parameter. It returns values for every space point. Only some numerical trouble with the wavenumbers may occur for weak signals.

**3 Stockwell transform**

We start from the formulation of Wright *et al.* (2017) which is for the threedimensional application. Reducing it back to our one-dimensional test case we retain the localization parameter (or dimensionless boxwidth factor) which is the width of the Gauss window

$$L = \frac{C_L}{f} = C_L \lambda \tag{7}$$

In the paper, a  $C_L$  value of 0.25 has been used for analysis of AIRS data in the horizontal dimension and 0.1 for the vertical dimension. In the conventional formulations (Wright 2010) this parameter is included as 1.0. These different choices have been tested – see Figure 1. As expected, a large value of the boxwidth factor result in smeared distributions with a precise wavenumber, while a smaller value returns accurate amplitudes and more suspect wavenumbers. Based on these findings we choose a *C* value of 0.25 as we have done it for the UWaDi-2.

| (a) | (b) |
|-----|-----|
|-----|-----|

---

## Author Response (AR3)

**Reply to minor revision on acp-2017-472**

L. Schoon and Ch. Zülicke

April 25, 2018

We thank the Co-Editor and the Reviewer #1 for their comments on the minor revision of the manuscript acp-2017-472.

In the following, we reply on all given comments.

The manuscript with the highlighted changes is attached to this reply, a clean manuscript is provided in an extra file.

**1. Reply on Review of Co-Editor**

Thanks very much for your thorough revision and in particular for addressing all of my editorial recommendations. This revision has now been evaluated by myself and one of the original reviewers. I am pleased to say that only a few remaining minor comments by reveiwer #1 need to be addressed before your manuscript will be ready for acceptance. I am looking forward to receiving the final revised manuscript. We thank you for the evaluation of our revision.

**2. Reply on Comments of Reviwer #1**

The authors have taken my major comments into account and I have no further comments to the content. A few points on language / clarity, which the authors may want to consider, are given below.

Thank you for your patience throughout the review process. Your comments were always helpful and constructive. We are pleased to hear that there are no further comments on the content.

P1L7 Resolve the 'It', e.g. This distribution is "It"-> "The local wave characteristics"

P2L5 why "binned"? From Wikipedia: Data binning or bucketing is a data pre-processing technique used to reduce the effects of minor observation errors. The original data values which fall in a given small interval, a bin, are replaced by a value representative of that interval, often the central value. As such I would not call the division of the original data into finite analysis volumes "binning".

Suggestion: was created for the analysis of three-dimensional data from remote sensing observations

We follow your suggestion in the new manuscript.

P2L17 Suggestion: It does not need to rely on choosing the size of an analysis volume aforehand.

We follow this suggestion.

P2L22 structure can Space inserted.

P3L20 locate the occurrence of GWs We follow this suggestion.

P3L27 MS-GWaves is a DFG Research Unit, but ROMIC a "Foerdermassnahme" of the ministry. My suggestion is to stay here on the project level and disentangel this cleanly in the acknowledgements.

We avoid to disentangle this in the acknowledgements because we are not part of this projects and do not want to cause confusion by stating those at the end of our paper. We reformulated this passage to give the right affiliations but avoid running into trouble by using words like "research unit".

P4L6 counts for -> isChanged accordingly. P4L8 Suggestion: Studying mountain waves, Doernbrack et al. (2012) and Khaykin et al. (2015) have shown that the divergence values correlate with the GW-induced temperature anomalies. Is one of the major advantages of using wind divergences that you do not need to explicitly calculate residuals which involves some technique to estimate a slowly varying atmospheric background and hence introduces uncertainties. If so, please state this here!

This passage was changed following your suggestions. Additional information on the choice of the horizontal divergence is added.

*P4L9 appearance -> occurrence* Changed

P4L10 The importance of oblique propagation of GWs in a general context was discussed, for instance, by ... Here we point out the important role ... This passage was reformulated and hopefully, appears to be more clearer now.

P4L14 New paragraph Paragraph added

P4L32 Not clear what you mean. Perhaps?: This way the method unifies wave parameter information with unreduced spatial resolution, hence the name Unified Wave Diagnostics. ?

I imagine there is a misunderstanding here. The sentence was rewritten. Maybe it also helps to state that there are two slightly different meanings of the word "unified". We do not mean the German translation "vereinigt" but the meaning "vereinheitlicht". With the new formulation this should be clearer.

P5L16 Please specify what you mean by retrievals.

Retrieval was certainly the wrong word because it is used mainly in the area of satellite observations. We changed it to "ECMWF analyses", what was meant here.

P5L17 That is colloquial However, adapting step 3 also other variables can be processed. This passage was rewritten.

P5L13 "The method is based on a Hilbert transform and returns an estimate for each

data grid point, thus, avoiding the use of pre-defined boxes for the analysis." Actually, you can run an S transform or an S3D also to provide estimates for each grid point, but still these methods use predefined box sizes, so better make that two points.

We changed this to two points and added the fact that our method is computational cheap.

P16L1 for a synthetic \*one dimensional\* test case. Added

P16L3 Their is an ambiguity in the sign of the wavenumber (direction of the wavevector), which, however, is the case ... Still, the method proves to be a reliable ... Changed according to your suggestion.

P16L10 ... the specific approach ?is well working? ?provides good results at low complexity? optimal would mean that nothing better can be found which you have not shown Rewritten following your latter suggestion.

One more advantage you might want to mention is that UWADI is comparably fast (S transform and S3D are quite computationally expensive) See four points above.

**A. Revised Manuscript**

[revised manuscript text omitted]